# Small molecule disruption of RARα/NCoR1 interaction inhibits chaperone-mediated autophagy in cancer

Mericka McCabe [1,2,3,4], Rajanya Bhattacharyya[1,3,4], Rebecca Sereda [2,3,4], Olaya Santiago-Fernández [2,3,4], Rabia R Khawaja [2,3,4], Antonio Diaz [2,3,4], Kristen Lindenau [2,3,4], Deniz Gulfem Ozturk [1,3,4], Thomas P Garner[1,3,4], Simone Sidoli [1,3,4], Ana Maria Cuervo [2,3,4,5✉] & Evripidis Gavathiotis [1,3,4,5,6✉]

## Abstract

**Chaperone-mediated autophagy (CMA), a type of selective degradation of cytosolic proteins in lysosomes, is commonly upregulated in cancer cells, contributing to their survival and growth. The lack of a specific target for CMA inhibition has limited CMA blockage to genetic manipulations or global lysosomal function inhibition. Here, using genetic modulation, transcriptional analysis, and functional studies, we demonstrate a regulatory role for the interaction of the retinoic acid receptor alpha (RARα) and its corepressor, the nuclear receptor corepressor 1 (NCoR1), on CMA in non-small cell lung cancer (NSCLC). By targeting the disruption of the NCoR1/RARα complex with a structure-based screening strategy, we identified compound CIM7, a potent and selective CMA inhibitor that has no effect on macroautophagy. CIM7 preferentially inhibits CMA in NSCLC cells over normal cells, reduces tumor growth in NSCLC cells, and demonstrates efficacy in an in vivo xenograft mouse model with no observed toxicity in blood or major tissues. These findings reveal a druggable mechanism for selective CMA inhibition and a first-in-class CMA inhibitor as a potential therapeutic strategy for NSCLC.**

**Keywords** Autophagy; CMA Inhibitor; NCoR1; Non-small Cell Lung Cancer; RARα

**Subject Categories** Autophagy & Cell Death; Cancer; Pharmacology & Drug Discovery

## Introduction

Autophagy is the cellular process of lysosomal degradation and recycling of cytosolic components, providing a cellular quality control mechanism and allowing for maintenance of cellular homeostasis (Jafari et al, 2022; Kaushik and Cuervo, 2018). In mammalian cells, autophagy exists as three main types: macro-autophagy, endosomal microautophagy, and chaperone-mediated autophagy (CMA). CMA is a selective form of autophagy in which proteins containing a KFERQ-like targeting motif are recognized by the chaperone protein heat shock cognate of 71 kDa (Hsc70), delivered to the lysosome-associate membrane protein type 2A (LAMP2A) at the surface of the lysosome, unfolded and internalized by the CMA translocation complex, and subsequently degraded (Jafari et al, 2022; Kaushik and Cuervo, 2018). CMA is activated in response to nutritional deprivation, oxidative stress, DNA damage, and hypoxia and CMA upregulation has been found to contribute to the survival and proliferation of most cancers, including NSCLC, supporting tumor growth and metastasis (Arias and Cuervo, 2020; Dong et al, 2025; Ichikawa et al, 2020; Kon et al, 2011). CMA enhances survival in various cancers through direct and indirect regulation of oncogenic proteins and pathways including, but not limited to, tumor-associated mutant p53, pro-apoptotic BBC3/PUMA, and oncogenic c-MYC, stabilization of the pro-survival protein MCL1, and sustained aerobic glycolysis (Warburg effect) (Gomes et al, 2017; Kon et al, 2011; Suzuki et al, 2017; Vakifahmetoglu-Norberg et al, 2013; Xie et al, 2015). In contrast, CMA decline has been found to contribute to various conditions including neurodegenerative and cardiometabolic diseases (Kaushik and Cuervo, 2018).

Despite the growing evidence of the role of CMA upregulation in cancer, a targetable mechanism that is selective for CMA inhibition is lacking. Targeting CMA, compared to macroautophagy, allows for a more selective modulation of autophagy focused on protein substrates. In line with this, macroautophagy dysregulation has been variably implicated as both a tumor suppressor and tumor promoter in various cancer types (Guo et al, 2022; White, 2012, 2015). Thus, while non-specific macroautophagy inhibitors (e.g., chloroquine, 3-methyladenine) and more direct small molecule modulators targeting macroautophagy regulators have been developed, these compounds and targeting strategies

[1]Department of Biochemistry, Albert Einstein College of Medicine, Bronx, NY 10461, USA. [2]Department of Developmental and Molecular Biology, Albert Einstein College of Medicine, Bronx, NY 10461, USA. [3]Montefiore Einstein Comprehensive Cancer Center, Albert Einstein College of Medicine, Bronx, NY 10461, USA. [4]Institute for Aging Research, Albert Einstein College of Medicine, Bronx, NY 10461, USA. [5]Department of Medicine, Albert Einstein College of Medicine, Bronx, NY 10461, USA. [6]Cancer Dormancy Institute, Albert Einstein College of Medicine, Bronx, NY 10461, USA. ✉E-mail: ana-maria.cuervo@einsteinmed.edu; evripidis.gavathiotis@einsteinmed.edu

have been limited in efficacy and safety (Guo et al, 2022; White, 2012, 2015). This is because they either completely disrupt all types of lysosomal degradation or affect additional cellular pathways.

We previously identified that retinoic acid receptor alpha (RARα), a ligand-dependent transcription factor, negatively regulates the expression of CMA effectors and modulatory proteins in non-transformed cells (Anguiano et al, 2013). Transcriptional activity of RARα and its active conformation is regulated by the binding of transcriptional co-repressors and co-activators. We have previously shown that the nuclear receptor co-repressor 1 (NCoR1), a transcriptional co-repressor of RARα, positively regulates CMA activation in non-transformed cells (Gomez-Sintes et al, 2022). These findings informed us about the significant contribution of the NCoR1/RARα complex to CMA upregulation. However, to date, this NCoR1/RARα-dependent regulation of CMA has not been established in cancer cells, nor has it been determined whether it can impact tumor growth. Furthermore, CMA inhibition has been achieved through genetic manipulation such as LAMP2A knockdown or knockout, or using nonselective inhibitors, which often inhibit total lysosomal degradation (Arias and Cuervo, 2020; Chude and Amaravadi, 2017). Thus, a small molecule CMA inhibitor would be a valuable chemical probe for further research and potential cancer therapeutic development.

In this work, we have investigated the cellular levels and functional interaction of NCoR1/RARα in relation to CMA regulation, focused on human NSCLC primary samples and cell lines. Furthermore, we sought to determine the impact of escaping the NCoR1/RARα interaction on CMA activity. Having established this, we set out to identify small molecule inhibitors targeting the NCoR1/RARα interaction using available crystal structures of RARα bound to NCoR1 and to steroid receptor coactivator (SRC). This involved in silico screening and experimental validation assays. We identified CMA Inhibitory Molecule 7 (CIM7), a potent inhibitor of CMA, which functions by disrupting the NCoR1/RARα interaction and altering the CMA regulatory transcriptional program to suppress CMA. We provide evidence that CIM7 functions via a unique mechanism compared to established RARα agonists and does not affect CMA in non-tumorigenic cells or activate macroautophagy in NSCLC, thus conferring its selectivity for both CMA and as a specific anti-cancer therapeutic. Considering the importance of CMA on tumor growth and survival, we utilized a NSCLC xenograft mouse model and demonstrated that in vivo administration of CIM7 results in marked reduction of tumor growth without noticeable toxicity. Through our studies, we have established a novel druggable CMA regulatory complex in NSCLC, identified a first-in-class selective CMA inhibitor by targeting this complex, and provided proof of concept for pharmacological inhibition of CMA as an anti-cancer therapeutic strategy in NSCLC.

# Results

## The NCoR1/RARα complex is upregulated in NSCLC and regulates CMA

We previously developed a fluorescent CMA reporter (KFERQ-PS-Dendra) to visualize and quantify CMA activity as the number of fluorescent puncta per cell, when the fluorescent CMA substrate protein is trafficked to the lysosome and changes from a diffuse distribution to a punctate pattern (Dong et al, 2020) (Fig. 1A, left schematic). We selected five non-small cell lung cancer (NSCLC) cell lines (H520, H23, H460, A549, and H1703) with differential

status for common mutated genes in lung cancer (Fig. EV1A) and one non-tumorigenic lung cell line (BEAS-2B). We found increased CMA activity in four of the five cancer cell lines compared to BEAS-2B cells (Fig. 1A). Trafficking of the reporter to lysosomes was confirmed by colocalization of Dendra with the lysosomal protein LAMP2A (Appendix Fig. S1A). By immunoblot, LAMP2A protein levels, the rate-limiting CMA component, increased proportionally to the increase in CMA activity observed in each of the cancer cell lines (Fig. EV1B). Further probing into the potential mechanism of action behind CMA activity upregulation was fast-tracked by expanding the study to the network of genes that participate in CMA (effectors and regulators) and using an algorithm that predicts CMA activity (CMA score) from the expression values of the CMA network genes (Bourdenx et al, 2021). Utilizing publicly available transcriptomics, we found an increase in both CMA score and NCoR1/RARα expression ratio in NSCLC cell lines, compared to non-tumorigenic lung cell lines (Fig. EV1C–F; Appendix Table S1).

Consistent with published transcriptomics, qPCR analysis of the CMA network components revealed that CMA score was significantly upregulated in our NSCLC cell line panel compared to BEAS-2B cells (Fig. 1B; Appendix Fig. S1B). We previously reported that enhanced interaction of RARα and its co-repressor NCoR1, promoted by small molecules, induced upregulation of CMA in non-transformed cells by suppressing the inhibitory effect of RARα on CMA (Gomez-Sintes et al, 2022). This prompted us to investigate whether cancer cells may also use the RARα/NCoR1 axis to sustain their elevated CMA activity. Analysis of RARα and NCoR1 nuclear protein levels by immunofluorescence demonstrated significantly higher NCoR1 and RARα nuclear content in all NSCLC cell lines compared to BEAS-2B cells (Fig. 1C; full fields in Appendix Fig. S1C). To investigate the contribution of NCoR1 and RARα to the upregulation of CMA in NSCLC cells, we modulated their expression levels and interaction. RNA interference against RARα in A549 cells resulted in a significant increase in CMA activity (Fig. 1D; efficiency of the knockdown (KD) is shown in Fig. EV2A). Treatment with the RAR ligand ATRA, which activates RARα transcriptional signaling, inhibited CMA in cancer cells (Figs. 1E and EV2B). Similarly, an established and selective RARα agonist, AM580, inhibited CMA in cancer cells (Figs. 1F and EV2C). Furthermore, NCoR1 KD in cancer cells resulted in significantly reduced CMA activity (Fig. 1G; efficiency of KD is shown in Fig. EV2D). To evaluate the impact of directly altering the NCoR1/RARα interaction on CMA, we introduced an RARα mutant incapable of binding NCoR1 (AHT RARα) (Grignani et al, 1998; Hörlein et al, 1995; Khan et al, 2001). CMA activity was significantly reduced in A549 cells overexpressing the AHT RARα mutant compared to cells expressing wild-type RARα (Figs. 1H and EV2E). Taken together, our data establish the NCoR1/RARα interaction as a key positive regulator of CMA activity in NSCLC cells and highlight the inhibitory effect of disrupting the NCoR1/RARα interaction on CMA.

## Identification of a novel potent and selective CMA inhibitor

While we found that RARα agonists can inhibit CMA in cancer cells (Fig. 1E,F), both ATRA and AM580 are also capable of inducing macroautophagy in certain conditions, including breast

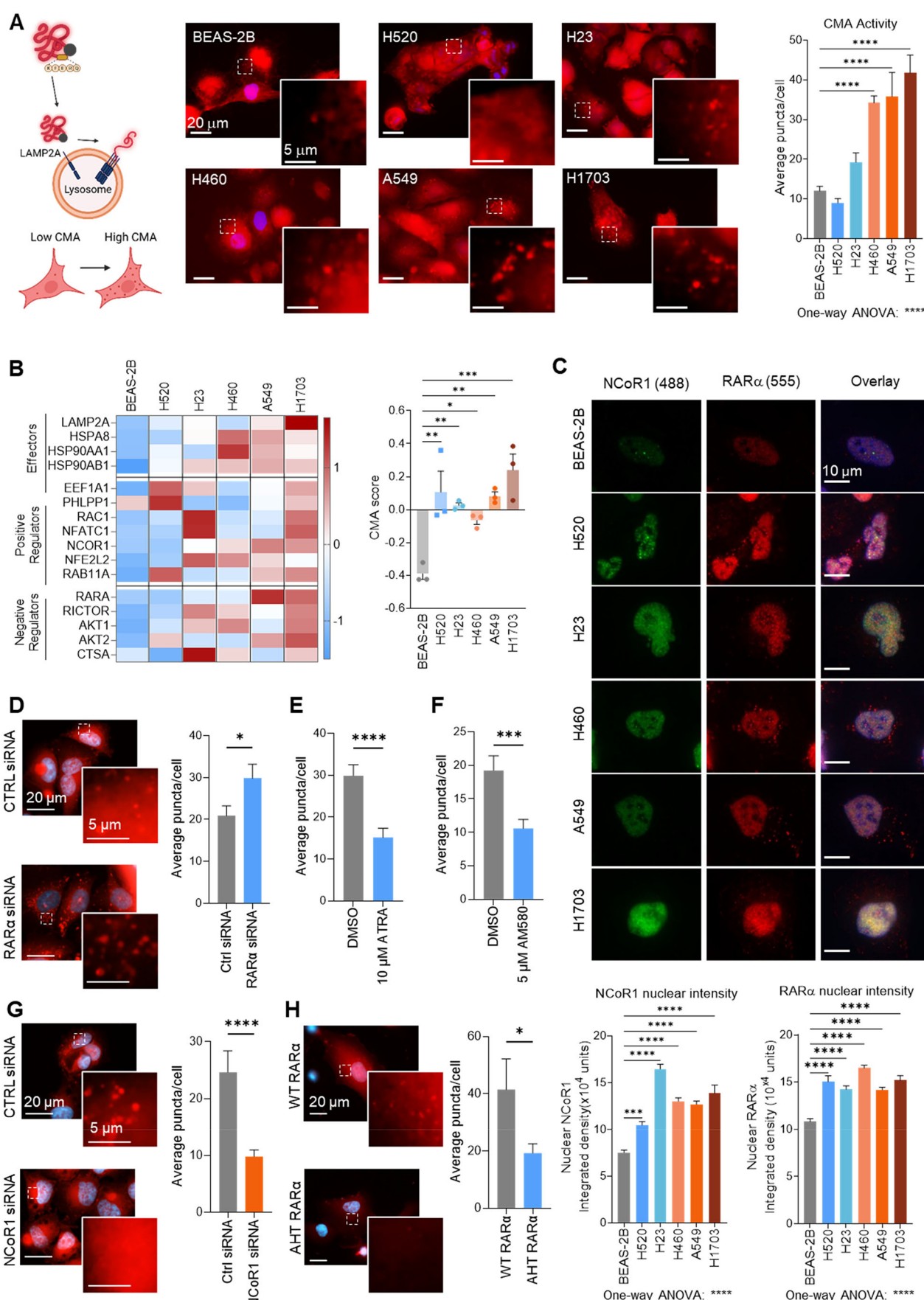

◄ **Figure 1. CMA activity in NSCLC is regulated by NCoR1/RARα.**

(A) CMA activity in 5 human NSCLC cell lines and lung epithelial BEAS-2B cells. CMA activity was measured using the KFERQ-PS-Dendra reporter assay (left) and quantified as fluorescent puncta per cell (right). Representative images (center) of cells expressing KFERQ-PS-Dendra in red and nuclei highlighted with DAPI. Inserts show higher magnification of the red channel. $n \geq 105$ cells in three independent experiments. (****$P \leq 0.0001$). Created in BioRender. https://BioRender.com/a8dxxz3. (B) Heatmap of the transcriptional differences of CMA-related genes (left) and calculated CMA score (right) in the same cell lines as (A). $n = 3$ independent experiments. (**$P = 0.0013$, **$P = 0.0051$, *$P = 0.0283$, **$P = 0.0019$, ***$P = 0.0001$). (C) Immunofluorescence staining for NCoR1 and RARα in the indicated cell lines. Top: representative images of single or merged (overlay) channels with nuclei highlighted with DAPI. Full field images are shown in Appendix Fig. S1C). Bottom: quantification of nuclear intensity for each protein. $n \geq 158$ cells in three independent experiments (***$P = 0.0004$, ****$P \leq 0.0001$). (D) CMA activity in A549 cells expressing KFERQ-PS-Dendra upon transfection with control (Ctrl) siRNA or siRNA targeting RARα. Representative images (left) as in (A) and quantification of fluorescent puncta per cell (right). $n \geq 212$ cells from six independent experiments (*$P = 0.0317$). (E) CMA activity in A549 cells expressing KFERQ-PS-Dendra and maintained in serum-free media supplemented with 10 μM ATRA or equal volume vehicle (DMSO). $n \geq 87$ cells from three independent experiments. Representative images are shown in Fig. EV2B. (****$P \leq 0.0001$). (F) CMA activity in A549 cells expressing KFERQ-PS-Dendra and maintained in media supplemented with 5 μM AM580 or equal volume vehicle (DMSO). $n \geq 72$ cells from two independent experiments. Representative images are shown in Fig. EV2C. (***$P = 0.0007$). (G) CMA activity in A549 cells expressing KFERQ-PS-Dendra upon transfection with control (Ctrl) siRNA or siRNA targeting NCoR1. Representative images (left) as in (A) and quantification of fluorescent puncta per cell (right). $n \geq 124$ cells from four independent experiments. (****$P \leq 0.0001$). (H) CMA activity in A549 cells expressing the KFERQ-PS-Dendra upon overexpression of either wild-type (WT) RARα or the AHT RARα mutant. Representative images (left) as in (A) and quantification of number of fluorescent puncta per cell (right) $n \geq 45$ cells from three independent replicates (*$P = 0.0492$). Data information: All values are mean + SEM, with individual data points when $n < 10$. Ordinary one-way ANOVA followed by Bonferroni's multiple comparisons post-hoc test (A–C) or unpaired two-tailed $t$ test (D–H) were used. Source data are available online for this figure.

cancer (Brigger et al, 2015) and NSCLC (Fig. EV2F–H shows increased number of autophagic vesicles and autophagic flux with either treatment). Additionally, ATRA has known off-target transcription-independent effects on ERK and PI3k-Akt signaling pathways in NSCLC that counteract RARα-mediated inhibition of cancer cells growth and make ATRA a poor candidate compound for NSCLC treatment (Choi et al, 2007; Garcia-Regalado et al, 2013; Quintero Barceinas et al, 2015). Given that ATRA and AM580 bind to the RARα active conformation and are non-selective CMA inhibitors, and that disruption of NCoR1/RARα interaction inhibits CMA in cancer cells (Fig. 1H), we reasoned that small molecules that directly reduce the NCoR1/RARα interaction would cause a more selective inhibition of CMA activity. NCoR1 binds RARα in a site adjacent to the ligand binding domain when RARα is in the inactive conformation, however, when RARα is in the active conformation this site can be bound by the SRC coactivator (le Maire et al, 2010). To identify small molecules that disrupt the NCoR1/RARα interaction for selective CMA inhibition, we performed high-throughput docking screens using a RARα co-crystal structure in the inactive conformation and bound to NCoR1 helix as well as a RARα co-crystal structure in the active conformation and bound to SRC helix (Fig. 2A; Appendix Fig. S2). Docking screens with Glide (Schrödinger, LLC) (Friesner et al, 2006) and subsequent ranking and filtering of compounds resulted in 94 compounds predicted to bind a RARα binding site outside of the canonical RARα ligand binding site that is typically occupied by ATRA and agonists such as AM580 (Fig. 2A; Appendix Fig. S2). Using the fluorescent CMA reporter assay in NIH-3T3 fibroblasts, we screened all 94 compounds (Appendix Table S2) at 10 μM for 24 h and identified 16 potential CMA inhibitors (Fig. 2B), 6 of which inhibited CMA in a dose-dependent manner after removal of serum from the culture media to activate CMA in fibroblasts to the levels observed in cancer cells (Fig. EV3A). All identified hit compounds were originally predicted to bind the inactive NCoR1-bound RARα conformation from the in silico screening, except for G11, which was predicted to bind the active SRC-bound conformation. These compounds were then tested in A549 cells at 1 μM for 24 h. Compound CMA Inhibitory Molecule 7 (CIM7) showed the best potency for CMA inhibition, based on a more

diffuse red signal and reduced number of puncta per cell (Fig. 2C,D). CIM7 has not been reported as a hit in previous screens in the PubChem database. Analysis of CIM7 structure by QikProp (Schrödinger, LLC) revealed promising drug-like and ADME properties (Fig. EV3B).

CIM7 proved to be a potent inhibitor of CMA with an $IC_{50}$ around 75 nM after 24 h of treatment in A549 cells (Fig. 2E), significantly lower than the concentrations required for CMA inhibition by ATRA or AM580 (Fig. 1E,F). We further confirmed the ability of CIM7 to inhibit CMA activity in additional NSCLC cell lines, H1703 and H23, which harbor different Kras and TP53 mutation statuses (Figs. EV3C and EV1A). CMA serves important functional roles in non-tumorigenic cells, making it necessary to identify a therapeutic window at which CIM7 could potently inhibit CMA in only NSCLC cells. Notably, at the maximal tested concentration (5 μM), CIM7 had no inhibitory effect on CMA activity in various non-tumorigenic cell lines (Fig. 2F; Appendix Fig. S3A). As a second complementary approach to confirm the inhibitory effect of CIM7 on CMA, we directly analyzed CIM7 effect on the lysosomal degradation of the artificial CMA substrate (KFERQ-Dendra) using biochemical procedures. Degradation of KFERQ-Dendra by CMA can be monitored by immunoblot based on the accumulation of this protein upon blocking lysosomal proteolysis with ammonium chloride and leupeptin (Dong et al, 2020). We found that cells treated with CIM7 display higher basal levels of KFERQ-Dendra protein than untreated cells due to a blockage in protein degradation, because, contrary to untreated cells, inhibition of lysosomal proteolysis in CIM7 treated cells does not result in a further increase in KFERQ-Dendra levels under these conditions (Fig. 2G). Next, we examined whether transcriptional changes induced by CIM7 are consistent with CMA inhibition. Indeed, we identified changes in the expression of CMA-related genes that result in a significant reduction in the transcriptional CMA score of A549 cells within just 1 h of CIM7 treatment at 5 μM, which ensures saturation of CMA inhibition (Fig. 2H; Appendix Fig. S3B). Notably, after 3 h of CIM7 treatment, CMA score is reduced in A549 cells, but not non-tumorigenic BEAS-2B cells, consistent with our finding that CIM7 CMA inhibitory effect has a therapeutic window (Fig. EV3D). Compared

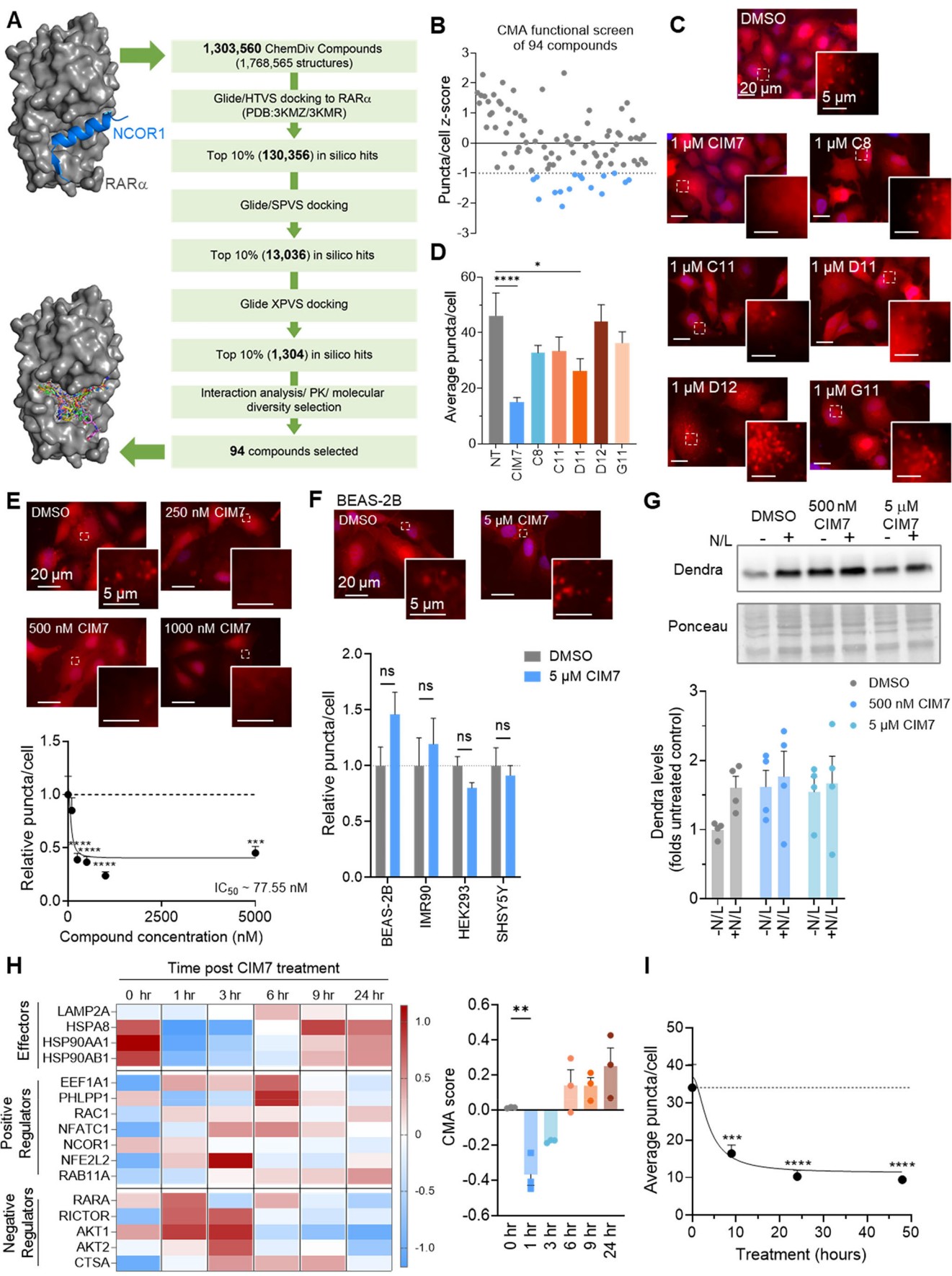

◄

**Figure 2.  Identification of a novel, potent CMA inhibitor CIM7 that functions through transcriptional regulation of CMA network genes in NSCLC.**

(A) Flowchart of the computational screen to identify small molecules predicted to bind the inactive (NCoR1-bound, PDB ID: 3KMZ) or active (SRC-bound, PDB ID: 3KMR) conformation of RARα. Structure of NCoR1-bound RARα used in the screen and docked compounds bound into NCoR1 site are shown on the left. (B) CMA activity in mouse fibroblasts (NIH-3T3 cells) expressing KFERQ-PS-Dendra, 24 h after treatment with 10 μM of compounds identified from (A), with each data point representing an individual compound. $n = 1500$–4000 cells imaged across nine fields. CMA activity is shown as z-score of the fluorescent puncta per cell values with potential inhibitors identified by z-score < −1 (blue dots below dashed line). (C, D) CMA activity in A549 cells expressing KFERQ-PS-Dendra treated with 1 μM of each putative CMA inhibitor compound or DMSO for 24 h. Representative images (C) and quantification of fluorescent puncta per cell (D). $n \geq 40$ cells. (****$P \leq 0.0001$, *$P = 0.0302$). (E) CMA activity in A549 cells expressing KFERQ-PS-Dendra treated with increasing concentrations of CIM7. Representative images (top) and quantification of fluorescent puncta per cell (bottom). $n > 96$ individual cells from three to four independent experiments. (****$P \leq 0.0001$, ***$P = 0.0009$). (F) CMA activity in non-cancer cells treated with 5 μM CIM7 or equal volume DMSO. Representative image of BEAS-2B cells (top; Images for other cell lines are in Appendix Fig. S3A) and quantification of fluorescence puncta per cell (bottom). $n > 20$ individual cells. (ns = not significant). (G) Representative immunoblot for Dendra in A549 cells treated with the indicated concentrations of CIM7 or DMSO in the absence or presence of ammonia chloride and leupeptin (N/L) (top) and densitometric quantification (below). Ponceau staining is shown as loading control. $n = 4$ independent experiments. (H) Heatmap of the z-scores of the transcriptional differences of CMA-related genes (left) and calculated CMA score (right) in A549 cells at the indicated times after addition of 5 μM CIM7. $n = 3$ independent experiments. (**$P = 0.0061$). (I) Time course of changes in CMA activity calculated by number of fluorescent puncta per cell in A549 cells treated with 5 μM CIM7. $n > 95$ cells from three independent experiments. (***$P = 0.0008$, ****$P \leq 0.0001$). Data information: All values are mean + SEM, with individual data points in bar graphs when $n < 10$. Ordinary one-way ANOVA followed by Bonferroni's multiple comparisons post-hoc test (D, E, H, I), or non-linear regression (E, I), or multiple *t* test (F) were used. Source data are available online for this figure.

to the early CIM7-induced transcriptional changes, the inhibitory effect on CMA activity in A549 cells, measured with the fluorescent reporter, becomes noticeable with some delay (by 9 h), as expected from the usually long half-life of the lysosomal proteins that participate in CMA (Figs. 2I and  EV3E). For the same reason, while expression of some components of the CMA network returns to normal levels by 6 h after treatment with CIM7 (Fig. 2H), the inhibition of CMA activity persists to at least 48 h (Figs. 2I and EV3E). Importantly, we determined that CIM7 induces selective CMA inhibition, contrary to ATRA and AM580 (Fig. EV2G,H), as addition of 5 μM CIM7 to A549 cells did not result in significant changes on macroautophagy activity when analyzed with the fluorescent tandem reporter mCherry-GFP-LC3 (Kimura et al, 2007) (Fig. EV3F shows no changes in total autophagosome content or in their maturation to autolysosomes) or by monitoring lysosomal degradation of LC3-II by immunoblot (Fig. EV3G). In summary, CIM7, which we identified based on rational targeting of the NCoR1/RARα interaction, is a first-in-class selective CMA inhibitor capable of potently inhibiting CMA in nanomolar concentrations through transcriptional regulation of key CMA components in NSCLC and not in non-tumorigenic cell lines.

## CIM7 disrupts the NCoR1/RARα interaction

Given the initial identification of CIM7 through an in silico docking screen with the NCoR1-bound RARα structure, we utilized molecular dynamics simulations (Bowers et al, 2006) to evaluate its binding pose. Simulations showed CIM7 to quickly shift within its initial RARα binding pocket, allowing for compound stabilization through π-π stacking interactions of its phenyl group with Phe228 and Phe302 residues, hydrogen bond interactions of its amide with Ala300 and Ser232, of its hydroxymethyl with Gly391, and of its imidazole with Trp225 (Fig. EV4A; Appendix Fig. S4A–C). Additional stabilization is achieved through van der Waals contacts of the trifluoromethoxy with Met284, Phe286, Met297, Ala300 and of the fluorobenzyl with Leu305, Met400, Leu266 (Fig. EV4A). In turn, this binding induces conformational changes over the course of 1000 ns that would shift RARα from the inactive conformation to an active conformation (helix H11 formed) that is preventive of NCoR1 binding (Fig. 3A; Appendix Fig. S4A–C) (le Maire et al,

2010). CIM7 directly binds recombinant RARα with a Kd of approximately 2 μM (Fig. 3B; Appendix Fig. S4D), as illustrated through isothermal titration calorimetry experiments. While this affinity is higher than the $EC_{50}$ of CMA inhibition, that may be, in part, due to differences in the conformation and interaction partners of full-length cellular RARα compared to recombinant RARα, which is limited to the protein's ligand binding domain. Moreover, we performed molecular dynamics simulations of CIM7 binding to RARα in the presence of NCoR1 and found that CIM7 is predicted to bind RARα outside of the canonical ligand binding pocket when NCoR1 is present and induce conformational changes that result in the dissociation of NCoR1 already bound to RARα (Fig. 3C,D; Appendix Fig. S4E–G). The capacity of CIM7 to compete NCoR1 binding was confirmed by fluorescent polarization anisotropy assays in which RARα was titrated into NCoR1 fluorescent peptide and the addition of CIM7 resulted in a reduced binding affinity of NCoR1 for RARα (Fig. 3E). Intriguingly, CIM7 does not alter the binding of the coactivator SRC to RARα (Fig. 3F). Conversely, the established agonists ATRA and AM580 do not have any effect on NCoR1 binding to RARα (Fig. 3G; Appendix Fig. S5A) and rather enhance binding of the coactivator, SRC (Fig. 3H; Appendix Fig. S5B), suggestive that CIM7 is not a typical RARα agonist, but rather a different class of compound that selectively regulates only co-repressor NCoR1 binding to RARα (Fig. 3I). This differential regulation of CIM7 on the co-repressor rather than coactivator, consistent with its distinct predicted RARα binding site and shift from the inactive to the active conformation, may explain the selectivity of CIM7 for CMA inhibition.

Finally, we evaluated several analogs of CIM7 with a different functional group than the trifluoromethoxy phenyl group (CIM7.1, CIM7.2 and CIM7.3) or without the hydroxymethyl group (CIM7.4) (Fig. EV4B). CIM7.1, CIM7.3, and CIM7.4 were still capable of CMA inhibition, while CIM7.2 had no significant effect on CMA activity (Fig. EV4C). CIM7.1, CIM7.3, and CIM7.4 all exhibit direct binding to recombinant RARα, although with less affinity compared to the parental CIM7 molecule (Fig. EV4D), which may also explain the lack of improved efficacy for CMA inhibition. These findings suggests a better fit of the trifluoro-methoxy phenyl group in the narrow ligand pocket, as well as little contribution of the hydroxymethyl to the RARα binding. Based on

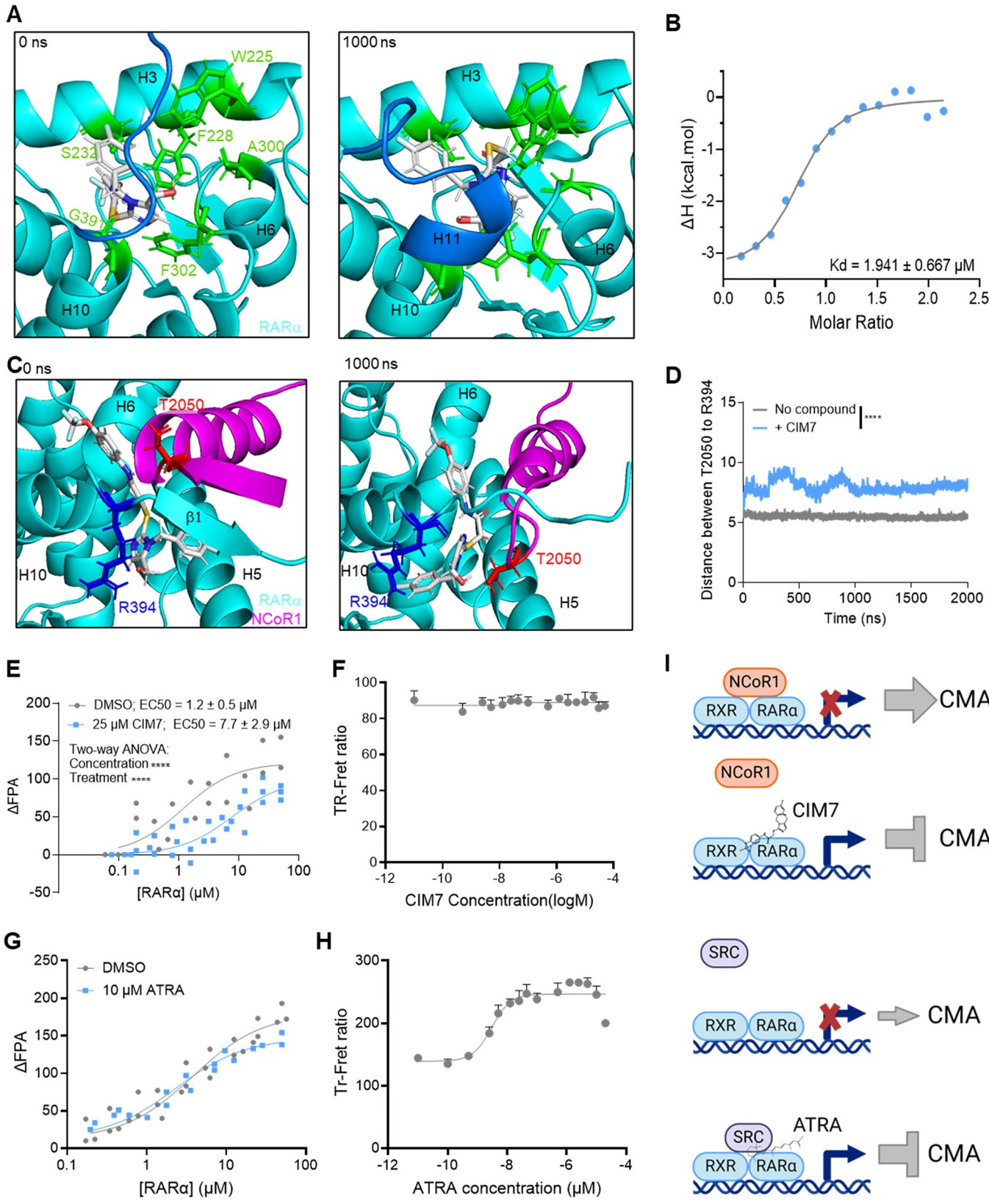

**Figure 3.    CIM7 modulates the NCoR1/RARα interaction.**

(A) Molecular dynamic simulations of CIM7 (white sticks) bound to the inactive RARα (cyan and blue, with sidechains of interacting residues in green) in the absence of NCoR1 peptide reveals a shift from the inactive conformation (left, 0 ns) to the active RARα conformation (right, 1000 ns) based on helix H11 (blue). (B) Representative isothermal titration calorimetry curve of CIM7 with recombinant RARα LBD showing enthalpy (ΔH) with increasing molar ratio. This experiment was repeated five times with consistent results. (C) Molecular dynamic simulations of CIM7 (white) bound to the inactive RARα (cyan) in the presence of NCoR1 peptide (magenta) (top, 0 ns) reveals a disruption of the b-sheet/b-sheet interaction necessary for NCoR1 to bind RARα conformation over time (bottom, 1000 ns). (D) The predicted distance from T2050 Cα atom on NCoR1 (red in C) and R394 Cα atom on RARα (blue in C) in simulations with CIM7 present or absent. $n = 3$–4 individual simulations. (****$P \leq 0.0001$). (E) The $EC_{50}$ (μM) of NCoR1 binding to RARα, determined by fluorescence polarization anisotropy (FPA) with RARα and the NCoR1 peptide in the presence of 25 μM of CIM7 or DMSO. $n = 2$–3 technical replicates. (****$P \leq 0.0001$). (F) TR-FRET evaluation of SRC binding to recombinant RARα with increasing concentrations of CIM7. $n = 4$ technical replicates. (G) FPA with RARα and the NCoR1 peptide in the presence of 10 μM ATRA or DMSO. $n = 2$–3 technical replicates. (H) TR-FRET evaluation of SRC binding to recombinant RARα with increasing concentrations of ATRA. $n = 4$ technical replicates. (I) Schematic representation of the differences between CIM7 and ATRA effect on coregulator binding to RARα. Created in BioRender. https://BioRender.com/qbb0uvv. Data information: All values are mean + SEM. Ordinary two-way ANOVA (D, E) and non-linear regression (E–H) were used. Source data are available online for this figure.

the predicted binding pose of CIM7 within RARα (Fig. EV4E), the extended phenyl group of CIM7.2 would be required to bind deeper in the narrow ligand pocket, preventing the formation of π-π interactions with RARα residues Phe228 and Phe302 (Fig. EV4F), as in the case of CIM7 (Fig. EV4A). These findings suggest that the CIM7 scaffold may be further refined to develop more potent CMA inhibitors. Overall, this data supports a distinctive mechanism of action by which CIM7 binds RARα and inhibits the interaction of NCoR1 to RARα, resulting in transcriptional changes in the CMA network genes and subsequent inhibition of CMA activity.

## CIM7 specifically targets cellular NCoR1/RARα binding and a subset of CMA substrates

Given the binding affinity of CIM7 for recombinant RARα, we predicted that its function as a CMA inhibitor would be entirely dependent on levels of RARα and NCoR1. In support of this prediction, the inhibitory effect of CIM7 on CMA activity was significantly diminished in cells with reduced levels of either RARα or NCoR1 (Fig. 4A). Furthermore, using proximity ligation assays (PLA), we observed a significant reduction of NCoR1 interaction with RARα in A549 cells treated with CIM7 (Fig. 4B; PLA assay validation and full field images in Appendix Fig. S5C), consistent with our in vitro findings that CIM7 disrupts the NCoR1/RARα axis. Additionally, using a biotin-conjugated CIM7 probe (biotin-CIM7), which we confirmed is capable of CMA inhibition similar to CIM7 and exhibits binding to recombinant RARα (Appendix Fig. S5D,E), we demonstrated direct binding of RARα to CIM7 in A549 cellular lysates (Fig. 4C). Notably, CIM7 does not bind the AHT RARα mutant that is incapable of binding NCoR1 (Fig. 4D), suggesting that CIM7 preferentially binds the inactive (NCoR1-bound) conformation of RARα. We additionally confirmed that CIM7 is specific to RARα and does not interact with the other RAR family members (Fig. EV5A).

Through RNA-seq analysis, we observed significant expression changes in 253 genes upon treatment with CIM7 which, to a large extent, resemble the changes that occur in cells expressing the AHT RARα mutant protein, which is unable to bind NCoR1 (Fig. 4E; Appendix Fig. S6A). Evaluation of differential gene expression in A549 cells treated with CIM7 compared to no treatment reveals enrichment in genes related to receptor ligand and signaling receptor activities (Fig. 4F). Genes changing significantly upon CIM7 treatment are linked to RARA and HOXD9 (Fig. 4G), although the latter is weakly expressed in our A549 cells in

comparison to RARα (Appendix Fig. S6B) and therefore an unlikely off-target. CIM7 treatment in cells expressing the AHT RARα mutant did not reproduce the transcriptional changes observed in the context of wild-type RARα, further supporting CIM7 targeting of the NCoR1/RARα interaction (Appendix Fig. S6A). However, we noticed a small subset of genes whose expression changed in AHT RARα-expressing cells upon CIM7 treatment (Appendix Fig. S6A). This suggests that CIM7 may have alternative targets and transcriptional effects when unable to bind RARα, such as the transcription factors NF-Y and FOXA1 (Fig. EV5B; Appendix Fig. S6C). We further evaluated potential off-target interactions of CIM7 against a panel of molecular targets (G-protein-coupled receptors, ionic channels, enzymes, transporters, and nuclear receptors) and observe in vitro binding to five targets (Appendix Fig. S6D,E), although these proteins have no or low expression in our A549 cells (Appendix Fig. S6F).

Finally, we evaluated the effect of CIM7 at the proteomic level by comparing changes in protein abundance in A549 cells upon exposure to CIM7 (Figs. 5A and EV5C show partial and full proteomes, respectively). Pathway enrichment revealed that CIM7 treatment preferentially impacted the proteome related to protein folding, chromatin, cytoplasmic translation, and the nucleosome (Fig. 5B,C). To identify CIM7-mediated changes due to inhibition of protein degradation by CMA, we focused on the fraction of the proteome undergoing degradation in lysosomes in A549 cells (proteins with increased levels upon inhibition of lysosomal proteolysis with N/L; control of N/L efficacy is shown in Appendix Fig. S7). We found that out of the 833 proteins degraded in lysosomes in A549 cells, CIM7 inhibits the degradation of 316 of them, often associated with an increase in their cellular levels (Figs. 5A and EV5D). We confirmed that the lysosomal degradation inhibited by CIM7 predominantly occurred via CMA, based on the failure of those proteins to be degraded in lysosomes in LAMP2A knockdown cells (Figs. 5D and EV5E,F; efficacy of L2AKD is shown in Appendix Fig. S7). As expected, L2AKD blocked the degradation of a larger number of CMA substrates than CIM7 (Fig. EV5F), since the latter preferentially affects NCoR1/RARα-regulated CMA, whereas LAMP2A KD blocks all CMA. Pathway enrichment analysis shows that CIM7 inhibition seems to spare CMA substrates involved in metabolism and protein translation, which were still efficiently degraded through CMA (Fig. EV5G). Examples of the inhibitory effect of CIM7 on the lysosomal degradation of CMA substrates are shown in Fig. 5E and Appendix Fig. S8A,B, whereas Appendix Fig. S8C

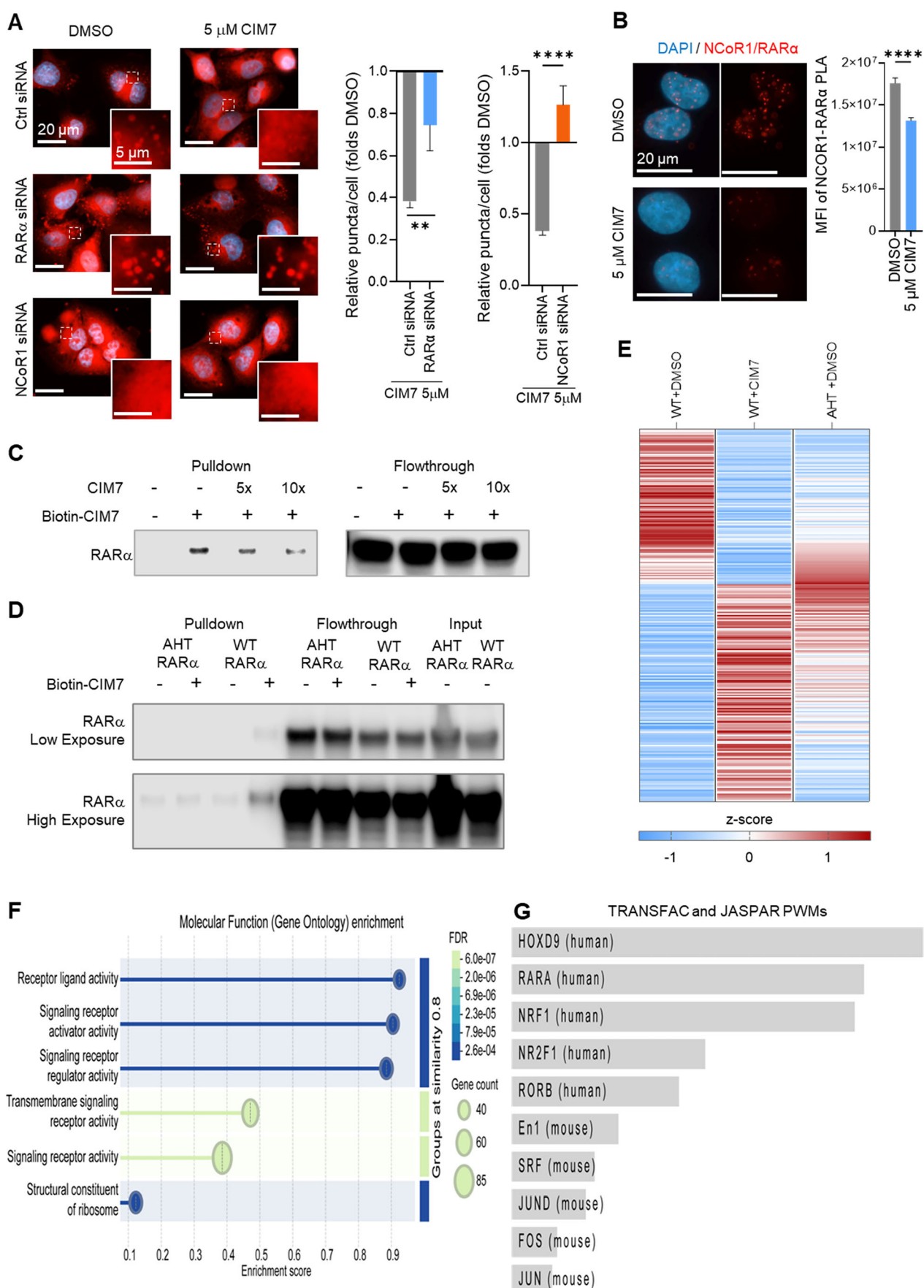

◀ **Figure 4. CIM7 selectively targets cellular NCoR1-bound RARα.**

(A) CMA activity in A549 cells expressing KFERQ-PS-Dendra with control (Ctrl) or targeted siRNA for RARα or NCoR1 after treatment with 5 µM of CIM7 for 24 h relative to untreated cells. Representative images (left) with KFERQ-PS-Dendra in red and nuclei highlighted with DAPI (inserts show higher magnification of the red channel) and quantification of fluorescent puncta per cell normalized to untreated cells in each siRNA background (right). $n \geq 115$ cells in three independent experiments. (**$P = 0.0023$, ****$P \leq 0.0001$). (B) Proximity ligation assay (PLA) of NCoR1 and RARα in A549 cells following 4-h treatment with DMSO or 5 µM CIM7. Representative images (left) and quantification of mean fluorescent intensity (MFI) of the NCoR1/RARα complex as visualized in red. Nuclei are highlighted with DAPI. $n \geq 95$ cells in two independent experiments. (****$P \leq 0.0001$). (C) Representative immunoblot for RARα of streptavidin pulldowns (left) and flowthrough (right) of A549 cellular lysate incubated without additions or with biotin-CIM7 (50 µM) and/or CIM7 (250 µM or 500 µM). $n = 3$ independent experiments. (D) Representative immunoblot for RARα of streptavidin pulldowns (left), flowthrough (center), and input (right) of cellular lysate from A549 cells expressing AHT or wild-type (WT) RARα incubated without additions or with biotin-CIM7 (50 µM). $n = 3$ independent experiments. (E) Heatmap of z-score for expression of genes changing significantly upon 3 h treatment with CIM7 in A549 cells expressing WT RARα and cells expressing AHT RARα. (F) Gene Ontology enrichment of all genes based on expression fold change of CIM7 treatment compared to DMSO, as predicted by STRING analysis. Bar coloring corresponds to false discovery rate (FDR). (G) TRANSFAC and JASPAR PWMs obtained through analysis with Enrichr sorted by rank-based ranking of genes significantly altered ( >1.5-fold or <0.5-fold change versus DMSO, $P < 0.05$ based on unpaired two-tailed $t$ test) upon CIM7 treatment in the presence of WT RARα in A549 cells. Data information: All values are mean + SEM. Unpaired two-tailed $t$ test (A, B) was used. Source data are available online for this figure.

shows the lack of inhibition by CIM7 on the degradation of non-CMA substrates (lysosomal degradation independent of LAMP2A).

## CIM7 affects lung cancer cellular viability and proliferation

As genetic inhibition of CMA in cancer cells resulted in reduced proliferation and cellular viability (Kon et al, 2011), we predicted that CIM7 would have a similar effect on NSCLC cells. Indeed, 72-h CIM7 treatment yields a dose-dependent decrease in viable cell population of all NSCLC cell lines assayed, with an $IC_{50}$ between 15 and 24 µM, depending on the cell line (Fig. 6A). Importantly, CIM7 shows a reduction of only ~35% in cellular viability of non-tumorigenic lung BEAS-2B cells at 50 µM, the highest concentration tested (Fig. 6A; $IC_{50}$ undetermined), supporting a therapeutic window for CIM7 treatment. To confirm that this effect on viability was a result of CMA inhibition, we utilized the previously described inactive analog, CIM7.2. Notably, CIM7.2 has minimal toxic effect in A549 cells when compared to CIM7 (Fig. 6B), suggesting that the cytotoxicity resulting from CIM7 treatment is likely due to its role as a CMA inhibitor. Daily treatment with CIM7 in A549 cells for five days to maintain constant CMA inhibition further enhanced CIM7 effect on toxicity, reducing the $IC_{50}$ to 7 µM, while daily treatments with ATRA and AM580 showed no dose-dependent effect on viability (Fig. 6C), suggesting that CIM7 is a more potent inhibitor of cellular viability, consistent with its higher potency in CMA inhibition compared to ATRA and AM580. Again, we observed a therapeutic window whereby 10 µM CIM7 results in a 75% reduction of cellular viability in A549 cells over the five-day treatment, while non-tumorigenic cell lines exhibit a reduction of only 30–40% viability under the same conditions (Fig. 6D). Additionally, CIM7 treatment significantly reduced cancer cell proliferation (Fig. 6E; $IC_{50} = 8.7$ µM), as evidenced by the lower number of colonies formed in the presence of CIM7 over one week. Together, these data suggest that the inhibitory effect of CIM7 on CMA through disruption of the NCoR1/RARα interaction could provide a selective small molecule strategy for targeting NSCLC growth.

## CIM7 treatment reduces CMA and NSCLC tumor growth in vivo

Next, we sought to evaluate the capacity of CIM7 to inhibit CMA and reduce NSCLC tumor growth in vivo as well as the

translational potential to inhibit CMA via NCoR1/RARα targeting in NSCLC. As a proof of concept, we evaluated the efficacy of CIM7 in vivo using a xenograft mouse model of A549 cells. We injected A549 cells subcutaneously into athymic nude mice (Crl:NU-Foxn1nu) and allowed tumors to grow to a volume of 150 mm³ at which point mice were divided into two groups. Mice were administered either vehicle or 25 mg/kg total body weight (b.w.) CIM7 daily for 30 days by intraperitoneal (i.p.) injection (Fig. 6F). Pharmacokinetic evaluation of CIM7 after 25 mg/kg IP dosing revealed a $C_{max}$ of 1382 ng/ml at 15 min (Appendix Fig. S9A), which is equivalent to ~3 µM CIM7, a dose at which CMA inhibition is maximal in cells. CIM7 also exhibits minimal binding to plasma protein (Appendix Fig. S9B). Despite the short half-life of CIM7 in plasma, measurement of tumor volume every three days revealed that mice treated with CIM7 had significantly reduced tumor growth over time compared to vehicle-treated mice (Fig. 6G). These findings mirror our previous observation of reduced tumor growth in A549 xenografts upon CMA inhibition through knock-down of LAMP2A (Kon et al, 2011).

Mice from both groups remained healthy over the course of treatment, with no change in weight or total blood counts (Fig. EV6A,B). Additionally, post-mortem evaluation of liver, heart, lung, and kidney revealed no toxicity based on H&E staining (Fig. EV6C, and pathology report in Appendix Note S1). We evaluated if the lack of toxicity in the animals could be due to differences in the CIM7 response between the human cancer cells and the mouse endogenous cells. Provided the highly conserved sequence of RARα between mice and humans (Appendix Fig. S9C), we predicted that CIM7 would interact with RARα in both the human A549 and mouse cells. However, we observed reduced interaction of biotin-CIM7 with RARα in mouse fibroblasts compared to A549 cells (Appendix Fig. S9D). Notably, biotin-CIM7 also does not interact with the human lung epithelial BEAS-2B cells (Appendix Fig. S9D), suggesting that CIM7 preferentially interacts with RARα in NSCLC cells over non-tumorigenic cells, consistent with its selective CMA inhibitory activity in NSCLC. In further support of this selectivity, isolated tumors from CIM7-treated mice revealed transcriptional downregulation of multiple CMA-related genes and reduced CMA score, compared to tumors from vehicle-treated mice (Fig. 6H), while only a minimal reduction was observed in mouse lung tissue and no reduction was observed in spleen or heart (Fig. EV6D). In agreement with our observations in vitro, TUNEL staining revealed a trend toward

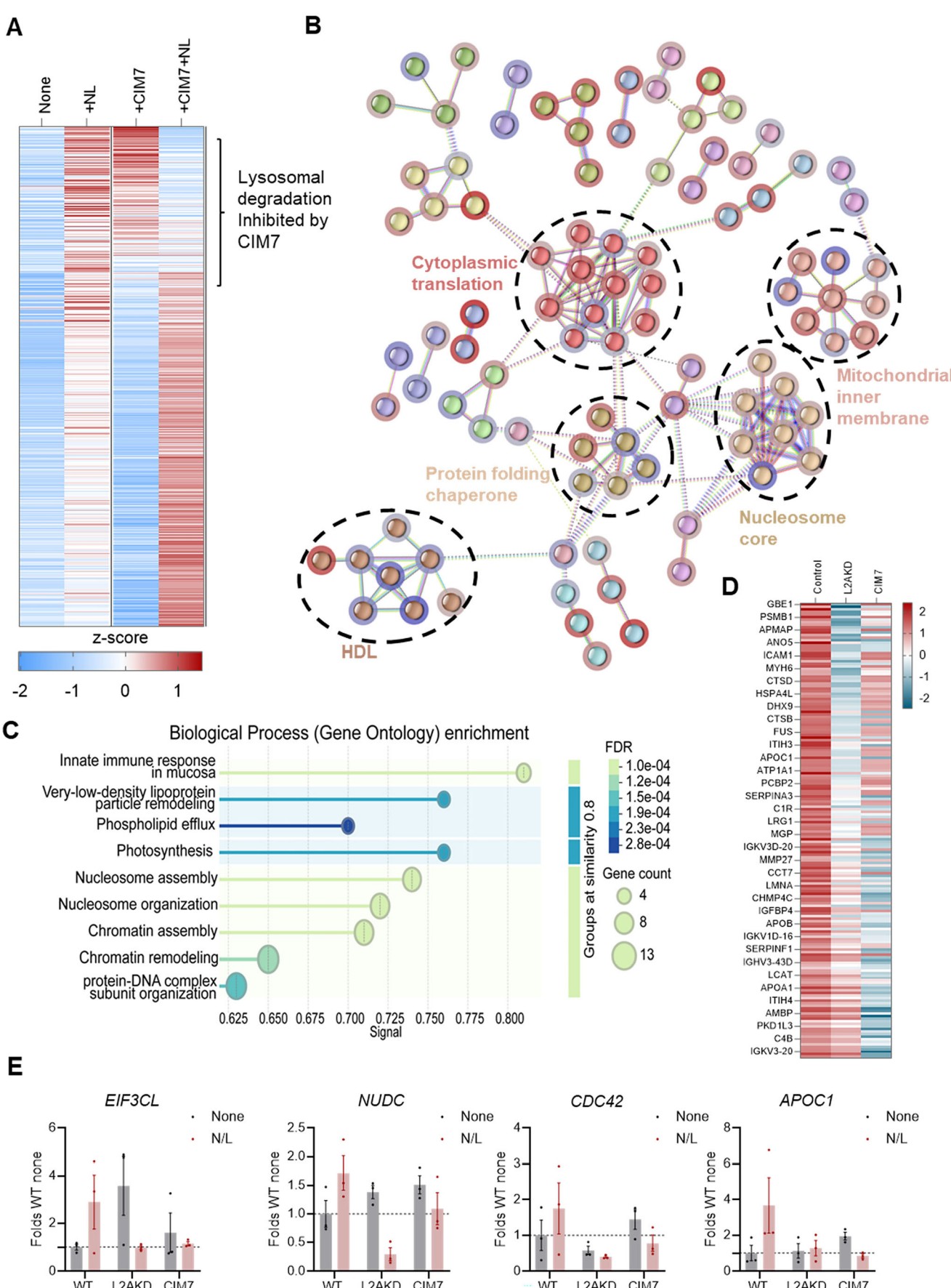

◄ **Figure 5.  CIM7 inhibits degradation of a subset of CMA substrates in NSCLC cells.**

(A) Heatmap (as z-score) of the proteome degraded in lysosomes in A549 cells (increased levels in presence of ammonium chloride and leupeptin (N/L)) and effect of 5 μM CIM7 treatment. (B) STRING analysis of proteins in A549 cells displaying significant (P < 0.05) changes in levels upon treatment with CIM7 compared to untreated A549 cells. The top five protein clusters are marked with dashed circles and labelled. (C) Gene Ontology enrichment of all proteins changing significantly ( >1.2-fold or <0.8-fold change versus DMSO, P < 0.05 based on unpaired two-tailed t-test) upon treatment with CIM7 in WT A549 cells as predicted by STRING analysis. Bar coloring corresponds to false discovery rate (FDR). (D) Heatmap (as z-score) of the proteome degraded in lysosomes in A549 cells (increased levels in presence of ammonium chloride and leupeptin (N/L) compared to N/L induced changes in LAMP2A knock-down (KD) or CIM7 treated A549 cells. (E) Representative examples of CMA substrates (no longer degraded in lysosomes in L2AKD A549 cells) whose degradation is also inhibited by CIM7. Values are mean + SEM and individual values. Data Information: All values are mean of three independent experiments. Source data are available online for this figure.

higher cell death in tumors from CIM7 treated mice (Fig. 6I). We noted that over 60% of CIM7-treated mice were highly responsive to CIM7 treatment, with virtually no tumor growth over the 30 days of treatment, while four mice were completely unresponsive to treatment (Fig. 6J; Appendix Fig. S10A). We did not observe obvious differences in gene expression and overall CMA score (Appendix Fig. S10B) between the responsive and unresponsive mice. We noted that tumors in CIM7-unresponsive mice were established significantly faster than those in the responsive mice (Appendix Fig. S10C), suggesting that CMA inhibition by CIM7 may not be sufficient to overcome more aggressive tumor growth. We additionally observed no difference in expression or protein levels of NCoR1, RARα, or RARβ in responsive versus nonresponsive mice, although RARγ was reduced only at the protein level in responsive mice (Appendix Fig. S11). Overall, these findings highlight that CMA inhibition via CIM7 treatment shows promise as an anti-cancer therapeutic strategy for NSCLC in vivo and future efforts to improve CIM7 pharmacokinetics may further improve this efficacy.

Finally, we analyzed changes in the regulatory complex in tumors by evaluating protein levels of NCoR1 and RARα in human tissue samples. In a panel of 39 NSCLC tumor tissues and 10 non-tumorigenic lung tissues from human patients, we found significantly higher levels of NCoR1 nuclear protein levels in NSCLC tumors (Fig. 7A), while nuclear protein levels of RARα were comparable between the two groups (Fig. 7B). The observed difference in protein levels may be a result of differences in protein stability between NSCLC and non-tumorigenic lung samples. As expected, the NCoR1/RARα ratio is significantly higher in NSCLC tumors than non-tumorigenic lung tissue (Fig. 7C). Additionally, analysis of gene expression in NSCLC cancer patients (Sanchez-Palencia et al, 2011) revealed a significant increase in both CMA score and NCoR1/RARα ratio in tumor tissue compared to non-tumorigenic lung tissue (Fig. EV7A,B). These findings are consistent with our analysis of human-derived cell lines (Figs. 1B and EV1C–E; Appendix Fig. S1B), suggesting that CMA is regulated by NCoR1/RARα in human NSCLC tumors and, therefore, disrupting the NCoR1/RARα interaction to inhibit CMA may have translational value in NSCLC tumors.

Previous studies have revealed consistent upregulation of CMA in cancers other than NSCLC (Arias and Cuervo, 2020; Kon et al, 2011). Additional evaluation of data generated by the TCGA Research Network (Cerami et al, 2012; de Bruijn et al, 2023; Gao et al, 2013; Liu et al, 2018) revealed an overall increase in the NCoR1/RARα ratio correlating to increasing CMA score between cancer types (Fig. 7D,E). The correlation between the NCoR1/RARα ratio and CMA score is additionally observed when

comparing individual patients in most cancers (Fig. EV7C). Finally, we observed a significant increase in both CMA score and the NCoR1/RARα ratio in patient samples with TP53 mutations compared to those expressing WT TP53, but not when comparing other mutation backgrounds. (Figs. 7F and EV7D–G), suggesting that the p53 oncogene may be a driver of these changes. Taken together, these data suggest that CIM7 and inhibition of the NCoR1/RARα interaction may have therapeutic potential beyond NSCLC, with broader pan-cancer applications.

## Discussion

In the last decade, the role of CMA on tumor growth and cancer cell survival has been investigated in many cancer cell types, highlighting a CMA-dependence that could be exploited as a cancer vulnerability and as a new area for development of anti-cancer therapeutics (Arias and Cuervo, 2020). Pharmacological inhibition of CMA allows for a more selective modulation of autophagy than in most previous studies and may help overcome the limitations of current strategies based on the use of non-specific modulators of autophagy that either affect other cellular pathways or completely disrupt all types of lysosomal degradation (Arias and Cuervo, 2020; Guo et al, 2022; White, 2012, 2015). While a growing number of studies support the role of CMA in regulating tumor growth and survival, to date there are no CMA-specific mechanisms or targets amenable for drug development for CMA inhibition. In this study, we have identified the NCoR1/RARα complex as a major transcriptional regulator of CMA activity in NSCLC and demonstrated its therapeutic value as a novel druggable target for pharmacological inhibition of CMA.

Our findings demonstrate upregulation of the NCoR1/RARα axis and elevated CMA activity in NCSLC tumor tissue and cell lines. We also showed that disrupting the specific interaction of NCoR1/RARα leads to selective inhibition of CMA, without contribution of macroautophagy, and that this inhibition of CMA in cancer cells can be attained without major changes to CMA in normal cells. It is noteworthy that CIM7's selective activity in cancer over normal cells provides a therapeutic window, which may be guided by specific binding to RARα in cancer cells and not in non-tumorigenic cells. Targeting the specific interaction of NCoR1 binding to the surface of RARα, outside of the canonical RAR ligand binding pocket, may help overcome the off-target effects and limited therapeutic index observed with RAR agonist treatment as a result of their binding to the ligand binding pocket and modulation of other nuclear receptors and their coregulators. Interestingly, bexarotene, a pan-RAR agonist like ATRA, in combination with

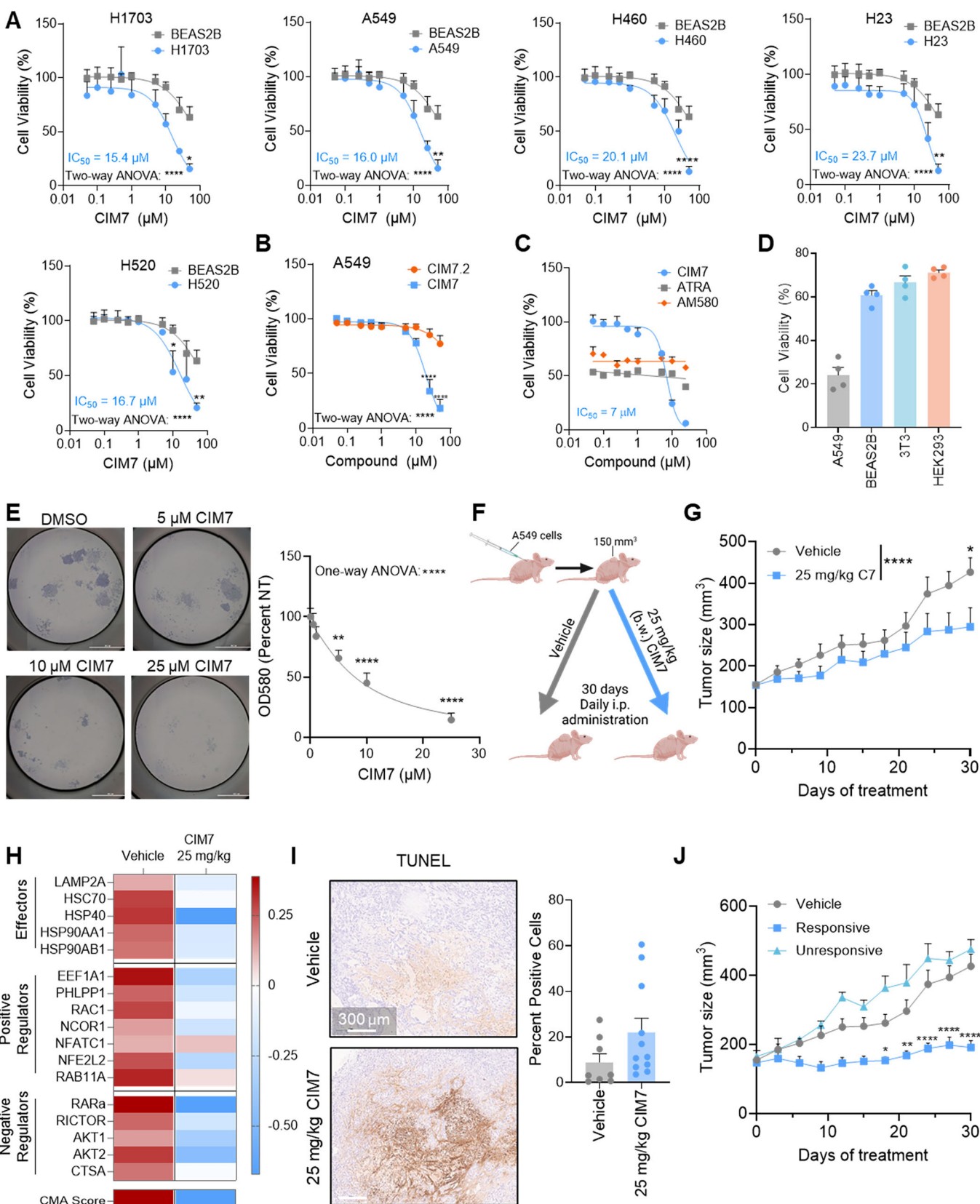

◀ **Figure 6. CIM7 affects NSCLC viability and inhibits human NSCLC tumor growth in vivo.**

(A) Cell viability of various NSCLC cell lines (blue circles), compared to BEAS-2B non-tumorigenic cells (grey squares) after exposure to increasing concentrations of CIM7 for 72 h. $n = 4$–6 independent experiments. (*$P = 0.0147$, **$P = 0.0073$, **$P = 0.0019$, *$P = 0.0264$, *$P = 0.0083$, ****$P \le 0.0001$). (B) Cell viability of A549 cells exposed to increasing concentrations of CIM7 or its dead derivative CIM7.2 for 72 h. $n = 3$ independent experiments. (****$P \le 0.0001$). (C) Cell viability of A549 cells exposed to increasing concentrations of CIM7, ATRA, or AM580 for five days, with treatment refreshed every 24 h. $n = 4$ technical replicates. (D) Cell viability of various cell lines exposed to 10 μM CIM7 for five days, with treatment refreshed every 24 h. $n = 4$ technical replicates. (E) Colony formation of A549 cells exposed to increasing concentrations of CIM7 over 1 week. Representative images (left) and quantification of crystal violet dye based on optical density at 580 nm (right). $n = 13$–19 wells from three independent experiments. (**$P = 0.0066$, ****$P \le 0.0001$). (F) Schematic of in vivo experimental design. Created in BioRender. https://BioRender.com/tcyxxdd. (G) Changes in tumor growth over time, measured every 3 days during the 30-day course of daily administration to mice of vehicle or 25 mg/kg of CIM7. $n = 9$–11 mice per treatment group. (*$P = 0.0248$, ****$P \le 0.0001$). (H) Heatmap of transcriptional changes in CMA-related genes and calculated CMA scores in tumors from vehicle- or 25 mg/kg CIM7-treated mice. $n = 9$–11 mice per treatment group. (I) TUNEL staining in representative tumors from vehicle- or 25 mg/kg CIM7-treated mice (left) and quantification of percentage of cells showing positive staining (right) in the same tumors as (H). (J) Changes in tumor growth over time, measured every 3 days during the 30-day course of daily administration to mice of vehicle or 25 mg/kg of CIM7 of the same mice in g-i with treated mice separated by responsive or unresponsive. $n = 4$–9 mice per group. (*$P = 0.0415$, **$P = 0.0049$, ****$P \le 0.0001$ responsive versus vehicle). Data information: All values are mean + SEM, with individual data points in bar graphs when $n < 10$. Ordinary two-way ANOVA (A, B, G, J) or one-way ANOVA (E) followed by Bonferroni's multiple comparisons post-hoc test and non-linear regression (A, E) were used. Source data are available online for this figure.

chemotherapy previously progressed to phase III trials in NCSLC, however the limited patient responses leave room for a more selective strategy with a higher therapeutic window (Ramlau et al, 2008). Unlike this pan-RAR agonist, CIM7 is selective to RARα. Furthermore, while previous studies have implicated NCoR1 in NCSLC tumor growth and survival, this effect can be mediated by its transcriptional repression and association, not only to RAR, but also to other nuclear receptors and specific transcription factors (Noblejas-López et al, 2018; Tan et al, 2019). Therefore, non-selective targeting of NCoR1 as a pharmacological strategy is likely to lead to pleiotropic effects, beyond disruption of NCoR1/RARα interaction and CMA inhibition.

In our study, we also showcase the rational discovery of the first selective CMA inhibitor, CIM7, using a structure-based screening strategy with complementary biophysical, biochemical, and cellular assays specific for disruption of NCoR1/RARα interaction that leads to dysregulation of the CMA transcriptional network in NCSLC cells. Our studies highlight an unprecedented mechanism for CMA inhibition, as CIM7 binding to RARα induces displacement of NCoR1, triggering of the RARα active conformation, and subsequent transcriptional effect. In contrast, our findings suggest that ATRA and agonist AM580 bind and stabilize the RARα active conformation, promoting recruitment of co-activators such as SRC (le Maire et al, 2010). ATRA and AM580 had no effect on the RARα conformation bound to NCoR1. The novel finding that CMA inhibition can be achieved through direct disruption of NCoR1 interaction with RARα and subsequent development of CIM7 builds upon our previous work that identified selective small molecule CMA activators through stabilization of NCoR1/RARα interaction and subsequent transcriptional upregulation of CMA genes (Bourdenx et al, 2021; Gomez-Sintes et al, 2022). These CMA activators have demonstrated CMA activation in normal cells and disease models with reduced CMA activity and significant efficacy in mouse models of neuronal and retinal neurodegeneration as well as in human hematopoietic cells from old donors (Bourdenx et al, 2021; Dong et al, 2021; Gomez-Sintes et al, 2022). It is noteworthy that either small molecule inhibition or activation of CMA through modulation of the NCoR1/RARα interaction can be accomplished without affecting the physiological modulatory effect of RARα on macroautophagy (Brigger et al, 2015; Zhong et al, 2015). We consider that CIM7's lack of effect on macroautophagy contributes to the therapeutic window of CIM7 and its favorable profile for

cancer treatment, since no toxicity was observed in blood or major tissues during in vivo studies. Furthermore, our proteomic analysis supports the notion that CIM7 selectively inhibits only the fraction of CMA regulated by NCoR1/RARα signaling, which, in the NCSLC model used in our study, accounts for ~37% of the total proteome degraded by CMA. A study published while our work was under revision (Zhou et al, 2025) highlights the potential risk of complete CMA blockage (using LAMP2A knockout instead of the knockdowns employed in other studies) in certain cancer stages, such as during mesenchymal cancer progression, when CMA prevents acquisition of pro-metastatic features.

CIM7 treatment is effective at inhibiting tumor growth in vivo, with a nearly complete cessation of tumor growth over the course of treatment in most mice. Provided the relative limited exposure of CIM7 in mouse plasma, future efforts should focus on improving upon the CIM7 scaffold for improved pharmacokinetic properties. Additionally, the effect of CIM7 on cellular viability is muted in comparison to the nanomolar efficacy by which this compound inhibits CMA in cells. This was in part expected given that inhibition of CMA through genetic knockdown of LAMP2A does not result in complete cell death (Kon et al, 2011) but rather reduces cancer cell proliferation. Future studies may also evaluate the value of CIM7 or an optimized derivative in combinatorial treatments with other cancer therapeutics, provided CMA's contribution to cancer drug resistance (Arias and Cuervo, 2020; Teixeira et al, 2024).

The discovery of the first selective CMA inhibitor represents a major milestone in our ability to pharmacologically induce selective CMA inhibition. This paves the way for evaluation and investigation of selective CMA inhibition not only in NCSLC but other cancer types, with potential applicability to additional tissues and disease models for which CMA inhibition is desirable, including autoimmune diseases like Lupus (Macri et al, 2015; Wang et al, 2020; Wilhelm et al, 2021). CIM7 is amenable to further optimization of its potency and pharmacological properties through structure-based drug design and medicinal chemistry to yield a clinical candidate CMA inhibitor. Recently, Polyphyllin D (PPD), a natural product with anti-tumor activity in NCSLC, was identified to have CMA inhibitory properties (Dong et al, 2025). However, PPD also affects macroautophagy (Dong et al, 2025; Liu et al, 2022) and the proposed potential targets are chaperone proteins involved in many other proteostasis events besides CMA.

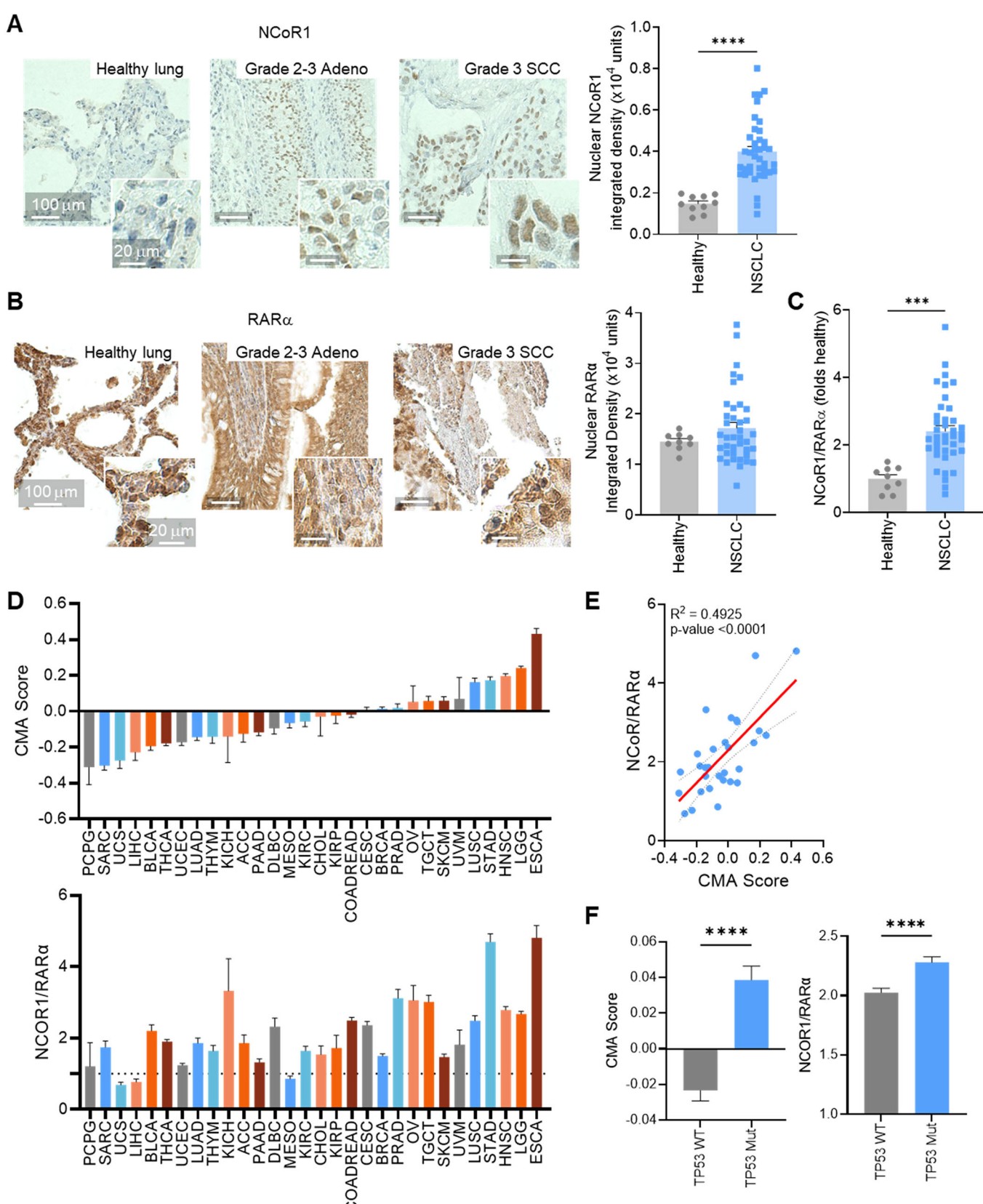

**Figure 7.  Changes in the CMA transcriptional network positively correlate with NCoR1/RARα axis expression in human samples.**

(A) Representative images (left) and quantification (right) of immunohistochemistry (IHC) for NCoR1 in a panel of 10 healthy lung and 39 NSCLC tumor samples. (****$P \leq 0.0001$). (B) Representative images (left) and quantification (right) from the same tissues as in (A) of IHC for RARα. ($P = 0.2784$). (C) Ratio of NCoR1 to RARα nuclear intensities from the IHC of tissues in (A, B), $n = 9$ healthy lung and 39 NSCLC tumor samples total. (***$P = 0.0002$). (D) CMA score (top) and NCoR1/RARα ratio (bottom) across 30 cancer types based on data from the TCGA. CMA score is calculated as a comparison between the cancers. (E) Correlation between NCoR1/RARα ratio and CMA score in the 30 cancer types from (D). (F) CMA score (left) from (D) and NCoR1/RARα ratio (right) from (E) in patient tumors with wild-type (WT) or mutant (Mut) TP53 based on data from the TCGA. $n > 1500$ individual tumors. (****$P \leq 0.0001$). Data information: All bar values are mean + SEM and dots represent individual samples, except in (E) where dots represent average expression in each cancer type. Unpaired two-tailed $t$ test (A–C, F) and simple linear regression (E) were used. Source data are available online for this figure.

In contrast, CIM7 is selective for CMA and shows no effect on macroautophagy, making it, to our knowledge, the first CMA inhibitor completely selective for CMA activity.

In summary, our study has identified targeting the NCoR1/RARα interaction as a feasible and effective way to selectively downregulate CMA in NSCLC and developed a first-in-class CMA inhibitory small molecule with a unique mechanism of action and in vivo therapeutic efficacy in NSCLC tumor xenografts. Considering the upregulation of CMA established in almost all solid cancer types assessed to-date (Arias and Cuervo, 2020; Dong et al, 2025; Ichikawa et al, 2020; Kon et al, 2011) and our observed correlation of increased NCoR1/RARα with increasing CMA score across a diverse panel of cancer types, we predict that our small molecule targeting strategy inhibiting NCoR1/RARα interaction with CIM7 or an optimized derivative may have translational value in a variety of cancers, as illustrated through the course of our study in NSCLC.

# Methods

### Reagents and tools table

| Reagent/resource | Reference or source | Identifier or catalog number |
|---|---|---|
| **Experimental models** | | |
| BL21-CodonPlus (DE3)-RIPL Competent Cells (*E. coli*) | Agilent | Cat# 230280 |
| One Shot™ TOP10 Chemically Competent (*E. coli*) | Thermo Fisher Scientific | Cat# C404003 |
| NCI-H520 (*H. sapiens*) | ATCC | Cat# HTB-182 |
| NCI-H23 (*H. sapiens*) | ATCC | Cat# CRL-5800 |
| NCI-H460 (*H. sapiens*) | ATCC | Cat# HTB-177 |
| A549 (*H. sapiens*) | ATCC | Cat# CCL-185 |
| NCI-H1703 (*H. sapiens*) | ATCC | Cat# CRL-5889 |
| BEAS-2B (*H. sapiens*) | ATCC | Cat# CRL-9609 |
| IMR-90 (*H. sapiens*) | ATCC | Cat# CCL-186 |
| HEK-293 (*H. sapiens*) | ATCC | Cat# CRL-1573 |

| Reagent/resource | Reference or source | Identifier or catalog number |
|---|---|---|
| SHSY-5Y (*H. sapiens*) | ATCC | Cat# CRL-2266 |
| NIH-3T3 (*M. musculus*) | ATCC | Cat# CRL-1658 |
| Female NU/NU Nude Crl:NU-Foxn1$^{nu}$ (*M. musculus*) | Charles River | Strain Code 088 |
| Female C57BL/6 J mice (*M. musculus*) | Charles River | Strain Code 027 |
| Lung non-small cell carcinoma NSCLC tissue array with normal lung tissue | US Biomax | Cat# LC10011a |
| **Recombinant DNA** | | |
| pcDNA Flag-RARα | Jonathan Kurie & Harish Srinivas (Addgene plasmid # 35555; http://n2t.net/addgene:35555; RRID:Addgene_35555) | Addgene #35555 |
| mCherry-GFP-LC3 | Dr. Fernando Macian | N/A |
| KFERQ-PS-Dendra | Laboratory stock Koga et al, 2011 (Nature Communications) | N/A |
| LentiCRISPR v2 | Feng Zhang (Addgene plasmid # 52961; http://n2t.net/addgene:52961; RRID:Addgene_52961) | Addgene #52961 |
| **Antibodies** | | |
| Mouse anti-dendra2 | OriGene | Cat# TA180094 |
| Rabbit anti-LAMP2A | Abcam | Cat# ab18528 |
| Rabbit anti-LC3 | Novus Biologicals | Cat# nb100-233 |
| Rabbit anti-NcoR1 | Cell Signaling | Cat# 5948 |
| Rabbit anti-RARα | Cell Signaling | Cat# 62294 |
| Mouse anti-RARα | Sigma-Aldrich | Cat# WH0005914M1 |
| Mouse anti-RARβ | Sigma-Aldrich | Cat# WH0005915M1 |
| Mouse anti-RARγ | ThermoFisher | Cat# TA810381 |
| Goat anti-Rabbit HRP | KPL | Cat# 5067205 |
| Goat anti-Mouse HRP | KPL | Cat# 5067202 |

| Reagent/resource | Reference or source | Identifier or catalog number |
|---|---|---|
| Goat anti-mouse Alexa Fluor 555 | Thermo Fisher Scientific | Cat# A32727 |
| Goat anti-Rabbit Alexa Fluor 488 | Thermo Fisher Scientific | Cat# A-11008 |
| Anti-rabbit IgG, HRP-linked Antibody #7074 | Cell Signaling | Cat# 7074 |
| **Oligonucleotides and other sequence-based reagents** | | |
| List of qPCR primers | This study | Appendix Table S3 |
| AHT mutant forward primer | Fisher Scientific Custom DNA Oligonucleotides | 5'-ggtgcgcaaagggggccaggaaacct-3' |
| AHT mutant reverse primer | Fisher Scientific Custom DNA Oligonucleotides | 5'-ttctcaatgagctccccc-3' |
| RARα siRNA | Origene | Cat# SR303989A |
| NCoR1 siRNA | Origene | Cat# SR423484A |
| LAMP2A sgRNA forward primer | Fisher Scientific Custom DNA Oligonucleotides | 5'-caccgcaacttccttgtgcccatag-3' |
| LAMP2A sgRNA reverse primer | Fisher Scientific Custom DNA Oligonucleotides | 5'-aaacctatgggcacaaggaagttgc-3' |
| **Chemicals, enzymes and other reagents** | | |
| FITC-NCoR1 | GenScript | FITC-Ahx-RLITLADHICQIITQDFAR |
| FITC-SRC | GenScript | FITC-Ahx-RHKILHRLLQEGS |
| Recombinant RARα LBD | This study | N/A |
| ChemDiv screening compounds | This study | Appendix Table S2 |
| Biotin-CIM7 | This study | N/A |
| All-trans retinoic acid | Sigma-Aldrich | Cat# R2625 |
| AM580 | Millipore Sigma | Cat# A8843 |
| Ammonium chloride | Sigma-Aldrich | Cat# A9434 |
| Leupeptin | Thermo Fisher | Cat# BP2662 |
| Hoechst | Life Technologies | Cat# 33342 |
| Paraformaldehyde solution 4% in PBS | Santa Cruz | Cat# sc-281692 |
| Dimethyl sulfoxide (DMSO) | Sigma-Aldrich | Cat# D2650 |
| Ethanol | Supelco | Cat# EX0276 |
| Polyethylene glycol 400 (PEG-400) | Nextal Biotechnologies | Cat# 133086 |
| Tween-80 | Sigma-Aldrich | Cat# P1754 |
| RPMI 1640 | Thermo Fisher | Cat# 11875119 |
| DMEM high glucose | Sigma | Cat# D5648 |
| DMEM/F-12 | Thermo Fisher | Cat# 11320033 |
| Fetal Bovine Serum (FBS) | Fisher | Cat# 35010CV |

| Reagent/resource | Reference or source | Identifier or catalog number |
|---|---|---|
| Neonatal Calf Serum (NCS) | Hyclone | Cat# SH30401 |
| Antibiotic-Antimycotic | Gibco | Cat# 15240096 |
| Myco-sniff™ mycoplasma PCR detection kit | MP Biomedical | Cat# 093050201 |
| Lipofectamine™ 2000 | Invitrogen | Cat# 11668027 |
| Rneasy® Plus Mini Kit | Qiagen | Cat# 74134 |
| QuantiTect Reverse Transcription Kit | Qiagen | Cat# 205314 |
| CellTiter-Glo® Luminescent Cell Viability Assay | Promega | Cat# G7570 |
| PureLink™ HiPure Plasmid Midiprep Kit | Invitrogen | Cat# K210004 |
| Q5® Site-Directed Mutagenesis Kit | New England Biolabs® | Cat# E0554S |
| ImmPRESS® HRP Horse Anti-Goat Polymer Detection Kit, Peroxidase | Vector Laboratories | Cat# MP-7405 |
| ImmPRESS® HRP Goat Anti-Rat IgG, Mouse adsorbed Polymer Detection Kit, Peroxidase | Vector Laboratories | Cat# MP-7444 |
| ImmPRESS® HRP Goat Anti-Rabbit IgG Polymer Detection Kit, Peroxidase | Vector Laboratories | Cat# MP-7451 |
| ImmPRESS® HRP Goat Anti-Mouse IgG Polymer Detection Kit, Peroxidase | Vector Laboratories | Cat# MP-7452 |
| Duolink In Situ PLA Probe Anti-Rabbit | Sigma-Aldrich | Cat# DUO92002 |
| Duolink In Situ PLA Probe Anti-Mouse | Sigma-Aldrich | Cat# DUO92004 |
| DeadEnd™ Colorimetric TUNEL System | Promega | Cat# G7130 |
| Western Lightning™ Plus ECL | Revvity | Cat# NEL103001 |
| SuperSignal™ West Femto Maximum Sensitivity Substrate | Thermo Fisher | Cat# 34095 |
| SafetyScreen44 Panel – FR | Eurofins | Cat#PP241 |

| Reagent/resource | Reference or source | Identifier or catalog number |
|---|---|---|
| Protein binding (plasma, mouse, CD-1) – US | Eurofins | Cat#2223 |
| **Software** | | |
| ImageJ v1.53r | https://imagej.net/ Schneider et al, 2012 (Nature Methods) | |
| GraphPad Prism v9 | https://www.graphpad.com/ | |
| Genevestigator® | https://genevestigator.com/ | |
| Schrödinger Software Suite Releases 2017-2023 | https://www.schrodinger.com/maestro | |
| PyMOL | http://www.pymol.org/ | |
| Microsoft Excel 365 Version 2305 | www.microsoft.com | |
| Columbus Image Data Storage and Analysis | www.perkinelmer.com | |
| QuPath | https://qupath.github.io/ Bankhead et al, 2017 (Scientific Reports) | |
| MicroCal PEAQ-ITC analysis software | Malvern Panalytical | |
| GeneSys | Syngene | |
| Image Studio 6.0 | LI-COR | |
| Leica Application Suite X | Leica Microsystems | |
| Biorender | https://www.biorender.com/ | |
| Proteome Discoverer v2.5 | ThermoFisher | |
| cBioPortal | https://www.cbioportal.org/ | |
| **Other** | | |
| Axiolab 5 fluorescent microscope | Zeiss | |
| Axiovert 200 fluorescence microscope | Zeiss | |
| G-BOX Chemi XX6 | SynGene | |
| Odyssey Fc Imager | LI-COR | |
| QuantStudio™ 5 Real-Time 384-well PCR system | Applied Biosystems™ | |
| Operetta High Content Imagine System | Perkin Elmer | |
| Infinite F200 PRO | Tecan | |
| MicroCal PEAQ-ITC | Malvern Panalytical | |

| Reagent/resource | Reference or source | Identifier or catalog number |
|---|---|---|
| Spark | Tecan | |
| Cytation 5 Cell Imaging Multimode Reader | BioTek | |
| Genesis™ | Oxford Science | |
| Human lung tumor and nontumor microarray | Sanchez-Palencia et al, 2011 (International Journal of Cancer) | |

## Cell culture and knockdowns

All cell lines were obtained from the American Type Culture Collection (ATCC) and validated by Short Tandem Repeat (STR) profiling. Cell lines were checked regularly for mycoplasma contamination using DNA staining protocol with Hoechst (Life Technologies, 33342) dye and any suspected contamination was further tested using Myco-sniff™ mycoplasma PCR detection kit (MP Biomedical, 093050201). All cells were maintained in a 37 °C incubator with 5% $CO_2$. Human cell lines H520, H23, H460, A549, H1703 and BEAS-2B were maintained in RPMI (Thermo Fisher, 11875119) with 10% FBS (Fisher, 35010CV) and 1% Antibiotic-Antimycotic (Gibco™, 15240096), IMR90 and HEK-293 cells were maintained in DMEM (Sigma, D5648) with 10% FBS, and 1% Antibiotic-Antimycotic, SHSY-5Y cells were maintained in DMEM/F-12 (Thermo Fisher, 11320033) with 10% FBS, and 1% Antibiotic-Antimycotic, and NIH-3T3 cells were maintained in DMEM (Sigma, D5648) with 10% NCS (Hyclone, SH30401) and 1% Antibiotic-Antimycotic. All cellular experiments were performed with these media unless otherwise noted. Knockdowns for RARα and NCoR1 were done by transfecting targeted siRNA (SR303989A and SR423484A, respectively) from Origene using Lipofectamine™ 2000 (Invitrogen, 11668027) and efficiency of knockdown was tested by immunoblot and quantitative RT-PCR.

## Small molecules

All compounds utilized in screening for CMA inhibitors were provided by ChemDiv; catalog numbers and molecular information are detailed in Appendix Table S2. CIM7 (G621-0375) was additionally synthesized by ChemDiv. The synthesis and analytical characterization of CIM7 and biotin-CIM7 (>98% purity) is provided in Appendix Methods and Appendix Figs. S12 and S13, respectively. Screening compounds provided by ChemDiv were >90% pure with purity confirmed by [1]H NMR and/or LC/MS. For cellular and in vitro assays, compounds were reconstituted in 100% DMSO to prepare 20 mM, 10 mM, and 5 mM stock solutions and diluted in cell culture medium or aqueous buffers for assays. All-trans retinoic acid (Sigma-Aldrich, R2625) was reconstituted in 100% DMSO to prepare 10 mM stock solutions and diluted in cell culture medium lacking FBS for cellular assays.

## CMA score from published transcriptomics

RNA-seq data of various lung cell lines was acquired from Genevestigator database (Hruz et al, 2008). RNA expression of

CMA network genes and oncogene mutation status of cancers from the TCGA PanCancer Atlas were acquired from CBioPortal (Cerami et al, 2012; de Bruijn et al, 2023; Gao et al, 2013; Liu et al, 2018). Data sets of RNA-seq in response to cellular treatment or from patients post-treatment were excluded from analysis. CMA score was calculated from these data sets as previously described (Bourdenx et al, 2021; Gomez-Sintes et al, 2022). Briefly, each component within the CMA network (Bourdenx et al, 2021) was assigned a specific weight. As LAMP2A is limiting in CMA, it was designated a weight of 2. All other components were assigned a weight of 1. Subsequently, each component was assigned a directional score of either $+1$ or $-1$, based on the established impact of that component on CMA activity. The cumulative score was then computed as the weighted and directed average of expression counts for each component within the CMA network.

## Immunohistochemistry of human lung samples

Lung tissue array (US Biomax, LC10011a) slides were deparaffinized and rehydrated with Histoclear (National Diagnostics, HS200) and a series of ethanol grades. Endogenous peroxidase activity was blocked with 0.5% hydrogen peroxide in methanol and antigen retrieval was achieved using Antigen Unmasking Solution (Vector Laboratories, H-3300). Slides were immunostained using ImmPRESS® HRP IgG Polymer Detection Kits (Vector Laboratories, MP-7405, MP-7444, MP-7451, or MP-7442). Briefly, slides were blocked for 1 h in ImmPRESS® ready-to-use (2.5%) serum, incubated in primary antibody in blocking serum at 4 °C overnight, washed and then incubated in ImmPRESS® IgG for 1 h. Slides were then stained with ImmPACT® DAB solution (Vector Laboratories, SK-4105) and counter stained with hematoxylin (Vector Laboratories, H-3401) before dehydration and mounting using Vecta-Mount® (Vector Laboratories, H-5000). Slides were imaged using an Axiolab 5 (Zeiss) fluorescent microscope. Individual nuclei, based on DAB staining and cellular morphology, were circled to define individual regions of interest (ROIs) and raw integrated density was quantified for each ROI using ImageJ Software (NIH (Schneider et al, 2012)). The primary antibodies were used as follows: mouse anti-RARα (1:250, Sigma-Aldrich, WH0005914M1) and rabbit anti-NCoR1 (1:1000, Cell Signaling, 5948).

## Immunofluorescence

Cells were plated on 12-well glass coverslips and fixed in 2% paraformaldehyde (PFA) in PBS the next day. Immunostainings were performed after 1 h blocking (10% goat serum, 2% Triton-X-100 in PBS) and subsequent incubation with primary and anti-species appropriate Alexa Fluor™ secondary antibody. Nuclei were stained for 15 min using Hoechst (Life Technologies, 33342) prior to mounting. Coverslips were imaged with an Axiovert 200 fluorescence microscope (Zeiss) using a 63x/1.4 oil objective lens and Rhodamine, FITC, and DAPI filter sets. Exposure times were kept consistent across all cell lines. Images were acquired with a high-resolution CCD camera. Nuclei were circled for individual ROIs and raw integrated density was quantified for each ROI using ImageJ software (NIH (Schneider et al, 2012)). The primary antibodies were used as follows: mouse anti-RARα (1:250, Sigma-Aldrich, WH0005914M1), rabbit anti-NCoR1 (1:1000, Cell Signaling, 5948), rabbit anti-LAMP2A (1:1000, Abcam, ab125068).

## Proximity ligation assay (PLA)

A549 cells were plated on 12-well glass coverslips and treated with 5 μM BMS493, 5 μM CIM7, or equal volume DMSO the next day. After 4 h, cells were fixed with 4% PFA in PBS and then permeabilized with 0.5% Triton-X for 1 h. The following steps followed the instruction manual of Duolink In Situ PLA Probe Anti-Rabbit (DUO92002-30RXN) and Anti-Mouse (DUO92004-30RXN). Antibodies used were Rabbit anti-NCoR1 (1:1000, Cell Signaling, 5948) and Mouse anti-RARα (1:250, Sigma-Aldrich, WH0005914M1). Images were acquired by fluorescence microscope (Zeiss) as above and quantification was done by counting number of PLA particles and MFI of NCoR1/RARα PLA.

## Immunoblot

Cells were lysed in RIPA buffer (150 mM NaCl, 1% NP-40, 0.5% NaDoc, 0.1% SDS, 50 mM Tris, pH 8) containing protease inhibitors (100 μM leupeptin, 100 μM AEBSF, 10 μM pepstatin, 10 μM EDTA, pH 8) and phosphatase inhibitor cocktail set II (Millipore Sigma, 524625) for 15 min on ice and then centrifuged at $16{,}000{\times}g$ for 15 min. Supernatant was isolated and protein was quantified using the Lowry method with bovine serum albumin as standard (Lowry et al, 1951). Proteins were resolved on an 8% bis-tris acrylamide gel and transferred to Amersham Protran 0.45 NC nitrocellulose membranes (GE Healthcare). Membranes were blocked for 1 h in 5% non-fat milk in TBS-Tween 20 (0.05% (v/v)) and probed with primary and secondary antibodies, sequentially. Antigen signals were detected using Western Lightning™ Plus ECL (Revvity, NEL103001) and visualized using a G-BOX Chemi XX6 (SynGene) system or SuperSignal™ West Femto Maximum Sensitivity Substrate (Thermo, 34095) and visualized with an Odyssey Fc imager (LI-COR). Densitometric analysis was performed using ImageJ software (NIH (Schneider et al, 2012)). Primary antibodies were as follows: Western blot: rabbit anti-RARα (1:500, Cell Signaling, 62294), rabbit anti-NCoR1 (1:1000, Cell Signaling, 5948), rabbit anti-LAMP2A (1:1000, Abcam, ab18528), rabbit anti-LC3 (1:1000, Novus Biologicals, nb100-2331), mouse anti-dendra2 (1:1000, Origene, TA180094), rabbit anti-RARβ (1:500, Sigma-Aldrich, WH0005915M1), RARγ (1:500, Thermo-Fisher, TA810381).

## mRNA quantitative PCR

RNA was extracted from cells using RNeasy® Plus Mini Kit (Qiagen, 74134). RNA was isolated from tumors and spleen tissue by addition of 500 ml QIAzol™ Lysis Reagent (Qiagen, 79306) and stainless-steel beads (Next Advance, SSB14B) to each sample followed by homogenization in a Bullet Blender® 24 Storm (Next Advance, BBY24M) at 4 °C. Hearts and lungs were pulverized with liquid nitrogen before addition of QIAzol™ Lysis Reagent. Once tissues were fully homogenized and suspended in QIAzol™ Lysis Reagent, 100 ml chloroform (LabChem, LC130402) were added to each sample, tubes were inverted to mix, and then centrifuged for 15 min at $12{,}000{\times}g$ at 4 °C. The colorless, upper phase (~350 ml) was removed, mixed with 350 ml 70% ethanol, and sample processing was completed as with cells beginning at step 4 of the RNeasy® Plus Mini Kit protocol. Reverse transcription was performed using QuantiTect Reverse Transcription Kit (Qiagen, 205314). Quantitative RT-PCR was performed using Power SYBR Green PCR Master Mix (Applied Biosystems™, 4368708) on a

QuantStudio™ 5 Real-Time 384-well PCR system (Applied Biosystems™). The sequence of all probes used for qPCR is included in Appendix Table S3. CMA index quantification was performed as previously described (Bourdenx et al, 2021).

## Autophagic measurements

### Macroautophagy activity

Macroautophagy activity was measured by two different procedures. Fluorescence-based assay of macroautophagy flux: Cells transduced with lentivirus carrying mCherry-GFP-LC3 (Kimura et al, 2007) were plated on glass coverslips in a 12-well plate, treated the next day with indicated compounds or equal volume DMSO. All treatments were done in complete media, except for treatment with ATRA, which was done in serum-free media to prevent binding of ATRA to serum and ensure cellular availability. After 24 h, cells were fixed for 15 min with 2% PFA in PBS (Santa Cruz, sc-281692). Coverslips were mounted to slides with DAPI Fluoromount-G (SouthernBiotech, 0100-20) and imaged as described for immunofluorescent staining. The number of fluorescent puncta per cell was quantified using ImageJ software (NIH (Schneider et al, 2012)).

Immunoblot assay of macroautophagy flux: Cells were plated in 6-well plates then treated with CIM7 or DMSO for 20 h before addition of 20 mM ammonium chloride (Sigma-Aldrich, A9434) and 100 μM leupeptin (Thermo Fisher, BP2662) for an additional 4 h. Protein was extracted and immunoblotted for LC3. Autophagic flux was calculated after densitometric quantification as the increase in the amount of LC3-II upon lysosomal proteolysis inhibition relative to samples in which lysosomal proteolysis was not inhibited.

### CMA activity

Cells were transduced with lentivirus carrying the KFERQ-PS-Dendra reporter as previously described (Koga et al, 2011). For experiments with NIH-3T3 cells: cells were photoactivated by 3-minute exposure to a 3.5 mA light emitting diode (LED: Norlux, 405 nm) and then plated in glass-bottom 96-well plates. Cells were treated and fixed, as described above, with the addition of Hoechst dye at 1:5000 for nuclear staining, solution was removed, and PBS was added to all wells, then imaged, and analyzed using high content microscopy (Operetta system, Perkin Elmer). For experiments with human lung cell lines: cells were plated on glass coverslips in a 12-well plate before photoactivation. Cells were treated as described above with the different compounds 2 h after photoswitching. In experiments involving treatment following knockdown of RARα or NCoR1 or expression of WT or AHT RARα, KFERQ-PS-Dendra expressing cells were transfected with siRNA or plasmid and 16 h later photoswitched and treated with the desired compounds in new media 2 h after photoswitching. At experimental stop point, coverslips were fixed, imaged, and quantified as described above. Changes in degradation of substrates by CMA were confirmed by monitoring their accumulation (by immunofluorescence or immunoblot against Dendra protein) upon inhibiting lysosomal proteolysis as described above.

## In silico small molecule screening

The structures of the RARα homodimer in complex with the small molecule reverse agonist BMS493 and NCoR1 peptide (PDB ID:

3KMZ) and in complex with the small molecule antagonist BMS614 and SRC peptide (PDB ID: 3KMR) were refined to remove one monomer and the NCoR1 or SRC peptide. The protein structures were prepared using Protein Preparation Wizard (Schrödinger, LLC). Compound structures from a library of 1,303,560 ChemDiv compounds were converted into three-dimensional all atom structures using LigPrep and Epik (Schrödinger, LLC), sampling different ionization states at pH 7.0 ± 2.0, stereochemistry and tautomeric forms resulting in a library of total of 1,768,565 screening compounds. In both structures, the ligand binding pocket was defined by a 20 Å box around three residues triangulating the peptide binding site: isoleucine 396, cystine 265, and valine 240. Compounds were docked in ligand flexible mode using Glide docking (Schrödinger, LLC) through a series of increasingly stringent docking modes (high-throughput (HTVS), standard precision (SP), and extra precision (XP)), with the top 10% of compounds carried to the next phase. Following the final Glide XP docking, the top 1304 in silico hits from each screen were further filtered to include only compounds with a HERG score > −5, Qplog(o/w) between −2–6.5, and QPPCaco ≥350. Compounds were analyzed for their molecular interactions and chemical structures in Maestro (Schrödinger, LLC) for final hit selection. From the NCoR1 screen, 5 compounds were selected based on distance from BMS493, 30 based on proximity to leucine 261, and 20 based on proximity to leucine 398. From the SRC screen, 5 compounds were selected based on the best GScore and the remaining compounds were sorted into 45 similarity clusters and the top scoring molecule (based on GScore) was selected from each cluster. Of the 105 selected compounds, 94 were available commercially and selected for further screening. Calculated properties and predicted ADME properties for CIM7 were generated by QikProp (Schrödinger, LLC). Using the PubChem database, it was confirmed that CIM7 has not been reported as a hit in previous screens.

## Molecular docking

The structure of the RARα homodimer in complex with the small molecule reverse agonist BMS493 and NCoR1 peptide (PDB ID: 3KMZ) was used for docking. The reverse agonist structure and one monomer were removed before the protein was prepared using Protein Preparation Wizard (Schrödinger, LLC). NCoR1 peptide was either removed or remained present for the different dockings. The ligand binding pocket was defined as above. CIM7 and CIM7 analogs were converted to 3D all atom structures using Ligprep (Schrodinger, LLC) and assigned partial charges with Epik (Schrodinger, LLC). Docking was performed in flexible ligand and flexible receptor mode using induced fit docking (IFD) (Schrödinger, LLC). Structures were analyzed using Maestro and Pymol (Schrödinger, LLC).

## Molecular dynamics simulations

Molecular dynamic simulations were performed using DESMOND (DESMOND, version 3, Schrodinger, LLC, 2020-21). Using Desmond System Preparation, the system containing the top scoring compound pose in complex with RARα from the induced fit docking was prepared with TIP3P solvent and orthorhombic box shape with volume minimized. The system was neutralized with $Na^+$ ions and 0.15 M $Na^+$ $Cl^-$ was added to the system. The prepared system was loaded into Desmond: Molecular Dynamics. Simulation runs were performed using OPLS4 force field, 300 K,

and the constant pressure of 1.0325 bar. The Nose–Hoover Chain thermostat and Martyna–Tobias–Klein barostat were used to maintain the temperature and pressure, respectively. Simulation time was set to a total of 2000 ns, trajectory of 250 ps, and 8000 frames. The model was relaxed before simulation. A different seed was utilized for three individual simulations. Simulations were analyzed with Desmond: Simulation Interactions Diagram (Schrödinger, LLC). Interatomic distances and root mean square deviation (RMSD) data obtained from Molecular Dynamic analysis were plotted using GraphPad Prism 9. Structures were analyzed using Maestro and Pymol (Schrödinger, LLC).

## Expression and purification of RARα LBD

The ligand binding domain (LBD) of human RARα (residues 176-462) was fused to a preceding his-tag and expressed in *Escherichia coli* BL21(DE3) cell strain. Cells were grown at 37 °C in LB medium containing 50 µg/ml kanamycin (Sigma-Aldrich, K1377) until $OD^{600}$ around 0.8. Isopropyl-b-D-thiogalactoside (IPTG) (Gold Biotechnology, I2481C) was added at a final concentration of 0.8 mM and cells were incubated at 20 °C overnight to induce T7 polymerase expression. Cell cultures were harvested by centrifugation at 8000×*g* for 15 min and pellets were resuspended in lysis buffer (20 mM Tris-HCl pH 8, 500 mM NaCl, 25 mM imidazole) supplemented with one complete™, EDTA-free protease inhibitor tablet (Roche, 11873580001) before lysis using a high-pressure homogenizer. Lysate was centrifuged at 35,000×*g* at 4 °C for 45 min. The supernatant was filtered and loaded onto a 5 ml affinity column prepared using HisPur™ Ni-NTA resin (Thermo Scientific™, 88222) and preequilibrated with lysis buffer. After three washes with lysis buffer, proteins were eluted with lysis buffer supplemented with 200 mM imidazole (Sigma-Aldrich, I5513). Eluted protein was buffer exchanged in FPLC buffer (10 mM HEPES, 150 mM NaCl, pH 8) and concentrated to 500 µl using an Amicon® Ultra-15 10 K centrifugal filter unit (Millipore Sigma, UFC901024) before being loaded onto a Superdex 200 Increase 10/300 GL gel filtration column (Fisher Scientific, 45002570) for final purification. Purified RARα fractions were pooled and supplemented with 5 mM dithiothreitol (Thermo Scientific™, R0862) and maintained on ice.

## Isothermal titration calorimetry

ITC measurements were performed at 25 °C on a MicroCal PEAQ-ITC (Malvern Panalytical). The buffer solution for assays used was 25 mM sodium phosphate pH 8.0, 150 mM sodium chloride, 1 mM TCEP. In a typical experiment, 295 µM RARα was injected (75 µl syringe) into 20 µM compound (200 µl sample cell) for 15 injections. The delay between injections was 150 s. ITC titration curves were analyzed using MicroCal PEAQ-ITC analysis software (Malvern Panalytical).

## Fluorescence polarization binding assays

Fluorescence polarization assays (FPA) were performed as previously described (Gomez-Sintes et al, 2022). Briefly, RARα LBD in 20 mM HEPES, 150 mM NaCl, 5 mM DTT, pH 7.5 was serially diluted starting at 50 µM and incubated with DMSO or 25 µM CIM7 before addition of 5 nM FITC-NCoR1 peptide. The fluorescein-tagged peptide of NCoR1, FITC-Ahx-

RLITLADHICQIITQDFAR (FITC-NCoR1) was synthesized and purified by Genscript and provided at a purity >95%, verified by HPLC and mass spectrometry. The peptide was reconstituted in 100% DMSO at a concentration of 10 mM, then diluted in aqueous buffers. Fluorescence polarization was measured at 1 h on a F200 PRO microplate reader (TECAN) with an excitation wavelength of 470 nm and emission wavelength of 530 nm. $EC_{50}$ values were calculated using GraphPad Prism software by non-linear regression analysis.

## Time-resolved fluorescence energy transfer binding assay

For time-resolved fluorescence energy transfer (TR-FRET) assays, 3 nM His-tagged RARα LBD in 1× PBS, 0.02% Pluronic F-127 (Invitrogen, P3000MP), 5 mM DTT, pH 7.4 was incubated with the donor, 0.5 nM HTRF Mab Anti-6HIS Tb cryptate Gold (Cisbio-Perkin Elmer, 61HI2TLA) and the acceptor, 50 nM FITC-SRC in 384-well flat bottom low volume plates. The fluorescein-tagged SRC peptide, FITC-Ahx-RHKILHRLLQEGS (FITC-SRC), was synthesized and purified by Genscript and provided at a purity >95%, verified by HPLC and mass spectrometry. The peptide was reconstituted in 100% DMSO at a concentration of 10 mM, then diluted in aqueous buffers. Compounds were dispensed into each well and DMSO matched. After a 1 h incubation, TR-FRET signal was measured on a TECAN Spark Instrument with an excitation at 320 nm and emission wavelengths at 495 nm and 520 nm. The 520/495 emission ratios (TR-FRET ratio) were plotted.

## Generation of the AHT RARα mutant

pcDNA Flag-RARα (Addgene, 35555) (Srinivas et al, 2005) was purified from the provided agar stab using a PureLink™ HiPure Plasmid Midiprep Kit (Invitrogen, K210004). The AHT RARα mutant was constructed by site-directed mutagenesis (New England Biolabs®, E0554S) of the pcDNA Flag-RARα DNA using the following primers: 5′-ggtgcgcaaaggggggccaggaaacct-3′ and 5′-ttctcaatgagctcccccc-3′. This converted $Ala^{194}$ and $His^{195}$ both into Gly. The mutated plasmid was transformed in One Shot™ TOP10 Chemically Competent E. Coli (Thermo Fisher, C404003) and purified using a PureLink™ HiPure Plasmid Midiprep Kit. Plasmids were verified by Sanger Sequencing (Azenta Life Sciences).

## Pulldown of biotin-CIM7 for endogenous proteins

At each stage, beads were separated from supernatant using a DynaMag™-2 magnet (Invitrogen, 12321D). In total, 50 µl Dynabeads™ M-280 Streptavidin (Invitrogen, 11205D) were prepared by washing twice with 1x PBS. Beads were incubated with 50 µM biotin-CIM7 or equal volume DMSO for 1–2 h at RT with constant rocking. Beads were then washed three times with 1× PBS at RT before incubation with 500 µl cellular protein lysate in RIPA buffer with protease and phosphatase inhibitors (Thermo Scientific, 78446), at a concentration of 1.5–1.8 mg/ml with or without addition of 250 or 500 µM CIM7. Beads and lysate were incubated overnight at 4 °C with continuous rotation. The next morning, the supernatant (flowthrough) was removed and stored at 4 °C. Beads were washed four times with 1× PBS at RT before incubation in 40 µl 1× sample buffer (10% glycerol, 2% w/v SDS, 1.25%

β-mercaptoethanol (Gibco™, 21985023), 0.1% w/v bromophenol blue (Polysciences Inc., 24995), 0.12 M Tris) in PBS at 96 °C for 15 min. The supernatant (pulldown) was removed, and beads were stored at −20 °C. Protein lysate not subjected to the beads (input), flowthrough, and pulldown samples were then subjected to SDS-PAGE and immunoblot.

## Evaluation of biotin-CIM7 in A549 cells expressing WT or AHT RARα

For the investigation of CIM7 interaction with AHT RARα, A549 cells were transfected with WT or AHT RARα mutant cDNA using Lipofectamine™ 2000. Cells were lysed using RIPA buffer with protease and phosphatase inhibitors (Thermo Scientific, 78446) 16 h later and pulldown was performed as described above with cellular lysate at a concentration of 1 mg/ml. For the investigation of CIM7 effect on gene expression, A549 cells were transfected with WT or AHT RARα mutant cDNA. Cells were treated 16 h later with DMSO or 5 μM CIM7. After 3 h, RNA was extracted using RNeasy Plus Mini Kit (Qiagen) and non-directional with PolyA mRNA sequencing was performed through Novogene (detailed in supplemental methods). Data analysis: Statistical regulation was assessed using heteroscedastic $T$ test (if $P$ value <0.05). Data distribution was assumed to be normal, but this was not formally tested. Changes in protein levels were thresholded as those above 1.5 or 0.5-fold relative to wild-type or DMSO. Pathway enrichment analysis was performed using Enrichr (https://maayanlab.cloud/Enrichr/) (Chen et al, 2013; Kuleshov et al, 2016; Xie et al, 2021).

## Generation of LAMP2A knockdown cells

CRISPR knockdown of human LAMP2A was achieved by inserting sgRNA against human LAMP2A into the lentiCRISPR v2 plasmid (Addgene, 52961) (Sanjana et al, 2014), with site-directed mutagenesis as described above. The following primers were used: 5'-caccgcaacttccttgtgcccatag-3' and 5'-aaacctatgggcacaaggaagttgc-3'. Lentivirus was introduced to A549 cells and knockdown was confirmed by immunoblot.

## Lysosomal degradation in CMA-deficient cells with CIM7

A549 and A549 LAMP2A KD cells were treated with DMSO or 5 μM CIM7 for 24 h with the addition of lysosomal protease inhibitors, NH$_4$Cl (20 mM) and Leupeptin (100 μM) for 12 h. Cells were pelleted and lysed in a buffer containing 5% SDS, 5 mM DTT and 50 mM ammonium bicarbonate (pH = 8) for 1 h at RT for disulfide bond reduction. Samples were alkylated with 20 mM iodoacetamide in the dark for 30 min. Phosphoric acid was then added to the sample at a final concentration of 1.2%. Samples were diluted with six volumes of binding buffer (90% methanol and 10 mM ammonium bicarbonate, pH 8.0), gently mixed and loaded on to a S-trap filter (Protifi) and spun at 500 rcf for 30 s. After washing the samples twice with binding buffer, sequencing grade trypsin (1 μg, Promega) diluted in 50 mM ammonium bicarbonate was added into the S-trap filter and samples were digested at 37 °C for 18 h. Peptides were eluted in three steps: (i) 40 μl of 50 mM ammonium bicarbonate, (ii) 40 μl of 0.1% TFA and (iii) 40 μl of 60% acetonitrile and 0.1% TFA. The peptide solution was pooled, spun at 1000 rcf for 30 s and dried in a vacuum centrifuge.

### Sample desalting

Prior to mass spectrometry analysis, samples were desalted using a 96-well plate filter (Orochem) packed with 1 mg of Oasis HLB C-18 resin (Waters). Briefly, the samples were resuspended in 100 μl of 0.1% TFA and loaded onto the HLB resin, which was previously equilibrated using 100 μl of the same buffer. After washing with 100 μl of 0.1% TFA, the samples were eluted with a buffer containing 70 μl of 60% acetonitrile and 0.1% TFA and then dried in a vacuum centrifuge.

### LC-MS/MS acquisition and analysis

Samples were resuspended in 10 μl of 0.1% TFA and loaded onto a Dionex RSLC Ultimate 300 (ThermoFisher), coupled online with an Orbitrap Fusion Lumos (ThermoFisher). Chromatographic separation was performed with a two-column system, consisting of a C-18 trap cartridge (300 μm ID, 5 mm length) and a picofrit analytical column (75 μm ID, 25 cm length) packed in-house with reversed-phase Repro-Sil Pur C18-AQ 3 μm resin. Peptides were separated using a 90 min gradient from 4 to 30% buffer B (buffer A: 0.1% formic acid, buffer B: 80% acetonitrile + 0.1% formic acid) at a flow rate of 300 nl/min. The mass spectrometer was set to acquire spectra in a data-dependent acquisition (DDA) mode. Briefly, the full MS scan was set to 300–1200 $m/z$ in the orbitrap with a resolution of 120,000 (at 200 $m/z$) and an AGC target of $5 \times 10^5$. MS/MS was performed in the ion trap using the top speed mode (2 s), an AGC target of $1 \times 10^4$ and an HCD collision energy of 35. Proteome raw files were searched using Proteome Discoverer software (ThermoFisher, v2.5) with the SEQUEST search engine and the SwissProt mouse database (updated April 2023). The search for total proteome included variable modification of N-terminal acetylation, and fixed modification of carbamidomethyl cysteine. Trypsin was specified as the digestive enzyme with up to two missed cleavages allowed. Mass tolerance was set to 10 pm for precursor ions and 0.2 Da for product ions. Peptide and protein false discovery rate was set to 1%.

### Data analysis

Following the search, data was processed as described (Aguilan et al, 2020). Briefly, proteins were log2 transformed, normalized by the average value of each sample and missing values were imputed using a normal distribution 2 standard deviations lower than the mean. Statistical regulation was assessed using heteroscedastic T-test (if $P$ value <0.05). Data distribution was assumed to be normal, but this was not formally tested. Changes in protein levels were thresholded as those above 1.5 or 0.8-fold relative to wild-type and degradation of protein as a change of 1.2-fold in lysosome inhibitor treated conditions relative to no lysosomal inhibitor. Pathway enrichment analysis was performed using the STRING database (https://string-db.org/) (Szklarczyk et al, 2023) and Enrichr (https://maayanlab.cloud/Enrichr/) (Chen et al, 2013; Kuleshov et al, 2016; Xie et al, 2021).

## Cellular viability assay

Cells were seeded on 96-well white-bottom plates overnight then treated with increasing concentration of CIM7 (DMSO matched) for 72 h before viability analysis was performed with the CellTiter-Glo® kit (Promega, G7570), as per manufacturer's protocol. For 5 day treatment experiments, cells were seeded on 384-well white-bottom

plates overnight then treated with increasing concentration of compound (DMSO matched). Media was changed and compound was added again every 24 h for a total of five treatments. On day six (24 h after the final treatment), viability analysis was performed with the CellTiter-Glo® kit. Luminescence was read using a F200 PRO microplate reader (TECAN) with a 500-millisecond integration time.

## Colony formation assay

A549 cells were plated in a clear, flat-bottom, 96-well plate at a density of 10 cells/well overnight. Cells were treated with increasing concentration of CIM7 (DMSO matched), with complete media removal and treatment refresh every 72 h. One week after cells had initially been plated, cells were fixed with 4% PFA in PBS (Santa Cruz, sc-281692) for 15 min, then incubated with crystal violet solution (Sigma-Aldrich, V5265) for 10 min, washed with tap water until the dye came off, and left to dry at room temperature. To measure crystal violet intensity, cells were incubated in 100% methanol for 10 min, then $OD_{580}$ was measured using a F300 PRO microplate reader (TECAN). Methanol was removed and cells were again left to dry at room temperature before imaging with a Cytation 5 Cell Imaging Multimode Reader (BioTek).

## Pharmacokinetic analysis

Three 7–9 week old C57BL/6 J female mice (strain code 027) were obtained from Charles River. Mice were fed and provided water ad libitum before the study. Animals were administered 2.5 mg/ml CIM7 in 10% ethanol, 40% PEG-400, 5% Tween-80, and 45% water by intraperitoneal (i.p.) injection to achieve a final dosage of 25 mg/kg total body weight (b.w.). Mice were monitored for 24 h following administration and plasma was collected at 0.08, 0.25. 0.5, 1, 2, 4, 8, and 24 h. Levels of CIM7 in plasma were analyzed using LC-MS/MS. Pharmacokinetic parameters were calculated using Phoenix WinNonlin 8.3.5.

## Evaluation of CIM7 binding to off-target molecular targets

SafetyScreen44 was performed by Eurofins with 10 µM CIM7 in DMSO. Briefly, 10 mM stock in DMSO was diluted to 10 µM before use in a series of enzymatic and binding biochemical assays. Two replicates were performed for each assay and positive control compounds were included.

## Evaluation of CIM7 binding to mouse plasma protein

Assessment of CIM7 binding to protein in mouse plasma was performed by Eurofins with 10 µM CIM7 in DMSO. Briefly, 10 mM stock in DMSO was diluted to 10 µM CIM7 in DMSO before addition to plasma and serum proteins in a microplate format in duplicate. Levels of CIM7 and control compounds were detected by HPLC-MS/MS to determine the bound and unbound fractions.

## Tumor xenografts studies

Three-month-old female nu/nu homozygous Nude mice (strain code 088) were obtained from Charles River. Animals were housed in ventilated cages in a barrier-controlled facility (19–23 °C 30–60% relative humidity; 12-h light/dark cycle) with ad libitum access to standard chow pellets and water. Only females were used in this study to be able to compare the efficacy of our intervention with other published studies (Kellar et al, 2015) and due to the enhanced engraftment of human cells observed in female immunodeficient mice (Sara et al, 2020). Animals were assigned randomly to the vehicle and drug groups. One mouse was removed mid-treatment due to infection and one vehicle-treated mouse was removed on treatment day 30 due to abdominal cystitis. All handling and treatments in this study were done according to protocol and all animal studies were under an animal study protocol approved by the Institutional Animal Care and Use Committee of Albert Einstein College of Medicine. To establish A549 cell xenografts, mice were injected subcutaneously in the upper right flank with $10 \times 10^6$ cells in saline solution. Tumor length and width was measured every 3 days and volume ($mm^3$) was calculated as $(length \times width^2)/2$. Once tumors reached a volume of ~150 $mm^3$, mice were randomly distributed into vehicle or treated groups. CIM7 was reconstituted daily in 10% ethanol (Supelco, EX0276), 40% polyethylene glycol 400 (PEG-400) (Nextal Biotechnologies, 133086), 5% Tween-80 (Sigma-Aldrich, P1754), and 45% water. Mice were treated daily by i.p. injection of freshly prepared 25 mg/kg total b.w. CIM7 in vehicle (as in pharmacokinetic analysis) or vehicle-alone for 30 consecutive days. On treatment day 30, mice were euthanized by perfusion under anesthetic (Covetrus, 11695067772) and tissues were dissected. In total, 9 mice were treated with vehicle and 11 were treated with 25 mg/kg b.w. CIM7.

## Immunohistochemistry of A549 xenograft samples

Tumors were dissected from nu/nu xenograft mice, fixed in 4% paraformaldehyde (PFA) in PBS (Santa Cruz, sc-281692) for 24 h, then paraffin-embedded and sectioned into four mm-thickness sections. IHC of dissected tumors were performed as above. Apoptotic cell death was visualized in tissue sections using TUNEL staining (Promega, G7130) and the percent of positive cells in full sections was evaluated using QuPath Positive Cell Detection.

## Histopathology of peripheral organs and blood cell counts

Livers, lungs, kidneys, spleens, and hearts from CIM7- or vehicle-treated animals were fixed in 4% PFA in PBS (Santa Cruz, sc-281692) for 24 h after dissection. Tissues were paraffin-embedded, sectioned, stained with hematoxylin and eosin (H&E), and analyzed by a blinded expert pathologist to score for possible presence of toxicity in these organs. Blood cell counts were performed by submandibular blood collection on days 0, 14, and 28 of treatment using a Genesis™ (Oxford Science).

## Statistical analysis and sample size determination

All numerical results are reported as mean + SEM and represent data from a minimum of three independent experiments unless otherwise stated. Experiments in cells in culture were performed on different days to confirm reproducibility of the procedures. When outliers were determined, we used the ROUT method (Q = 1%). Statistical significance was performed by: unpaired two-tailed t-test

## The paper explained

### Problem

Lung cancer is the leading cause of cancer-related death worldwide, with most diagnoses falling into the subcategory of non-small cell lung cancer (NSCLC). Consistent upregulation of chaperone-mediated autophagy (CMA) in NSCLC supports tumor growth, making CMA a promising target for NSCLC treatment. However, current pharmacologic strategies to inhibit CMA also impact macroautophagy, which can in turn allow for cancer cell survival. Therefore, a need exists for a specific and potent small molecule CMA inhibitor.

### Results

In this study, we demonstrated that the druggable NCoR1/RARα axis is a promising target for selective inhibition of CMA in NSCLC. Furthermore, we developed CIM7, a first-in-class small molecule inhibitor of CMA which functions through disruption of this protein-protein interaction and prevents the degradation of a subset of proteins by CMA. Treatment with CIM7 in vivo effectively reduces tumor growth in a mouse model of NSCLC.

### Impact

Our study identifies the first selective CMA inhibitor to date by disrupting NCoR1/RARα interaction. This small molecule may be further refined for use as a potent anti-cancer therapeutic. Furthermore, CMA is consistently upregulated in a variety of cancer types, suggesting that a selective CMA inhibitor such as CIM7 may be utilized for a pan-cancer strategy.

of the means (in instances of single comparisons), one-way analysis of variance (ANOVA) followed by the Bonferroni post-hoc test (in instances of multiple comparisons with one independent variable), or two-way analysis of variance (ANOVA) followed by post-hoc test (in instances of multiple comparisons with two independent variables). Statistical analysis was performed for all assays and significant differences, as determined by significance level $P \le 0.05$, are noted in graphical representations. The number of animals used for in vivo experiments was calculated by power analysis based on previous results. No mouse was excluded unless they were determined to be in poor health. The following software programs were utilized: Microsoft Excel 365 (Version 2305) for basic data handling, Prism software (v9 - GraphPad Software Inc) for data analysis, ImageJ software (v1.53r, NIH(Schneider et al, 2012)) for image analysis and quantification, GeneSys (Syngene) for acquisition of gel scans, Schrödinger Software Suite (Releases 2017-2023 – Schrödinger, LLC) for structural and small molecule data analysis, and Leica Application Suite X (LAS X) for acquisition of microscopy images. Schematics were created with BioRender.com.

## Data availability

Proteomic data has been deposited to ProteomeXchange (PRIDE) with the dataset accession number PXD061323. Transcriptomic data has been deposited to the gene expression omnibus (GEO) with the dataset accession number GSE291058. Source data are available with this paper.

The source data of this paper are collected in the following database record: biostudies:S-SCDT-10_1038-S44321-025-00254-y.

## Peer review information

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

## Acknowledgements

We thank all members of the Cuervo and Gavathiotis laboratories for their guidance, especially Dr. Francisco Marques for his guidance in xenograft establishment and Ranee Harrison and Sandra Pelka for their guidance on tumor immune cell infiltration. We thank Dr. Kira Gritsman for use of the Genesis™ machine for complete blood cell counts and Dr. Britta Will for use of the Cytation 5 machine for imaging colony formation assays. We additionally thank Dr. Elizabeth Neyens from the Einstein Histology and Comparative Pathology Facility for serving as an expert pathologist, the Einstein Macromolecular Therapeutics Development Facility for use of the Operetta high-content microscope, and the Einstein Animal Housing and Studies Facility for animal housing and husbandry. Studies were supported by National Institutes of Health grants P01AG031782 (AMC and EG), P30AG038072 (AMC and EG), the Rainwaters Foundation (AMC and EG), the Freedom Together foundation (AMC), Hevolution foundation (AMC and EG), the Irma T Hirschl Trust Career Award (EG) and the generosity of Robert and Renee Belfer (AMC). NIH grant T32AG023475 provided support to MM and RRK; NIH grant T32GM007491 and NIH F31AG084192 provided support to RS; and IRACDA program grant K12GM102779 to RRK. Partial support of various facilities was provided by grants R01CA223243, R01CA178394, P30CA013330 and S10OD01630 grant for the NMR resources.

## Author contributions

**Mericka McCabe**: Investigation; Writing—original draft; Writing—review and editing. **Rajanya Bhattacharyya**: Investigation. **Rebecca Sereda**: Investigation. **Olaya Santiago-Fernández**: Investigation. **Rabia R Khawaja**: Investigation. **Antonio Diaz**: Investigation. **Kristen Lindenau**: Investigation. **Deniz Gulfem Ozturk**: Investigation. **Thomas P Garner**: Investigation. **Simone Sidoli**: Data curation; Investigation. **Ana Maria Cuervo**: Conceptualization; Supervision; Funding acquisition; Investigation; Writing—original draft; Writing—review and editing. **Evripidis Gavathiotis**: Conceptualization; Supervision; Funding acquisition; Investigation; Writing—original draft; Writing—review and editing.

Source data underlying figure panels in this paper may have individual authorship assigned. Where available, figure panel/source data authorship is listed in the following database record: biostudies:S-SCDT-10_1038-S44321-025-00254-y.

## Disclosure and competing interests statement

EG has received compensation for consulting or serving on scientific advisory boards, and has equity ownership from BaxGen Therapeutics, BeanPod Biosciences, Life Biosciences, Stelexis Biosciences; and has consulted for Boehringer Ingelheim and Guidepoint. AMC consults for Generian Pharmaceutics and Cognition Therapeutics and has equity ownership from Life Biosciences. All EG and AMC consulting activities are outside of the submitted work. The remaining authors declare no competing interests.

# Expanded View Figures

**Figure EV1.  Profiling of CMA in NSCLC.**  ▶

(**A**) Table of our five NSCLC cells lines, their CMA activity, and status of oncogenes. (**B**) Representative immunoblot for LAMP2A (left) and quantification of LAMP2A levels (right) in the indicated NSCLC cell lines and non-tumorigenic control BEAS-2B cells normalized to levels of BEAS-2B cells in each experiment (right). $n = 5$–6 independent experiments. (****$P \leq 0.0001$). (**C–F**) Transcriptional analysis of CMA-related genes (**C**), calculated CMA z-score (**D**), expression (as z-score) of CMA effectors, positive regulators, and negative regulators (**E**), and NCoR1/RARα ratio (**F**), in 17 non-tumorigenic lung cell lines and 15 NSCLC-derived cell lines. The heat map in (**C**) shows differential gene expression. Cell lines are listed in Appendix Table S1. Data from (Hruz et al, 2008). (**D**): **$P = 0.0031$; (**E**): ***$P = 0.0001$, ****$P \leq 0.0001$, *$P = 0.0403$, ***$P = 0.0004$, ***$P = 0.0005$, *$P = 0.0108$, **$P = 0.0012$; (**F**): **$P = 0.0029$). Data information: All values are mean + SEM or individual data points to represent individual samples. One-way ANOVA (**B**), unpaired two-tailed $t$ test (**D, F**), or multiple unpaired $t$ tests with individual variances and Bonferroni post-hoc analysis (**E**) were performed. Source data are available online for this figure.

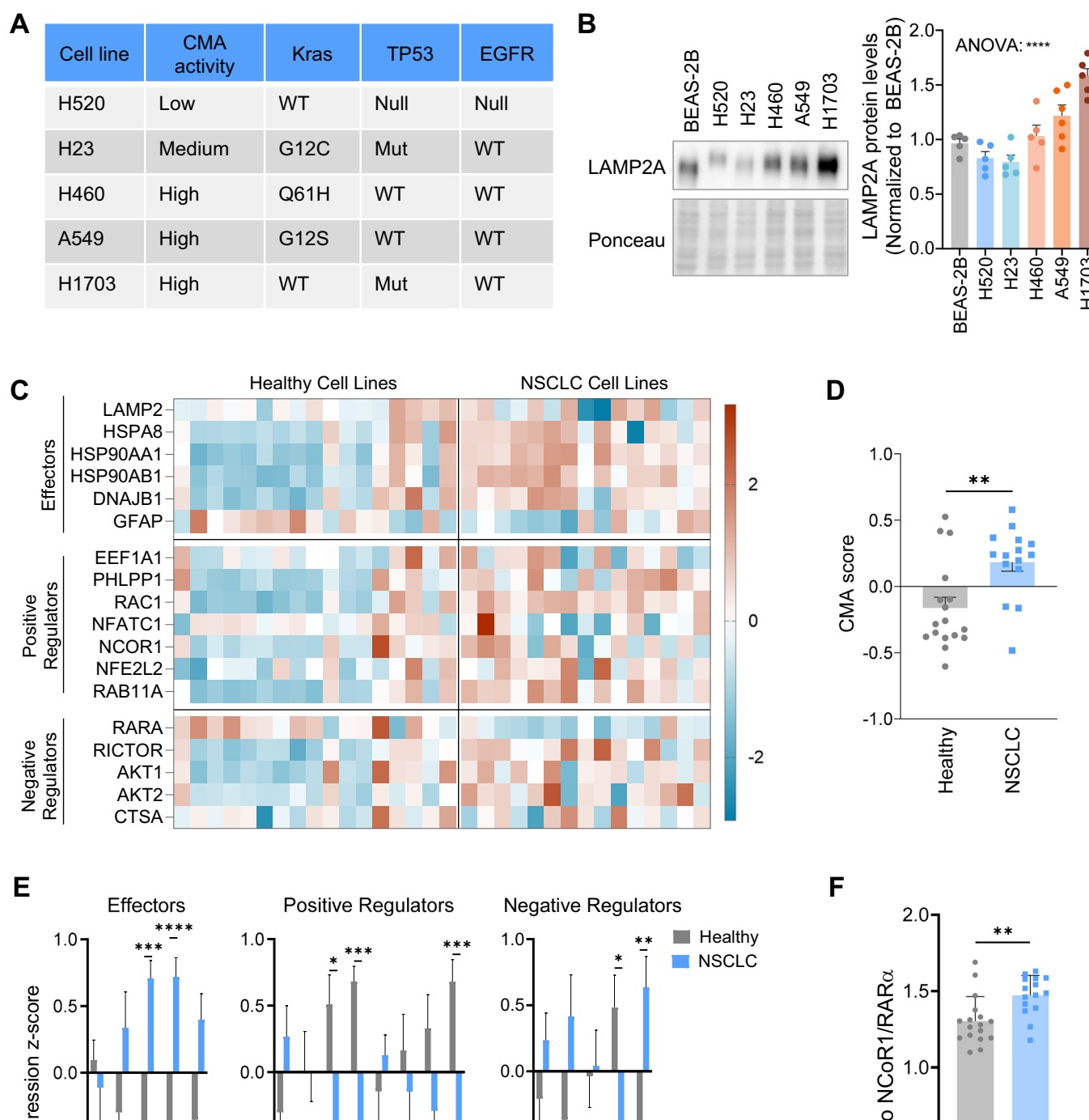

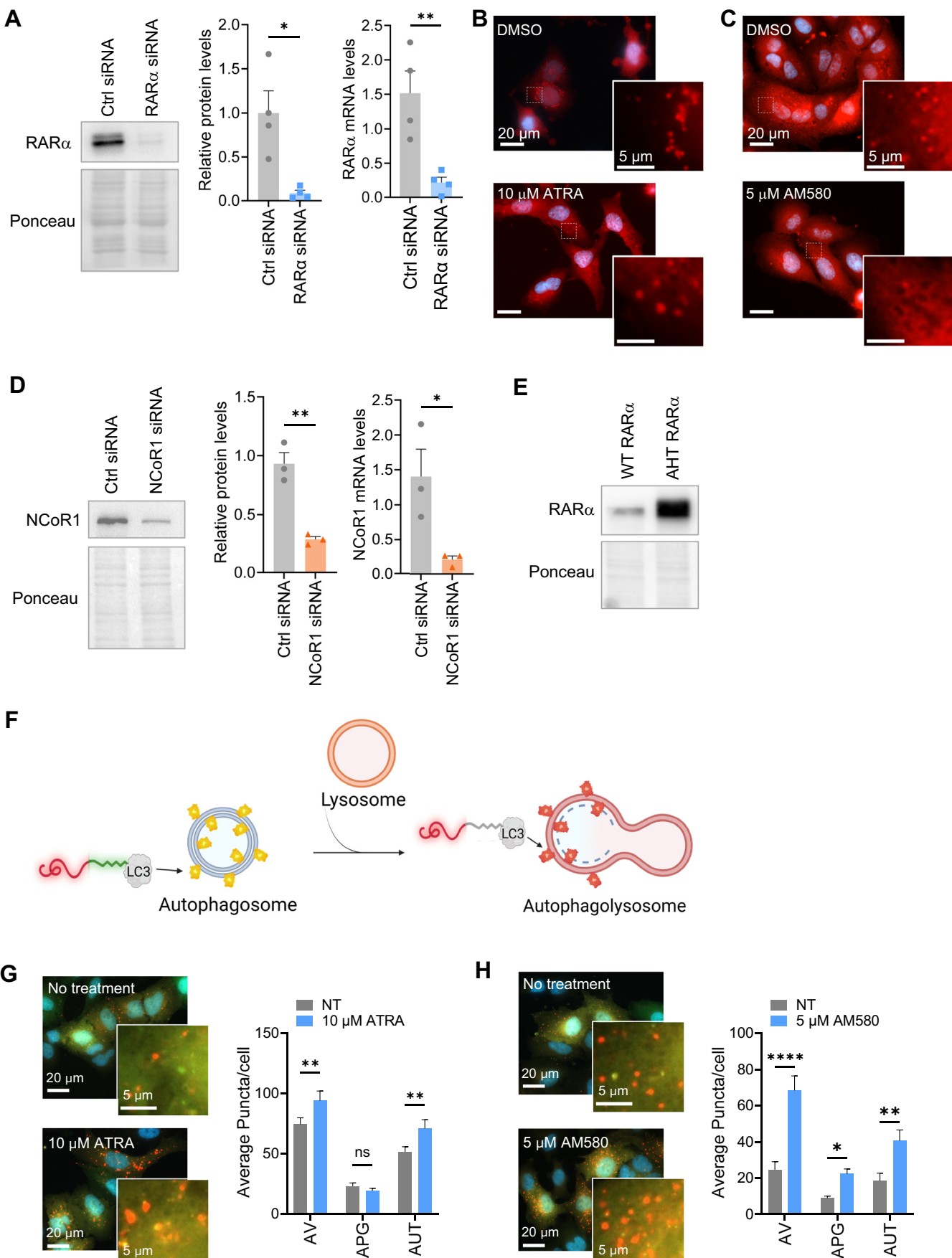

**Figure EV2. Validation of knockdowns and effect of RAR agonists on macroautophagy.**

(A) RARα protein levels and RNA expression in A549 cells transfected with control (ctrl) or RARα siRNA. Representative immunoblot (left), protein level quantification (center), and RNA expression normalized to control siRNA (right) are shown. $n = 4$ independent experiments. (*$P = 0.0107$, **$P = 0.0081$). (B) Representative fluorescence images of A549 cells expressing KFERQ-PS-Dendra treated with 10 µM ATRA or equal volume DMSO in serum-free media for 24 h. Quantification is shown in Fig. 1E. (C) Representative fluorescence images of A549 cells expressing KFERQ-PS-Dendra treated with 5 µM AM580 or equal volume DMSO for 24 h. Quantification is shown in Fig. 1F. (D) NCoR1 protein levels and RNA expression in A549 cells transfected with control (ctrl) or NCoR1 siRNA. Representative immunoblot (left), protein level quantification (center), and RNA expression normalized to control siRNA (right) are shown. $n = 3$ independent experiments. (**$P = 0.0028$, *$P = 0.0400$). (E) Representative immunoblot (left) of A549 cells expressing KFERQ-PS-Dendra transfected with equal amounts of DNA of WT RARα or AHT RARα. (F) Schematic of mCherry-GFP-LC3 reporter (Kimura et al, 2007). When the reporter is associated to autophagosomes, both mCherry and GFP fluoresce and autophagosomes are visualized as yellow puncta. Once the autophagosome fuses with a lysosome GFP fluorescence is quenched and autophagolysosomes appear as red fluorescent puncta. Created in BioRender. https://BioRender.com/oqvzi51. (G, H) Macroautophagy activity in A549 cells expressing mCherry-GFP-LC3 treated with 10 µM ATRA (G) or 5 µM AM580 (H) or equal volume DMSO (No treatment; NT) for 24 h. Representative images (left) and quantification of number of the indicated autophagic compartments (right). AV: autophagic vacuoles (total number of fluorescent red puncta), APG: autophagosomes (total number of green fluorescent puncta) and AUT: autolysosomes (total number of red – green fluorescent puncta). $n = 35$–45 cells from 2 independent experiments. (G: **$P = 0.0086$, ns = not significant, **$P = 0.0098$, H: ****$P \leq 0.0001$, *$P = 0.0448$, **$P = 0.0015$). Data Information: All values are mean + SEM. Unpaired two-tailed $t$ test (A, D) and two-way ANOVA (G, H) were used. Insets show higher magnification and nuclei are highlighted with DAPI. Source data are available online for this figure.

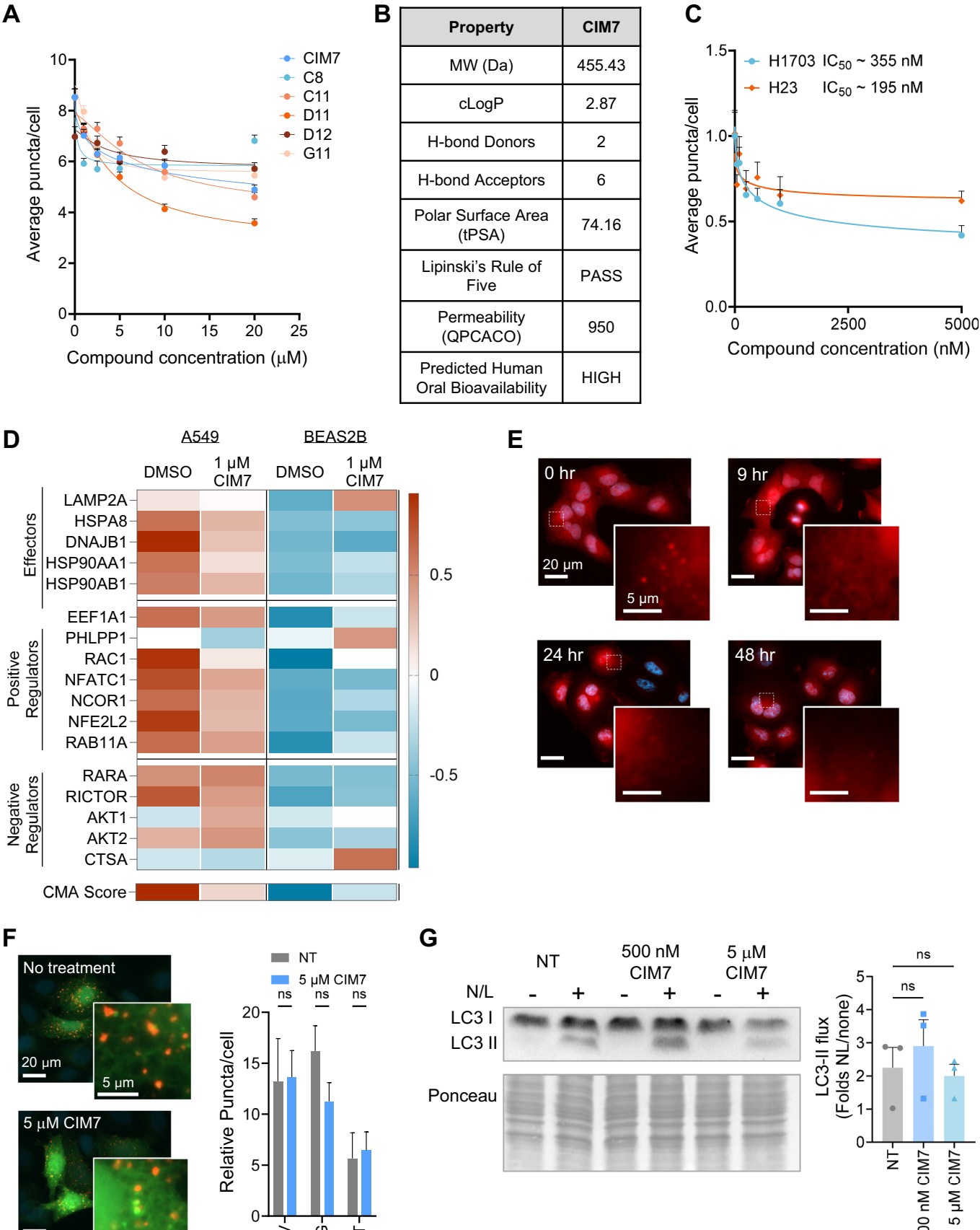

◄ **Figure EV3. CIM7 selectively inhibits CMA in NSCLC cells.**

(A) CMA activity in mouse fibroblasts (NIH-3T3) expressing KFERQ-PS-Dendra and treated with increasing doses of the indicated compounds in media lacking serum quantified as fluorescent puncta per cell. $n = 9$ fields. (B) Calculated chemical properties and predicted ADME properties of CIM7, from QikProp (Schrödinger, LLC) analysis. (C) CMA activity in H1703 and H23 expressing KFERQ-PS-Dendra treated with increasing concentrations of CIM7 based on quantification of fluorescent puncta per cell. $n > 40$ individual cells from 1–3 independent experiments. (D) Heatmap of the z-scores of the transcriptional differences of CMA-related genes (top) and calculated CMA score (bottom) in A549 and BEAS-2B cells 3 h after addition of 1 μM CIM7. $n = 4$ independent experiments. (E) Representative images of A549 cells expressing KFERQ-PS-Dendra treated with 5 μM CIM7 for the indicated times. Insets show higher magnification and nuclei are highlighted with DAPI. Quantification is in Fig. 2I. (F) Macroautophagy activity in A549 cells expressing mCherry-GFP-LC3 treated with DMSO (no treatment: NT) or 5 μM CIM7 for 24 h. Representative images (left) and quantification of number of the indicated autophagic compartments (right). AV autophagic vacuoles, APG autophagosomes, AUT autolysosomes. $n = 85–87$ cells from 3 independent experiments. (ns = not significant). (G) Immunoblot for LC3 in A549 cells treated with the indicated doses of CIM7 in the absence or presence of ammonia chloride and leupeptin (N/L). Representative immunoblot (left) and quantification of LC3-II flux (calculated by levels of LC3-II in the presence of N/L over levels of LC3-II in the absence of N/L) (right). Ponceau staining is shown as loading control. $n = 3$ independent experiments. (ns = not significant). Data information: All values are mean + SEM with individual data points to represent individual experiments. Nonlin fit (A, C), two-way ANOVA (F), or ordinary one-way ANOVA (G) followed by Bonferroni's multiple comparisons post-hoc test were used. Source data are available online for this figure.

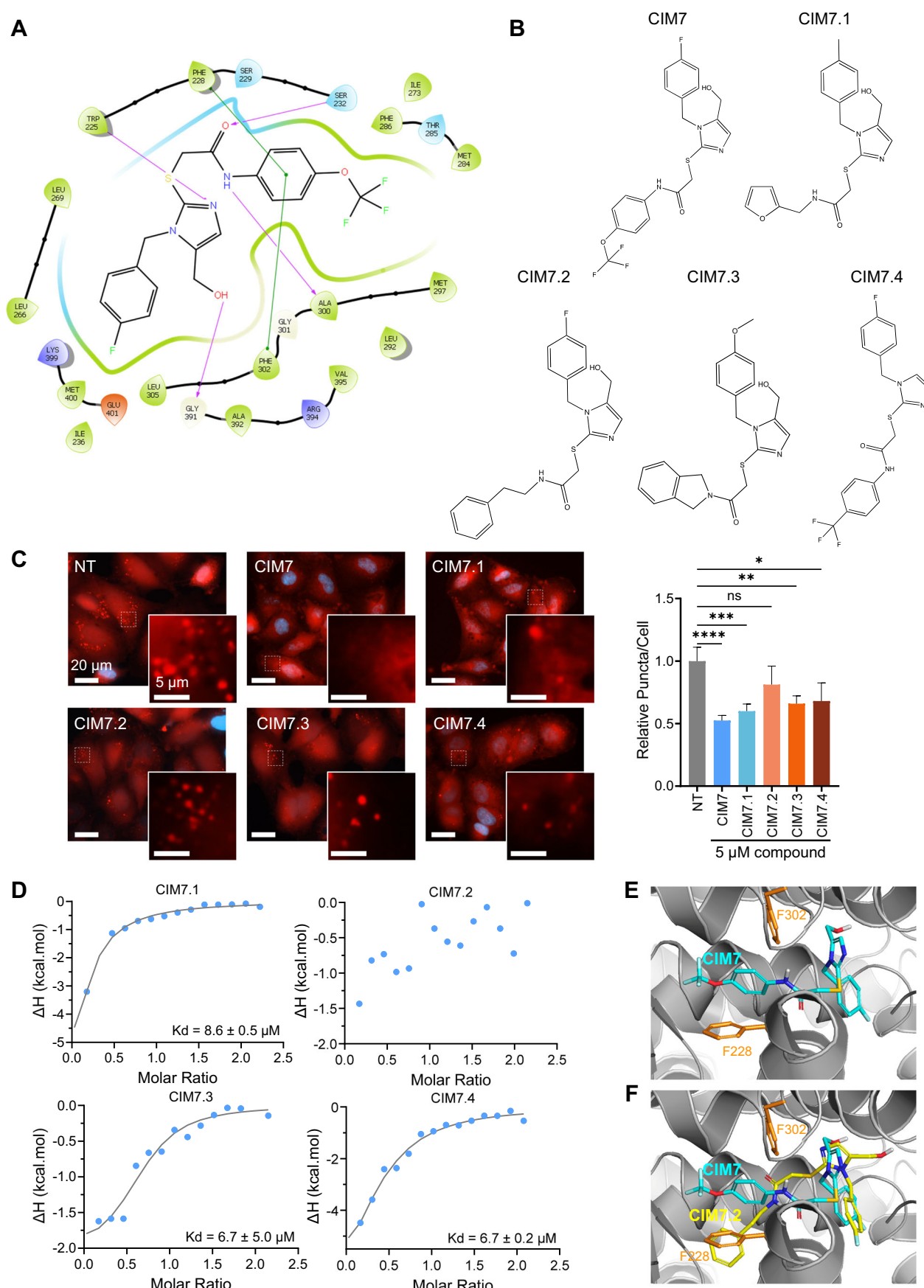

◄ **Figure EV4. Evaluation of CIM7 analogs.**

(A) Predicted ligand interactions occurring between CIM7 and RARα based on 1000 ns molecular dynamic simulation. (B) Compound structures of CIM7 and four analogs. (C) CMA activity, relative to no treatment (NT), in A549 cells expressing KFERQ-PS-Dendra treated with 5 μM compound for 24 h. Representative images (left), with insets showing higher magnification. Nuclei are highlighted with DAPI. CMA activity quantified by number of fluorescent puncta/cell (right). $n = 63$–132 cells from 2–3 independent experiments. (****$P \leq 0.0001$, ***$P = 0.0004$, ns=not significant, **$P = 0.0036$, *$P = 0.0195$). (D) Representative isothermal titration calorimetry curves for the four CIM7 analogs binding to recombinant RARα. These experiments were repeated 2–3 times with consistent results. (E, F) CIM7 (cyan) binding pose generated after 1000 ns molecular dynamic simulation (E) or with the overlay of CIM7.2 (yellow) induced fit docking binding pose (F). Phe228 and Phe302 residues are highlighted in orange. Data information: Values are mean + SEM. Ordinary one-way ANOVA (C) was used. Source data are available online for this figure.

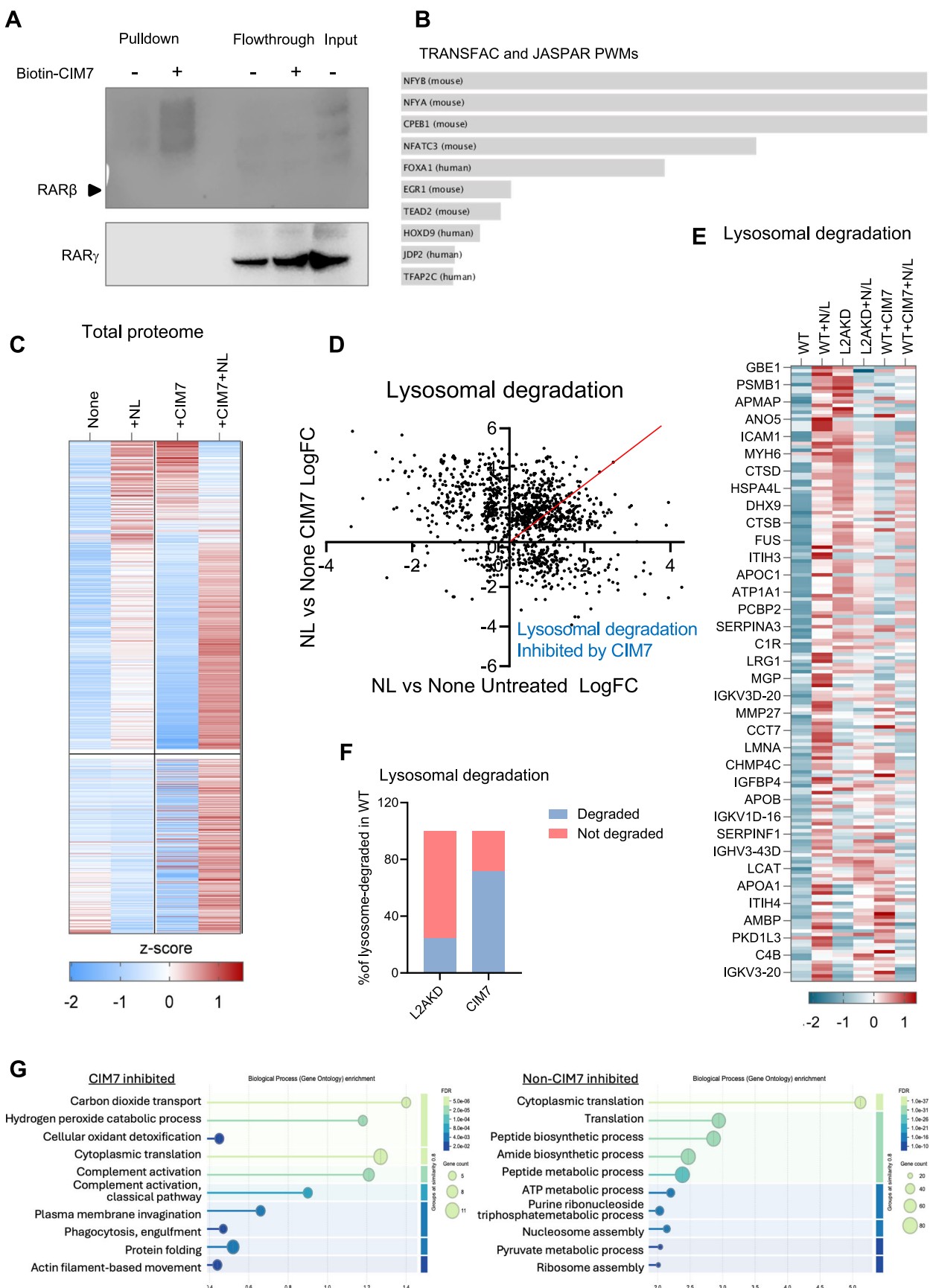

Figure EV5. **CIM7 selectively targets RARα and affects degradation of a subset of CMA substrates.**

(A) Representative immunoblot for RARβ (top) and RARγ (bottom) of streptavidin pulldowns (left) and flowthrough (right) of A549 cellular lysate incubated without additions or with biotin-CIM7 (50 μM). This experiment was repeated twice with consistent results. (B) TRANSFAC and JASPAR PWMs obtained through analysis with Enrichr of genes significantly altered upon CIM7 treatment in the presence of AHT RARα in A549 cells. (C) Heat map as Z-score of the full proteome of A549 cells untreated (none) or treated with CIM7 in presence or not of lysosomal proteolysis inhibitors (N/L). Top quadrant is shown in Fig. 5A. (D) Log2 fold changes (logFC) in rates of lysosomal degradation in untreated against CIM7-treted A549 cells. (E) Heat map (of z-scores) of changes in the subset of proteins degraded in lysosomes in A549 cells untreated (WT), in presence of CIM7 or upon LAMP2A knockdown (L2AKD). (F) Percentage of lysosome-degraded proteins in A549 cells displaying inhibited lysosomal degradation in the same cells as in (E). (G) Gene Ontology enrichment of CMA substrates whose degradation is inhibited (left) or not (right) by CIM7 treatment in WT A549 cells, as predicted by STRING analysis. Bar coloring corresponds to false discovery rate (FDR). Source data are available online for this figure.

**A**

| | | Vehicle | 25 mg/kg CIM7 | Normal Range |
|---|---|---|---|---|
| **WBC (K/mL)** | Day 0 | 4.8 ± 0.5 | 3.7 ± 0.4 | 1.42 – 10.25 |
| | Day 14 | 4.6 ± 0.3 | 4.9 ± 0.4 | |
| | Day 28 | 5.0 ± 0.6 | 6.2 ± 0.7 | |
| **RBC (K/mL)** | Day 0 | 10.1 ± 0.2 | 10.8 ± 0.2 | 6.82 – 10.53 |
| | Day 14 | 10.0 ± 0.2 | 9.3 ± 0.2 | |
| | Day 28 | 9.7 ± 0.2 | 9.5 ± 0.3 | |
| **Platelets (K/mL)** | Day 0 | 1034 ± 95 | 958 ± 107 | 376 – 1796 |
| | Day 14 | 964 ± 64 | 871 ± 89 | |
| | Day 28 | 881 ± 30 | 836 ± 96 | |

**B**

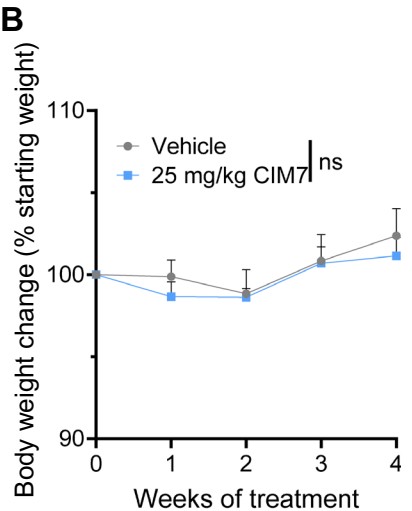

**C**

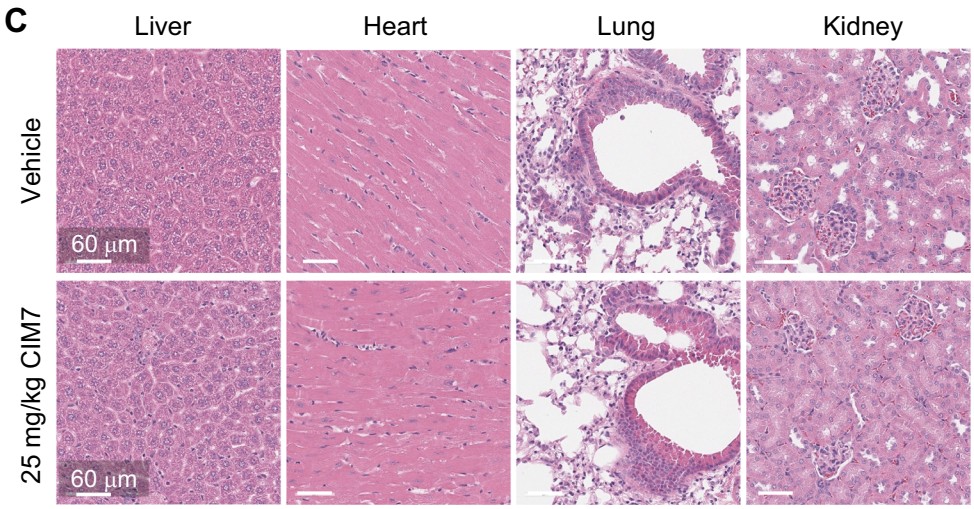

**D**

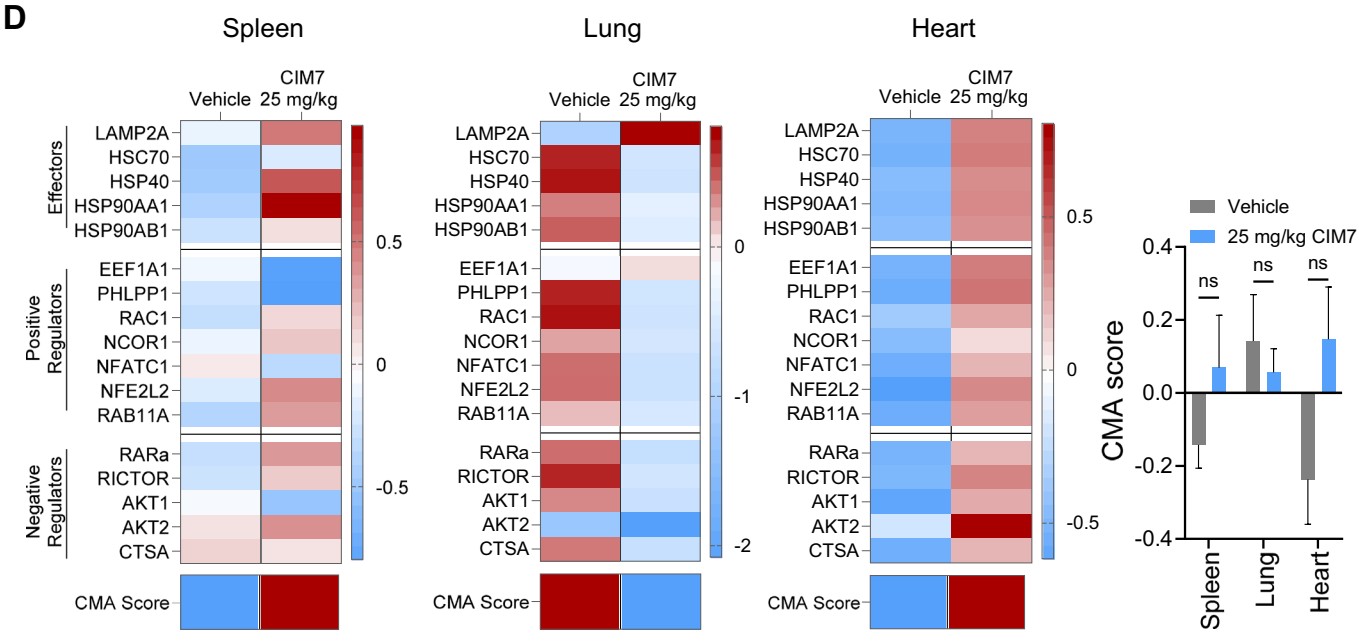

◀ **Figure EV6. CIM7 has no apparent toxicity in vivo.**

(A) Complete blood cell counts of vehicle- or 25 mg/kg CIM7-treated mice at the indicated days of treatment. The normal range for each blood count is shown in the right column. $n = 9$–11 mice per treatment group. (B) Changes in weight, as a percentage of initial body weight, over time in vehicle- or 25 mg/kg CIM7-treated mice, measured weekly. $n = 9$–11 mice per treatment group. (ns = not significant). (C) H&E staining of liver, heart, lung, and kidney from representative vehicle- and 25 mg/kg CIM7-treated mice. (D) Heatmaps (left) and quantification (right) of CMA score changes occurring in the spleen, lung, and heart of vehicle- or 25 mg/kg CIM7-treated mice. $n = 9$–11 mice per treatment group. (ns = not significant). Data information: All values are mean + SEM. Ordinary two-way ANOVA followed by multiple comparisons post-hoc test (B, D) was performed. RBC red blood cell, WBC white blood cell. Source data are available online for this figure.

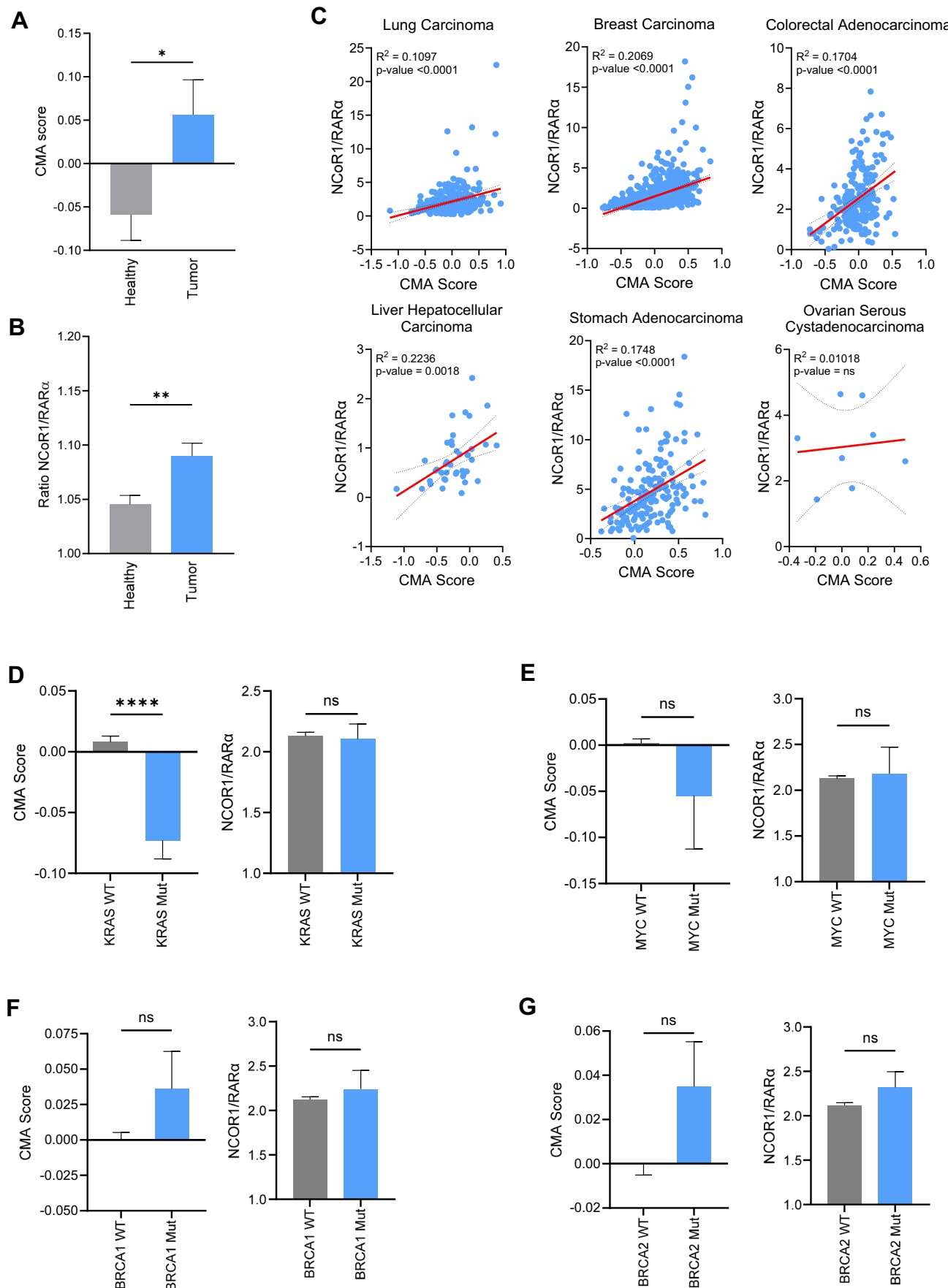

◄ **Figure EV7. Evaluation of CMA score in various cancer types.**

(A, B) CMA score (A) and NCoR1/RARα ratio (B) calculated from RNA-seq of tumor and surrounding healthy lung tissue from NSCLC patients. Data from (Sanchez-Palencia et al, 2011). $n > 44$ individual patients. (A): *$P = 0.0248$, (B): **$P = 0.0029$. (C) Correlation of NCoR1/RARα ratio and CMA score in select cancers. Each individual data point represents an individual patient sample. Data from the TCGA Pan Cancer Altas (Liu et al, 2018). $n > 8$ individual tumors. (D–G) CMA score (left) and NCoR1/RARα ratio (right) calculated in tumors with wild-type (WT) or mutant (Mut) KRAS (D), MYC (E), BRCA1 (F), or BRCA2 (G). Data from the TCGA Pan Cancer Altas (Liu et al, 2018). $n > 38$ individual tumors. (****$P < 0.0001$, ns = not significant). Data information: All bar values are mean + SEM. Unpaired two-tailed $t$ test (A, B, D–G) and simple linear regression (C) were used. Source data are available online for this figure.

