## [Peer Review File · EMBO Molecular Medicine]

Small Molecule Disruption of RAR α /NCoR1 Interaction Inhibits Chaperone-Mediated Autophagy in Cancer

Mericka McCabe, Rajanya Bhattacharyya, Rebecca Sereda, Olaya Santiago-Fernández, Rabia Khawaja, Antonio Diaz, Kristen Lindenau, Deniz Gulfem Ozturk, Thomas Garner, Simone Sidoli, Ana Cuervo, and Eviropidis Gavathiotis

Corresponding authors: Eviropidis Gavathiotis (evriropidis.gavathiotis@einsteinmed.edu) , Ana Cuervo (ana-maria.cuervo@einsteinmed.edu)

Review Timeline:

Submission Date:	24th Jul 24
Editorial Decision:	3rd Sep 24
Revision Received:	21st Mar 25
Editorial Decision:	30th Apr 25
Revision Received:	10th May 25
Accepted:	16th May 25

Editor: Lise Roth

Transaction Report:

3rd Sep 2024

Dear Prof. Gavathiotis,

Thank you for the submission of your manuscript to EMBO Molecular Medicine, and please accept my apologies for the delay in getting back to you in this busy time of the year. We have now received feedback from the three reviewers who agreed to evaluate your manuscript. As you will see from the reports below, the referees acknowledge the interest of the study, but nevertheless make a number of suggestions to improve the translational potential of the study, including analysis of the off-target properties of CIM7, inclusion of an established CMA inhibitor as positive control, generation of a RAR α resistant version, study of the synergistic potential of CIM7 inhibition and targeted therapies / chemotherapies, pharmacodynamics of CIM7 in normal and tumor tissues in vivo, etc.

Upon further cross-commenting with the referees, we acknowledged that addressing all concerns would require a lot of additional work, and that priority should be given to providing details on compound specificity and on the biochemical mechanisms leading to growth inhibition in vitro and in vivo. On the other hand, while certainly interesting, studies of synergy with other drugs will not be mandatory for further consideration of the manuscript and could be discussed in the manuscript instead.

Acceptance of the manuscript will entail a second round of review. EMBO Molecular Medicine encourages a single round of revision only and therefore, acceptance or rejection of the manuscript will depend on the completeness of your responses included in the next, final version of the manuscript. For this reason, and to save you from any frustrations in the end, I would strongly advise against returning an incomplete revision.

We are expecting your revised manuscript within 3-4 months, if you anticipate any delay, please contact us.

We require:

- 1) A .docx formatted version of the manuscript text (including legends for main figures, EV figures and tables). Please make sure that the changes are highlighted to be clearly visible.
- 2) Individual production quality figure files as .eps, .tif, .jpg (one file per figure). For guidance, download the 'Figure Guide PDF' (<https://www.embopress.org/page/journal/17574684/authorguide#figureformat>).
- 3) At EMBO Press we ask authors to provide source data for the main figures. Our source data coordinator will contact you to discuss which figure panels we would need source data for and will also provide you with helpful tips on how to upload and organize the files.
- 4) A .docx formatted letter INCLUDING the reviewers' reports and your detailed point-by-point responses to their comments. As part of the EMBO Press transparent editorial process, the point-by-point response is part of the Review Process File (RPF), which will be published alongside your paper.
- 5) A complete author checklist, which you can download from our author guidelines (<https://www.embopress.org/page/journal/17574684/authorguide#submissionofrevisions>). Please insert information in the checklist that is also reflected in the manuscript. The completed author checklist will also be part of the RPF.
- 6) All Materials and Methods need to be described in the main text using our 'Structured Methods' format. According to this format, the Methods section includes a Reagents and Tools Table (listing key reagents, experimental models, software and relevant equipment and including their sources and relevant identifiers) followed by a Methods and Protocols section describing the methods, ideally using a step-by-step protocol format. The aim is to facilitate adoption of the methodologies across labs. Please download and fill our Reagents and Tools Table template (.docx), which you can find in our author guidelines: <https://www.embopress.org/page/journal/14693178/authorguide#structuredmethods>. When submitting your revised manuscript, please do not include the Reagents and Tools Table in the Methods section of the manuscript but upload it as a separate file choosing the file type "Reagent Table".
- 7) Please note that all corresponding authors are required to supply an ORCID ID for their name upon submission of a revised

manuscript.

8) It is mandatory to include a 'Data Availability' section after the Materials and Methods. Before submitting your revision, primary datasets produced in this study need to be deposited in an appropriate public database, and the accession numbers and database listed under 'Data Availability'. Please remember to provide a reviewer password if the datasets are not yet public (see <https://www.embopress.org/page/journal/17574684/authorguide#dataavailability>).

9) For data quantification: please specify the name of the statistical test used to generate error bars and P values, the number (n) of independent experiments (specify technical or biological replicates) underlying each data point and the test used to calculate p-values in each figure legend. The figure legends should contain a basic description of n, P and the test applied. Graphs must include a description of the bars and the error bars (s.d., s.e.m.). Please provide exact p values.

10) Our journal encourages inclusion of *data citations in the reference list* to directly cite datasets that were re-used and obtained from public databases. Data citations in the article text are distinct from normal bibliographical citations and should directly link to the database records from which the data can be accessed. In the main text, data citations are formatted as follows: "Data ref: Smith et al, 2001" or "Data ref: NCBI Sequence Read Archive PRJNA342805, 2017". In the Reference list, data citations must be labeled with "[DATASET]". A data reference must provide the database name, accession number/identifiers and a resolvable link to the landing page from which the data can be accessed at the end of the reference. Further instructions are available at .

11) We replaced Supplementary Information with Expanded View (EV) Figures and Tables that are collapsible/expandable online. A maximum of 5 EV Figures can be typeset. EV Figures should be cited as 'Figure EV1, Figure EV2" etc... in the text and their respective legends should be included in the main text after the legends of regular figures.

12) The paper explained: EMBO Molecular Medicine articles are accompanied by a summary of the articles to emphasize the major findings in the paper and their medical implications for the non-specialist reader. Please provide a draft summary of your article highlighting

- the medical issue you are addressing,

- the results obtained and

- their clinical impact.

13) Author contributions: CRedit has replaced the traditional author contributions section because it offers a systematic machine readable author contributions format that allows for more effective research assessment. Please remove the Authors Contributions from the manuscript and use the free text boxes beneath each contributing author's name in our system to add specific details on the author's contribution. More information is available in our guide to authors.

Please also suggest a striking image or visual abstract to illustrate your article as a PNG file 550 px wide x 300-600 px high. A cropped portion of this image will serve as thumbnail for the table of content on our webpage.

16) As part of the EMBO Publications transparent editorial process initiative (see our Editorial at <http://embomolmed.embopress.org/content/2/9/329>), EMBO Molecular Medicine will publish online a Review Process File (RPF)

to accompany accepted manuscripts.

In the event of acceptance, this file will be published in conjunction with your paper and will include the anonymous referee reports, your point-by-point response and all pertinent correspondence relating to the manuscript. Let us know whether you agree with the publication of the RPF and as here, if you want to remove or not any figures from it prior to publication. Please note that the Authors checklist will be published at the end of the RPF.

I look forward to receiving your revised manuscript.

Yours sincerely,

Lise Roth

***** Reviewer's comments *****

Referee #1 (Comments on Novelty/Model System for Author):

In the study by McCabe M et al identified a new small molecular chemical CMA inhibitor CIM7. They reported the underlying mechanism by disruption of the NCoR1/RARa complex, leading to suppression of CMA in NSCLC cells. They further performed in vivo animal studies to demonstrate the potential therapeutic value of CIM7 for NSCLC. Overall, the whole study appears to be well executed with substantial amount of data to support the conclusion.

Referee #1 (Remarks for Author):

However, there are key issues in this MS that make the conclusion of the whole study confusing and re-reliable.

- 1) Conflicting hypothesis regarding the role of the NCoR1/RARa complex in CMA. It was clearly stated in the Introduction section that this complex plays a positive role in CMA. However, their data shows that "treatment with the RAR ligand ATRA, which activates RARa transcriptional signaling, inhibited CMA (Fig. 1g, Extended Data Fig. 2f). Similarly, an established and selective RARa agonist, AM580, inhibited CMA (Fig. 1h, Extended Data Fig. 2g)" (page 6/48). If so, this is directly conflicting to the proposed mechanism of CIM7 on CMA which is screened based on its effect on disruption of the NCoR1/RARa complex.
- 2) It is not clearly described how CIM7 disrupts the NCoR1/RARa complex. In Figure 3, the authors used in silico docking screen and molecular dynamics simulations and provided data to show that CIM7 directly binds to RARa, which results in dissociation of NCoR1 already bound to RARa, leading to suppression of the NCoR1/RARa complex. If so, is NCoR1 a corepressor or coactivator of RARa? Moreover, the following statement made on page 10/48 "This differential regulation of CIM7 on the corepressor rather than coactivator, consistent with its distinct predicted RARa binding site and shift from the inactive to the active conformation, may explain the selectivity of CIM7 for CMA inhibition" caused more confusion on the mechanisms for the inhibitory effect of CIM7 on the NCoR1/RARa complex.
- 3) For the in vivo study, the authors should include a well established CMA inhibitor as a positive control to strengthen the conclusion that CIM7's therapeutic effects is mediated via the suppression of CMA.
- 4) This reviewer strongly suggest the authors add one illustration to summarize the main findings esp to show the mechanisms how CIM7 inhibits CMA via disruption of the NCoR1/RARa complex.

Referee #2 (Remarks for Author):

Synopsis and general recommendations

In their manuscript, McCabe et al describe a potentially therapeutic approach to inhibiting the chaperone-mediated autophagy

(CMA) pathway in cancer cells. The group of Ana Maria Cuervo originally characterized the CMA pathway and previously published on its importance for pathophysiology of various diseases, most remarkably neurodegeneration and more recently cancer. Here, McCabe et al. specifically looked at the transcriptional program behind the CMA upregulation in non-small cell lung cancer (NSCLC) cell lines consisting of the RARa/NCoR1 axis. They showed that:

- (1) The RARa/NcoR1 complex is upregulated (e.g., immunofluorescence staining data) in several NSCLC cell lines compared to an immortalized human lung epithelial cell line BEAS-2B, which corresponds to an increased CMA activity as shown by a reporter system (KFERQ-PS-Dendra) and a transcriptomics analysis for genes playing a critical role in the CMA (e.g. LAMP2A and HSP90AA1).
- (2) Downregulation of the individual components of the RARa/NcoR1 complex by RNAi impacted the CMA pathway activity: RARa downregulation leads to CMA activation, and NcoR1 downregulation leads to CMA inhibition. Also, disruption of the RARa/NcoR1 complex formation via the AHT RARa mutant leads to inhibition of the CMA. Like in the previously published work by Cuervo et al on the negative regulation of the CMA by RARa (Anguiano et al 2013 Nat Chem Biol, Gomez-Sintez et al 2022 Nat Comms), also here the established RARa agonists (ATRA and AM580) led to CMA inhibition.
- (3) It is possible to identify small molecules that disrupt the RARa/NcoR1 complex formation by molecular docking using the RARa co-crystal model with an NcoR1 alpha-helix (outside of the canonical RARa binding site). These compounds inhibited the CMA pathway in the NIH-3T3 reporter cell line as well as in NSCLC cells A549. Based on cellular potency in the CMA assay (IC50 of 75 nM), CIM7 was identified as the lead compound for further studies.
- (4) CIM7, the newly identified CMA inhibitor (based on the suggested RARa:NcoR complex disruption), impacts the transcriptional program mediated by the RARa/NcoR1 axis. The changes in the CMA genes induced by CIM7 had a remarkable difference between the NSCLC A549 cells (with intrinsically high CMA activity) and normal lung cells BEAS-2B (with intrinsically low CMA activity).
- (5) CIM7 disrupts the RARa:NcoR complex likely by fixing RARa in an active conformation, preventing NcoR from binding to RARa. CIM7 directly binds RARa as demonstrated by the use of biotinylated CIM7 in a pulldown assay. An ITC (isothermal calorimetry) experiment determined the CIM7:RARa binding constant (Kd) of approx. 2 μ M, which may be quite high for a specific interaction (the authors explain the measured high Kd by using a fragment of the RARa protein rather than the full-length construct). Importantly, in an FP (fluorescence polarization) assay, CIM7 (at 25 μ M) was able to displace NcoR1 from its complex with RARa. Also crucially, CIM7 does not alter the binding of RARa to its other cognate partner transcriptional coactivator SRC, differentiating CIM7 from classical RARa agonists.
- (6) Given its claimed mode of action (MoA) blocking direct interaction between RARa and NcoR, CIM7's inhibitory potential depends on the relative levels of RARa and NcoR in cells. Thus, in A549 cells treated with siRNA against RARa or NcoR, CIM7's activity on the CMA pathway seems to be reduced.
- (7) CIM7 inhibits proliferation of NSCLC cell lines with IC50 of 15-24 μ M, while the normal lung cell line BEAS-2B is less sensitive to the CMA inhibition by CIM7. A modified CIM7 analog CIM7.2, which fails to bind RARa consistently, also does not show any toxicity against A549 cells suggesting specificity of CIM7's cytotoxic activity via RARa binding. CIM7 also inhibited colony formation b A549 cells with an IC50 of 8.7 μ M.
- (8) CIM7 reduces CMA and A549 tumor growth in mice in vivo. CIM7 was given QD at 25 mg/kg using the i.p. route of administration, which slowed down the growth of this tumor type in the athymic nude mice. PD (pharmacodynamic) analysis of isolated tumors showed changes in CMA gene expression as predicted based on the assumed MoA of CIM7. Interestingly, more aggressive tumors were refractory to CIM7 treatment, suggestive some differences between the more and less aggressive tumors towards CMA inhibition (of note, CMA gene expression score in response to CIM7 treatment were similar between the responder and non-responder tumors).
- (9) While RARa levels are similar across NSCLC and normal lung specimens obtained from humans, NcoR expression is increased in the case of lung cancers, impacting the RARa/NcoR expression ratio in analyzed human lung cancers. By extrapolation, this may suggest an increased CMA activity and higher sensitivity to CMA inhibition of NSCLC tumors to compounds, like CIM7.

This is an interesting and important study that attempts to link suggested perturbations in protein homeostasis (reflected in an increased CMA activity suggestive of the need to dispose of more proteins by the CMA-dependent delivery in the lysosomes) in NSCLC tissues with a specific transcriptional program mediated by the RARa/NcoR transcriptional complex and exploit this link therapeutically. The study is comprehensive in that it starts by characterizing the CMA pathway and its dependence on the RARa/NcoR complex and then moves to identifying and characterizing a compound that inhibits the CMA pathway by interfering with the RARa/NcoR complex formation, while locking RARa in an active conformation. The compound first discovered through molecular docking and then validated for binding to its target and the alleged MoA is demonstrated to inhibit the target pathway (the CMA) and cancer cell proliferation both in vitro and in vivo. Finally, the study also provides some evidence for the relevance of the RARa/NcoR pathway in human NSCLC tissue.

I would support publication of the authors' findings provided specific comments below can be addressed to improve the translational significance of the presented work.

Specific comments

- (1) Because of the central idea of this study to link RARa/NCoR expression ratio to the CMA activity in cancer, it would be important to present a bioinformatics analysis that looks at RARa-dependent & CMA pathway gene expression more broadly across human entities (e.g., in the TCGA databank). How common are CMA pathway alterations in cancer? Is there any cancer-specific driver (an oncogene or tumor suppressor or perhaps cancers with serious perturbation in protein homeostasis, like

multiple myeloma) to explain the RAR α -CMA link? This topic should in the very least be mentioned in the Introduction and Discussion sections.

(2) Based on the authors' analysis, disruption of the RAR α /NCoR complex by siRNA (i.e., reduction of NCoR expression) or small molecule (CIM7) leads to CMA pathway inhibition. Similarly, expression of the AHT RAR α mutant leads to inhibition of the CMA. Does the AHT RAR α mutant still bind CIM7? Is that possible to generate a RAR α mutant that would not be bound to CIM7 and in this way insensitive to CIM7 activity? Such a tool would be ideal to demonstrate the specificity of CIM7 to RAR α in the cells. For example, transcriptomic changes measured upon CIM7 treatment of the CIM7-insensitive RAR α mutant could reveal which other targets CIM7 may bind and cause off-target effects. More generally, the off-target properties of CIM7 need to be discussed. There are commercially available protein panels (such as CEREP) that can be used to screen for non-specific CIM7 binding. In the very least CIM7 should be assessed bioinformatically and/or experimentally for its binding to other RAR family members (e.g., RAR-beta and RAR-gamma).

(3) In addition to the KFERQ-PS-Dendra assay, which reveals CIM7 activity towards the CMA pathway, it would be desirable to have a more natural CMA substrate assessed for its accumulation upon CIM7 treatment. This would be key for assessing pharmacodynamics of CIM7 in normal and tumor tissues in vivo. Gene expression changes demonstrated by the authors as a PD biomarker can be prone to other non-CMA effects of RAR α activation.

(4) I appreciate the authors demonstrating that CIM7 disrupts RAR α :NCoR binding (Fig. 3f). However, it would be key to determine the IC₅₀ for CIM7 inhibitory activity in the RAR α :NCoR binding assay. Correlating the K_d value of RAR α :CIM7 binding with the value for the IC₅₀ of RAR α :NCoR binding and then with the IC₅₀ on CMA pathway in a cellular assay can help guide compound optimization efforts started in this study. It can add further information regarding the specificity of the suggested MoA of the small molecule inhibitor.

(5) When determining CIM7 PK in mice, it would be desirable to introduce the FUB (Fraction Unbound) value for the compound and then declare the free compound concentration in the plasma and possibly in the tumor. Free PK would allow correlation between the IC₅₀ measured in the cellular assays in vitro and the activity observed in vivo. Is the compound above the free IC₅₀-90 in the plasma / in the tumor?

(6) Given the drastic differences in CIM7 responses observed for rapidly growing and slowly growing A549 tumors in mice, it would be important to test whether RAR α /NCoR expression ratio or absolute levels dynamically change during the tumor growth, making them more or less sensitive to CMA inhibition. Even though the authors claim that downstream CMA gene expression scores were similar between responders and non-responders, suggesting similar RAR α /NCoR expression ration, absolute levels of RAR α and NCoR might have been different, so that the same dose of CIM7 may have had a different impact based on different stoichiometry between the compound (CIM7) and its target (RAR α).

(7) As mentioned above, inhibition of CMA in some key normal tissues would be important to report as this would support the claim on the available therapeutic window for the CIM7 compound. Is there a way to measure CMA substrate accumulation in healthy tissues?

Referee #3 (Comments on Novelty/Model System for Author):

Expansion to more relevant models needed.

Referee #3 (Remarks for Author):

In the present work, McCabe et al developed CIM7, a novel inhibitor of the chaperone-mediated autophagy (CMA), which appears to act by preventing the interaction of the NCoR1 co-repressor with the RAR α complex. By leveraging biochemical and functional assay combined with preclinical models of non-small cell lung cancer (NSCLC), the authors demonstrate the potential of CMA targeting to reduce the NSCLC cells growth in vivo without affecting healthy tissues.

Overall, the study elucidates a relevant biological mechanism sustaining NSCLC growth and provides critical preclinical data supporting the rationale for CMA targeting in NSCLC patients. While the authors have made significant efforts in attempting to demonstrate that the identified compound specifically interacts with RAR α , evidence mostly relying on docking models and assays showing the impact of treatment in cells and xenografts on the RAR α -NCoR interaction, there are significant issues that should be addressed to univocally demonstrate that the growth effects observed are indeed linked to the invoked biochemical mechanism. This also given the fact that RAR α signaling is involved in numerous processes. And given the low potency of the compound and the potential disconnect with its potency in vivo. I would recommend to further explore the impact of the compound on overall gene expression comparing sensitive and resistant cells. I would also recommend the generation of an RAR α resistant version, that could be used to prove the specificity of the observed effects.

To improve the translational relevance of CMA targeting in NSCLC patients, I would recommend addressing the following points:

1. The authors reported no toxicity in mice upon CIM7 administration, which might be due to cross-species differences in the NCoR1/RAR α complex structure and/or protein sequences. The authors should provide information about CIM7 target engagement in mouse cells, both in vitro and in vivo.
2. NSCLC comprises lung squamous carcinoma (LUSC) and lung adenocarcinoma (LUAD), two diseases having different

histopathological features and genomic traits. Among the cell lines used for the CMA score, it would be interesting to evaluate whether the effects of oncogene addiction (e.g. EGFR, KRAS) frequent in LUAD or the loss of cell cycle checkpoints (e.g. TP53, CDKN2A) prominent in LUSC impact on the CMA activity. Additionally, would the CMA activity be affected by targeted therapy such as EGFR inhibitors or the new class of KRAS inhibitors? Exploring the synergistic potential of CIM7 inhibition and targeted therapies would be insightful.

3. Autophagy is known to be activated in response to various stressors, including chemotherapy, which can induce a potent autophagic flux. Would CIM7 exhibit synergy with platinum-based or other disease relevant treatment/chemotherapy?

4. I also suggest extending the in vivo analysis to include at least one additional NSCLC cell line. It may be worthwhile to include a model of tail vein injection to evaluate the effects of CIM7 on cell seeding.

5. The authors should improve the evaluation of NCoR1/RAR α destabilization upon CIM7 inhibition in vitro.

6. Would suggest modifying terms: untransformed, healthy (non-transformed or non-tumorigenic); demonstrates (suggests, opens the way to) a therapeutic strategy; characterized (found) to contribute; sustainment of aerobic glycolysis (sustained)

We greatly appreciate all the reviewers' insights and valuable comments. To facilitate the review process, we indicated in each answer the figure panels or pages where the new results and text are displayed in the revised manuscript. Changes in the manuscript that address the reviewers' comments are marked in blue fonts.

Referee #1:

In the study by McCabe M et al identified a new small molecular chemical CMA inhibitor CIM7. They reported the underlying mechanism by disruption of the NCoR1/RARa complex, leading to suppression of CMA in NSCLC cells. They further performed in vivo animal studies to demonstrate the potential therapeutic value of CIM7 for NSCLC. Overall, the whole study appears to be well executed with substantial amount of data to support the conclusion.

Authors Response (AR): We appreciate this reviewer's comments about the execution of this study.

However, there are key issues in this MS that make the conclusion of the whole study confusing and re-reliable.

1) Conflicting hypothesis regarding the role of the NCoR1/RARa complex in CMA. It was clearly stated in the Introduction section that this complex plays a positive role in CMA. However, their data shows that "treatment with the RAR ligand ATRA, which activates RARa transcriptional signaling, inhibited CMA (Fig. 1g, Extended Data Fig. 2f). Similarly, an established and selective RARa agonist, AM580, inhibited CMA (Fig. 1h, Extended Data Fig. 2g)" (page 6/48). If so, this is directly conflicting to the proposed mechanism of CIM7 on CMA which is screened based on its effect on disruption of the NCoR1/RARa complex.

AR: We apologize for the confusion regarding our hypothesis. RARa on its own, or in association with the co-activator, SRC, plays an inhibitory role on CMA. The association of NCoR1 with RARa, inhibits RARa activity and its inhibitory role in CMA, and consequently results in CMA activation. The RARa agonists ATRA and AM580 enhance SRC binding to RARa, while CIM7 was screened based on disruption of NCoR1 from RARa. The mechanism by which CIM7 was screened is most similar to the introduction of the AHT RARa mutant, which lacks the ability to bind NCoR1 and results in a subsequent reduction in CMA activity compared to cells expressing WT RARa. To provide clarity about the CIM7 mechanism in the manuscript, we have reordered the panels in **Fig. 1E-H**, added a schematic in **Fig. 3I**, and updated the text on **Page 6-7**.

2) It is not clearly described how CIM7 disrupts the NCoR1/RARa complex. In Figure 3, the authors used in silico docking screen and molecular dynamics simulations and provided data to show that CIM7 directly binds to RARa, which results in dissociation of NCoR1 already bound to RARa, leading to suppression of the NCoR1/RARa complex. If so, is NCoR1 a corepressor or coactivator of RARa? Moreover, the following statement made on page 10/48 "This differential regulation of CIM7 on the corepressor rather than coactivator, consistent with its distinct predicted RARa binding site and shift from the inactive to the active conformation, may explain the selectivity of CIM7 for CMA inhibition" caused more confusion on the mechanisms for the inhibitory effect of CIM7 on the NCoR1/RARa complex.

AR: We again apologize for the confusion. NCoR1 is a corepressor of RARa, while SRC is a coactivator. CIM7 binds to a distinct site of RARa, resulting in the dissociation of NCoR1 as shown by molecular dynamics simulations (**Fig. 3A,C,D**), FP binding assay (**Fig. 3E**), and new data from a proximity ligation assay (**Fig. 4B**). Dissociation of NCoR1 by CIM7 is predicted to induce protein conformation change of RARa from the inactive (NCoR1-bound) towards the active RARa conformation. The active RARa conformation is compatible with SRC coactivator binding. Therefore, the binding of CIM7 is distinct from agonists like ATRA and AM580 that bind RARa and enhance the interaction with SRC coactivator. To clarify further in the manuscript, we have expanded the scheme comparing CIM7 to ATRA in **Fig. 3I**.

3) *For the in vivo study, the authors should include a well established CMA inhibitor as a positive control to strengthen the conclusion that CIM7's therapeutic effects is mediated via the suppression of CMA.*

AR: We appreciate the reviewer's suggestion. Unfortunately, the field lacks a specific CMA inhibitor to serve as a positive control, thus limiting our experimental capabilities. Previous studies in ours and other laboratories have utilized knockdown or knockout of LAMP2A, the rate-limiting component of CMA. We have included a reference to the results of our previous study (Kon et al. *Sci Transl Med*, 2011) using LAMP2A knockdown for CMA inhibition in NSCLC, which behaves similarly to CIM7 treatment, on **page 14 and 15**.

4) *This reviewer strongly suggest the authors add one illustration to summarize the main findings esp to show the mechanisms how CIM7 inhibits CMA via disruption of the NCoR1/RARa complex.*

AR: We appreciate the suggestion and have provided a **synopsis image** that summarizes our findings and mechanism of CIM7 inhibiting CMA.

Referee #2:

In their manuscript, McCabe et al describe a potentially therapeutic approach to inhibiting the chaperone-mediated autophagy (CMA) pathway in cancer cells. The group of Ana Maria Cuervo originally characterized the CMA pathway and previously published on its importance for pathophysiology of various diseases, most remarkably neurodegeneration and more recently cancer. Here, McCabe et al. specifically looked at the transcriptional program behind the CMA upregulation in non-small cell lung cancer (NSCLC) cell lines consisting of the RARa/NCoR1 axis. They showed that:

(1) The RARa/NcoR1 complex is upregulated (e.g., immunofluorescence staining data) in several NSCLC cell lines compared to an immortalized human lung epithelial cell line BEAS-2B, which corresponds to an increased CMA activity as shown by a reporter system (KFERQ-PS-Dendra) and a transcriptomics analysis for genes playing a critical role in the CMA (e.g. LAMP2A and HSP90AA1).

(2) Downregulation of the individual components of the RARa/NcoR1 complex by RNAi impacted the CMA pathway activity: RARa downregulation leads to CMA activation, and NcoR1 downregulation leads to CMA inhibition. Also, disruption of the RARa/NcoR1 complex formation

via the AHT RARa mutant leads to inhibition of the CMA. Like in the previously published work by Cuervo et al on the negative regulation of the CMA by RARa (Anguiano et al 2013 Nat Chem Biol, Gomez-Sintez et al 2022 Nat Comms), also here the established RARa agonists (ATRA and AM580) led to CMA inhibition.

(3) It is possible to identify small molecules that disrupt the RARa/NcoR1 complex formation by molecular docking using the RARa co-crystal model with an NcoR1 alpha-helix (outside of the canonical RARa binding site). These compounds inhibited the CMA pathway in the NIH-3T3 reporter cell line as well as in NSCLC cells A549. Based on cellular potency in the CMA assay (IC₅₀ of 75 nM), CIM7 was identified as the lead compound for further studies.

(4) CIM7, the newly identified CMA inhibitor (based on the suggested RARa:NcoR complex disruption), impacts the transcriptional program mediated by the RARa/NcoR1 axis. The changes in the CMA genes induced by CIM7 had a remarkable difference between the NSCLC A549 cells (with intrinsically high CMA activity) and normal lung cells BEAS-2B (with intrinsically low CMA activity).

(5) CIM7 disrupts the RARa:NcoR complex likely by fixing RARa in an active confirmation, preventing NcoR from binding to RARa. CIM7 directly binds RARa as demonstrated by the use of biotinylated CIM7 in a pulldown assay. An ITC (isothermal calorimetry) experiment determined the CIM7:RARa binding constant (K_d) of pprox.. 2 uM, which may be quite high for a specific interaction (the authors explain the measured high K_d by using a fragment of the RARa protein rather than the full-length construct). Importantly, in an FP (fluorescence polarization) assay, CIM7 (at 25 uM) was able to displace NcoR1 from its complex with RARa. Also crucially, CIM7 does not alter the binding of RARa to its other cognate partner transcriptional coactivator SRC, differentiating CIM7 from classical RARa agonists.

(6) Given its claimed mode of action (MoA) blocking direct interaction between RARa and NcoR, CIM7's inhibitory potential depends on the relative levels of RARa and NcoR in cells. Thus, in A549 cells treated with siRNA against RARa or NcoR, CIM7's activity on the CMA pathway seems to be reduced.

(7) CIM7 inhibits proliferation of NSCLC cell lines with IC₅₀ of 15-24 uM, while the normal lung cell line BEAS-2B is less sensitive to the CMA inhibition by CIM7. A modified CIM7 analog CIM7.2, which fails to bind RARa consistently, also does not show any toxicity against A549 cells suggesting specificity of CIM7's cytotoxic activity via RARa binding. CIM7 also inhibited colony formation b A549 cells with an IC₅₀ of 8.7 uM.

(8) CIM7 reduces CMA and A549 tumor growth in mice in vivo. CIM7 was given QD at 25 mg/kg using the i.p. route of administration, which slowed down the growth of this tumor type in the athymic nude mice. PD (pharmacodynamic) analysis of isolated tumors showed changes in CMA gene expression as predicted based on the assumed MoA of CIM7. Interestingly, more aggressive tumors were refractory to CIM7 treatment, suggestive some differences between the more and less aggressive tumors towards CMA inhibition (of note, CMA gene expression score in response to CIM7 treatment were similar between the responder and non-responder tumors).

(9) While RARa levels are similar across NSCLC and normal lung specimens obtained from humans, NcoR expression is increased in the case of lung cancers, impacting the RARa/NcoR expression ratio in analyzed human lung cancers. By extrapolation, this may suggest an increased CMA activity and higher sensitivity to CMA inhibition of NSCLC tumors to compounds, like CIM7.

This is an interesting an important study that attempts to link suggested perturbations in protein homeostasis (reflected in an increased CMA activity suggestive of the need to dispose of more

proteins by the CMA-dependent delivery in the lysosomes) in NSCLC tissues with a specific transcriptional program mediated by the RARa/NcoR transcriptional complex and exploit this link therapeutically. The study is comprehensive in that it starts by characterizing the CMA pathway and its dependence on the RARa/NcoR complex and then moves to identifying and characterizing a compound that inhibits the CMA pathway by interfering with the RARa/NcoR complex formation, while locking RARa in an active conformation. The compound first discovered through molecular docking and then validated for binding to its target and the alleged MoA is demonstrated to inhibit the target pathway (the CMA) and cancer cell proliferation both in vitro and in vivo. Finally, the study also provides some evidence for the relevance of the RARa/NcoR pathway in human NSCLC tissue.

I would support publication of the authors' findings provided specific comments below can be addressed to improve the translational significance of the presented work.

AR: We appreciate the reviewer's positive feedback regarding our study and useful suggestions and comments.

Specific comments

(1) Because of the central idea of this study to link RARa/NCoR expression ratio to the CMA activity in cancer, it would be important to present a bioinformatics analysis that looks at RARa-dependent & CMA pathway gene expression more broadly across human entities (e.g., in the TCGA databank). How common are CMA pathway alterations in cancer? Is there any cancer-specific driver (an oncogene or tumor suppressor or perhaps cancers with serious perturbation in protein homeostasis, like multiple myeloma) to explain the RARa-CMA link? This topic should in the very least be mentioned in the Introduction and Discussion sections.

AR: Thank you very much for this useful suggestion. We have added an analysis of CMA score and the NCoR1/RARa expression ratio across human entities and 30 cancer types utilizing data from the TCGA. Analysis revealed an overall increase in the NCoR1/RARa ratio correlating to increasing CMA score between cancer types. We additionally evaluated CMA score and the NCoR1/RARa ratio in relation to oncogenic mutations. Interestingly, we observed a significant increase in both CMA score and the NCoR1/RARa ratio in patient samples with TP53 mutations compared to those expressing WT TP53. These findings are shown in **Figure 7** and **Appendix Figure S14** and described on **Page 17**.

(2) Based on the authors' analysis, disruption of the RARa/NCoR complex by siRNA (i.e., reduction of NCoR expression) or small molecule (CIM7) leads to CMA pathway inhibition. Similarly, expression of the AHT RARa mutant leads to inhibition of the CMA. Does the AHT RARa mutant still bind CIM7? Is that possible to generate a RARa mutant that would not be bound to CIM7 and in this way insensitive to CIM7 activity? Such a tool would be ideal to demonstrate the specificity of CIM7 to RARa in the cells. For example, transcriptomic changes measured upon CIM7 treatment of the CIM7-insensitive RARa mutant could reveal which other targets CIM7 may bind and cause off-target effects.

AR: We performed pulldowns of biotin-CIM7 with AHT RARa and determined no binding, as shown in **Figure 4D**. Following the reviewer's suggestions, we performed RNA-seq analysis and first confirmed that CIM7 treatment resulted in relatively discrete changes in the transcriptome (252 genes out of 26388 genes) and that those changes phenocopied in part those observed by

expressing the AHT RAR α mutant unable to bind NCoR1 (**Fig. 4E** and **Appendix Fig. S8A**). Interestingly, analysis showed that genes changing significantly upon CIM7 treatment are linked to RAR α and HOXD9, however, HOXD9 is weakly expressed in our A549 cells (**Appendix Fig. S8B**) and therefore an unlikely off-target. As anticipated by the reviewer and in agreement with the proposed mechanism of action of CIM7, addition of CIM7 to cells expressing AHT RAR α failed to reproduce the transcriptional changes observed in presence of WT RAR α (**Appendix Fig. S8A**). The discrete transcriptional changes induced by CIM7 when expressing the AH RAR α mutant suggests possible alternative targets and transcriptional effects when CIM7 is unable to bind RAR α . However, none of those changes occurred in the WT RAR α background, reinforcing the higher preference of CIM7 for NCoR1/RAR α interaction. These results are described in **Pages 12-13**.

More generally, the off-target properties of CIM7 need to be discussed. There are commercially available protein panels (such as CEREP) that can be used to screen for non-specific CIM7 binding. In the very least CIM7 should be assessed bioinformatically and/or experimentally for its binding to other RAR family members (e.g., RAR-beta and RAR-gamma).

AR: We now illustrate the lack of interaction between CIM7 and other RAR family members in A549 cells in **Figure EV4A**. Furthermore, following the reviewer's recommendation, we have evaluated potential off-target interactions of CIM7 against a commercial panel of molecular targets (G-protein-coupled receptors, ionic channels, enzymes, transporters, and nuclear receptors) in **Appendix Figure S8D,E**. We observed *in vitro* binding to five targets (**Appendix Fig. S8D,E**), although these proteins have no or low expression in our A549 cells (**Appendix Fig. S8F**). These results are described in **Page 13**.

(3) In addition to the KFERQ-PS-Dendra assay, which reveals CIM7 activity towards the CMA pathway, it would be desirable to have a more natural CMA substrate assessed for its accumulation upon CIM7 treatment. This would be key for assessing pharmacodynamics of CIM7 in normal and tumor tissues in vivo. Gene expression changes demonstrated by the authors as a PD biomarker can be prone to other non-CMA effects of RAR α activation.

AR: To address this important point, we have performed a comparative whole cell proteomics analysis (**Fig. EV4C**) in the presence or absence of inhibitors of lysosomal proteolysis to identify the subset of the proteome degraded by lysosomes in A549 cells. By comparing changes in the presence or absence of CIM7, we were able to dissect CIM7's specific effect on the inhibition of lysosomal degradation (**Fig. 5A** and **Fig. EV4D**). Furthermore, by adding a LAMP2A KD, we separated the effect on CMA substrates from that of other proteins degraded in lysosomes in a LAMP2A-independent manner (no CMA substrates) (**Fig. 5D, E** and **EV4E-G**). We confirmed that CIM7 was able to inhibit the degradation of endogenous CMA substrates (examples shown in **Fig. 5E** and **Appendix Fig. S10A**). As anticipated, not all proteins undergoing degradation in lysosomes were sensitive to CIM7 inhibition, as these proteins are likely reaching lysosomes through other types of autophagy (as we confirmed by their independence from LAMP2A levels, **Appendix Fig. S10C**). Interestingly, and as somewhat anticipated, since not all CMA is under NCoR1/RAR α regulation, there were also other CMA substrates (degraded in lysosomes in a LAMP2A-dependent manner) whose degradation was not affected by the presence of CIM7 (**Appendix Fig. S10B**). This selectivity of CIM7 for a subset of the CMA-degraded proteome also helps dissipate concerns about the possible promotion of metastasis observed after full ablation of LAMP2A in human cancers (Zhou et al., 2025). The impact of CIM7 on endogenous CMA substrates is described in the text on **Pages 13 and 14**, and the implications of this additional CIM7 selectivity are discussed on **Page 21**.

(4) I appreciate the authors demonstrating that CIM7 disrupts RARα:NCoR binding (Fig. 3f). However, it would be key to determine the IC50 for CIM7 inhibitory activity in the RARα:NCoR binding assay. Correlating the Kd value of RARα:CIM7 binding with the value for the IC50 of RARα:NCoR binding and then with the IC50 on CMA pathway in a cellular assay can help guide compound optimization efforts started in this study. It can add further information regarding the specificity of the suggested MoA of the small molecule inhibitor.

AR: We appreciate the reviewer's suggestion to determine the IC50 value for CIM7 inhibitory activity to further investigate its correlation with the direct Kd value for RARα binding and its IC50 on CMA inhibition in cells. We previously provided a competitive FP assay of RARα:NCoR1 binding, showing that CIM7 displaces the complex at 25 μM (**Fig. 3E**), and a direct binding Kd value of ~2 μM for RARα determined by ITC (**Fig. 3B**). We attempted to determine an IC50 using the FP assay at a constant RARα:NCoR1 concentration while titrating CIM7. However, due to the weak interaction between NCoR1 and RARα, the data showed high variability despite multiple repetitions. Although an IC50 in the range of ~0.5 μM could be estimated in some cases, we feel that the variability limits the reliability of this experiment. In the revised manuscript, we also provide additional evidence of CIM7-induced dissociation of the RARα/NCoR1 interaction in cells using a proximity ligation assay (PLA) at 5 μM CIM7 (**Fig. 4B**) described on **Page 12**, further supporting its mechanism of action. Probing endogenous protein interactions with PLA or a BRET-based read out, combined with ITC-based direct binding analysis to RARα, will help guide the optimization of CIM7 potency and specificity for further development.

(5) When determining CIM7 PK in mice, it would be desirable to introduce the FUB (Fraction Unbound) value for the compound and then declare the free compound concentration in the plasma and possibly in the tumor. Free PK would allow correlation between the IC50 measured in the cellular assays in vitro and the activity observed in vivo. Is the compound above the free IC50-90 in the plasma / in the tumor?

AR: We evaluated CIM7 binding to mouse protein in plasma and determined that binding levels are below the limit of quantification, shown in **Appendix Figure S11**. As a result, and as described on **Page 15**, CIM7 can reach a concentration in plasma at which CMA inhibition is maximal in NSCLC cells.

(6) Given the drastic differences in CIM7 responses observed for rapidly growing and slowly growing A549 tumors in mice, it would be important to test whether RARα/NCoR expression ratio or absolute levels dynamically change during the tumor growth, making them more or less sensitive to CMA inhibition. Even though the authors claim that downstream CMA gene expression scores were similar between responders and non-responders, suggesting similar RARα/NCoR expression ratio, absolute levels of RARα and NCoR might have been different, so that the same dose of CIM7 may have had a different impact based on different stoichiometry between the compound (CIM7) and its target (RARα).

AR: Although a very interesting suggestion, when we evaluated differences in NCoR and RARα gene expression and protein levels in tumors of responsive and unresponsive mice we observed no difference, as shown in **Appendix Figure S13**. We additionally measured the other RAR family members and observed some differences, although they did not reach significance. These levels could only be measured at the end of the 30-day course of treatment, so it is possible that there were differences in stoichiometry between NCoR1 and RARα prior to

treatment that equalized over the course of treatment between the two mouse groups.

(7) As mentioned above, inhibition of CMA in some key normal tissues would be important to report as this would support the claim on the available therapeutic window for the CIM7 compound. Is there a way to measure CMA substrate accumulation in healthy tissues?

AR: We have now evaluated the transcriptional CMA index in several healthy tissues from the treated mice, shown in **Figure EV5D**. In contrast to tumors, transcriptional downregulation of multiple CMA-related genes and reduced CMA score was not observed in healthy mouse tissues. These findings are described on **Page 16**.

Referee #3:

Expansion to more relevant models needed.

In the present work, McCabe et al developed CIM7, a novel inhibitor of the chaperone-mediated autophagy (CMA), which appears to act by preventing the interaction of the NCoR1 co-repressor with the RAR α complex. By leveraging biochemical and functional assay combined with preclinical models of non-small cell lung cancer (NSCLC), the authors demonstrate the potential of CMA targeting to reduce the NSCLC cells growth in vivo without affecting healthy tissues.

Overall, the study elucidates a relevant biological mechanism sustaining NSCLC growth and provides critical preclinical data supporting the rationale for CMA targeting in NSCLC patients. While the authors have made significant efforts in attempting to demonstrate that the identified compound specifically interacts with RAR α , evidence mostly relying on docking models and assays showing the impact of treatment in cells and xenografts on the RAR α -NCOR interaction, there are significant issues that should be addressed to univocally demonstrate that the growth effects observed are indeed linked to the invoked biochemical mechanism. This also given the fact that RAR α signaling is involved in numerous processes. And given the low potency of the compound and the potential disconnect with its potency in vivo. I would recommend to further explore the impact of the compound on overall gene expression comparing sensitive and resistant cells. I would also recommend the generation of an RAR α resistant version, that could be used to prove the specificity of the observed effects.

To improve the translational relevance of CMA targeting in NSCLC patients, I would recommend addressing the following points:

Authors Response (AR): We appreciate the reviewer's positive feedback and have taken at heart the recommendations to improve the translational relevance of our study.

1. *The authors reported no toxicity in mice upon CIM7 administration, which might be due to cross-species differences in the NCoR1/RAR α complex structure and/or or protein sequences. The authors should provide information about CIM7 target engagement in mouse cells, both in vitro and in vivo.*

AR: While the lack of toxicity in CIM7-treated mice could be due to cross-species differences, as proposed by the reviewer, the inhibitory effect on CMA observed in mouse fibroblasts when using high CIM7 concentrations (**Fig. 2B**), speaks against in. However, to directly address the ability of CIM7 to inhibit CMA in mouse fibroblasts, supported by the conserved protein sequence of RAR α between mice and humans, we performed pulldown experiments using the biotin-CIM7 probe for target engagement. Biotin-CIM7 binding to RAR α in mouse cells, was observed although to a lesser extent than in A549 cells, as described on **page 15-16** and in **Appendix Figure S11C, D**. The weak target engagement of CIM7 in mouse fibroblasts aligns with the observed lack of CMA inhibition in healthy mouse tissues (**Fig. EV5D**), but it also matches the lower inhibitory effect and weak engagement of CIM7 with RAR α observed in non-tumoral human lung epithelial BEAS cells (**Fig. 2F** and **Appendix Figure S11D**). These findings further support CIM7's selectivity for cancer cells, both in vitro and in vivo.

2. NSCLC comprises lung squamous carcinoma (LUSC) and lung adenocarcinoma (LUAD), two diseases having different histopathological features and genomic traits. Among the cell lines used for the CMA score, it would be interesting to evaluate whether the effects of oncogene addiction (e.g. EGFR, KRAS) frequent in LUAD or the loss of cell cycle checkpoints (e.g. TP53, CDKN2A) prominent in LUSC impact on the CMA activity. Additionally, would the CMA activity be affected by targeted therapy such as EGFR inhibitors or the new class of KRAS inhibitors? Exploring the synergistic potential of CIM7 inhibition and targeted therapies would be insightful.

AR: We appreciate the reviewer's insightful suggestion. We have added a table of oncogene mutation status e.g. KRAS, TP53, EGFR in our panel of five NSCLC cell lines in **Figure EV1A**. We further illustrate CIM7 ability to inhibit CMA in additional NSCLC cell lines with differential KRAS and TP53 mutation status in **Figure EV2C**. As our focus was on investigating the biochemical effects, mechanism of action, and selectivity of CIM7 as a single agent, we did not specifically assess synergy of CIM7 with EGFR or KRAS inhibitors. This will be an important area for future investigation that will require thorough evaluation.

3. Autophagy is known to be activated in response to various stressors, including chemotherapy, which can induce a potent autophagic flux. Would CIM7 exhibit synergy with platinum-based or other disease relevant treatment/chemotherapy?

AR: We conducted a preliminary evaluation of CIM7 in combination with cisplatin treatment in synergy studies, which suggests a potential additive effect. However, further studies will be needed to fully assess the therapeutic synergy of CIM7 with chemotherapy. Preliminary results are provided below only for reviewer's consideration.

Figure for reviewers removed.

4. I also suggest extending the in vivo analysis to include at least one additional NSCLC cell line. It may be worthwhile to include a model of tail vein injection to evaluate the effects of CIM7 on cell seeding.

AR: We appreciate the reviewer's suggestion to expand the in vivo analysis. Due to time constraints, we were unable to include an additional NSCLC model in this study. However, we agree that future investigations incorporating additional in vivo models will be valuable to further validate CIM7's therapeutic potential and have incorporated these interesting remaining questions in the discussion.

5. The authors should improve the evaluation of NCoR1/RAR α destabilization upon CIM7 inhibition in vitro.

AR: We have further illustrated the NCoR1/RAR α destabilization effect in A549 NSCLC cells through proximity ligation assay, shown in **Figure 4B**. As described now on **Page 12**, treatment with CIM7 significantly reduced the amount of signal generated by the RAR α and NCoR1 interaction.

6. Would suggest modifying terms: untransformed, healthy (non-transformed or non-tumorigenic); demonstrates (suggests, opens the way to) a therapeutic strategy; characterized (found) to contribute; sustainment of aerobic glycolysis (sustained)

AR: We have updated the wording throughout the manuscript per the reviewer's suggestions.

30th Apr 2025

Dear Prof. Gavathiotis,

Thank you for submitting your revised study, and please accept my apologies for the delay in getting back to you as one referee needed more time to complete his/her report. We have now received the three reports, and as you will see below, the referees are satisfied with the revisions. I will therefore be able to accept your manuscript once the following editorial comments are addressed:

1/ Manuscript text:

- Please remove the blue font text and only keep in track changes mode any new modification.
- There is a name discrepancy between the manuscript and the submission system: Olaya Santiago-Fernández vs. Olaya Santiago-Fernández; please check and correct where needed.
- An email bounced for anamaria.cuervo@einsteinmed.edu, please check and correct.
- "Materials and Methods" should be renamed "Methods":
 - o Please indicate the origin and age of the C57BL/6J mice.
 - o A callout is incomplete in the Reagents and Tools table ("Appendix Table")
- The Data Availability Section should be placed after the Methods and before the Acknowledgements. Please note that the Data Availability Section is restricted to new primary data that are part of this study, not previously published datasets. Please include URLs for your deposited data, and note that they must be publicly available before acceptance of the manuscript.
- Acknowledgements: the information provided in the manuscript should match the information provided in the submission system. Currently, there is a discrepancy between P30AG038072 vs P30AG03807, and the Rainwaters Foundation, the JPB foundation, Hevolution foundation, the Irma T. Hirschl Trust Career Award, Robert and Renee Belfer; AG023475, AG084192, T32AGAG023475 are missing in the submission system.

2/ Figures:

- The blots in Figure 4C and D, as well as in Appendix Figure S11D, are over-contrasted (also in the source data). Please check your source data and inform us of any contrast enhancements that may have been applied. Please provide the original, unprocessed source data in its pre-enhancement state, preferably in the 16-bit TIFF format as captured from the original imaging system. Additionally, we request that you confirm whether any adjustments were made to brightness, contrast, or gamma levels. If any modifications were applied, please submit a version of the blots without these adjustments for comparison.
- Your manuscript contains error bars based on $n=2$ (figures 3E, G). Please use scatter blots showing the individual datapoints in these cases. The use of statistical tests needs to be justified.
- Please note that if you want to make some of your Appendix figures EV figures, we can now accommodate more than 5 EV figures.
- Please carefully check the figure referenced in the text: "Supplementary" (Methods, Tables, Figures, information files, etc.) should be removed and updated to the correct callouts; there is a missing a callout for Appendix Table S3; the callout for Appendix Table 2 needs correction - missing "S"
- Appendix: please add page numbers on the title page for each Appendix item. "Supplementary" titles need to be removed throughout the file as "Appendix" nomenclature is used.
- Please address the queries from our copy editors in the figure legends:
 1. Please make sure that exact p values are provided wherever possible.
 2. Please define the annotated p values ****/****/**/* as well as provide the exact p-values for the same in the legend of figure 7B as appropriate.
 3. Please indicate the statistical test used for data analysis in the legend of figure 4G

3/ Source Data: please upload as one zip folder per figure and each folder should have separate files, one file per each panel.

4/ Checklist: Please fill in the line on inclusion/exclusion criteria (experimental study design and statistics).

5/ I slightly edited your synopsis text, please let me know if you agree with the following or amend as you see fit:

"NCoR1/RAR α interaction was identified as a target for selective chaperone-mediated autophagy (CMA) inhibition in lung cancer. A first-in-class small molecule CMA inhibitor targeting this interaction was developed, CIM7, that led to reduced tumor growth.

- The NCoR1/RAR axis transcriptionally regulates chaperone-mediated autophagy in non-small cell lung cancer (NSCLC).
- Small molecule CIM7 binds RAR and disrupts the NCoR1/RAR interaction promoting the transcriptional downregulation of CMA network genes.
- CIM7 selectively inhibits CMA in NSCLC cells compared to non-tumorigenic cells, with no effect on macroautophagy.
- CIM7 affects viability in NSCLC cells and inhibits human NSCLC tumor growth in vivo."

Thank you for providing a nice visual abstract. Please upload it as a separate file in jpeg, TIFF or png format: 550 pixels wide x

300-600 pixels high

6/ As part of the EMBO Publications transparent editorial process initiative (see our Editorial at <http://embomolmed.embopress.org/content/2/9/329>), EMBO Molecular Medicine will publish online a Review Process File (RPF) to accompany accepted manuscripts.

This file will be published in conjunction with your paper and will include the anonymous referee reports, your point-by-point response and all pertinent correspondence relating to the manuscript. Let us know whether you agree with the publication of the RPF and as here, if you want to remove or not any figures from it prior to publication.

I look forward to receiving your revised manuscript.

Yours sincerely,

Lise Roth

***** Reviewer's comments *****

Referee #1 (Comments on Novelty/Model System for Author):

The authors have carefully addressed my comments with adequate changes in the MS. I have no further comments.

Referee #2 (Remarks for Author):

I would like to thank the authors for taking my prior concerns and suggestions seriously. I am satisfied with the results and reasoning behind the additional experiments performed to address open issues.

Referee #3 (Remarks for Author):

I appreciate the authors' efforts in addressing the reviewers' concerns to the best of their abilities. I would have still liked further demonstration of the impact of expressing a resistant mutant protein on the biological response to the inhibitor, but this could likely be done in the future, ahead of further translational investments.

The authors addressed the remaining editorial issues.

16th May 2025

Dear Prof. Gavathiotis,

Thank you for submitting your revised files. I am pleased to inform you that your manuscript is accepted for publication and is now being sent to our publisher to be included in the next available issue of EMBO Molecular Medicine!

Yours sincerely,

Lise Roth
